# Elucidating molecularly stratified single agent, and combination, therapeutic strategies targeting MCL1 for lethal prostate cancer

Metastatic castration-resistant prostate cancer (mCRPC) is a lethal disease requiring additional therapeutic strategies. MCL1, an anti-apoptotic BCL2 family member, promotes cancer-cell survival, but its role in mCRPC remains poorly understood. Here, we characterise MCL1 in multiple mCRPC biopsy cohorts and patient-derived models, assessing responses to MCL1 inhibition. *MCL1* copy number gain (14%–34%) correlates with increased MCL1 expression and worse outcomes. MCL1 inhibition exhibits anti-tumour effects in *MCL1*-gained mCRPC models. Co-inhibition of MCL1 and AKT induces cancer-specific cell death in PTEN-loss/PI3K-activated models in vitro and in vivo, modulating BAD-BCLXL and BIM-MCL1 interactions, with durable anti-tumour activity in models with AKT inhibitor acquired resistance. Finally, CDK9-mediated MCL1 downregulation combined with AKT inhibition recapitulates these findings, providing further opportunities for clinical translation. These data support early phase clinical trials targeting MCL1, both as monotherapy for *MCL1*-gained mCRPC, and in combination with AKT inhibition for PTEN-loss/PI3K-activated mCRPC.

Despite significant progress in the treatment of metastatic castration-resistant prostate cancer (mCRPC), the disease remains lethal, with a median overall survival (OS) ranging from 2 to 3 years[1–3]. Additional therapeutic strategies are urgently required to improve the outcome for patients with advanced disease. Cell death escape and circumvention of apoptosis is a hallmark of cancer, driving tumourigenesis and therapeutic resistance[4]. Intrinsic apoptosis is tightly regulated by a multitude of pro- and anti-apoptotic B-cell lymphoma 2 (BCL2) proteins, which interact on the mitochondrial outer membrane. Among the anti-apoptotic members, Myeloid Cell Leukaemia 1 (MCL1) has been shown to promote cancer cell survival in various malignancies, including prostate cancer (PC)[5,6]. Moreover, *MCL1* is one of the most frequently amplified genes in human malignancies, with amplification found in 10.9% of cancers across multiple tumour types[7].

Beyond its anti-apoptotic function, it is postulated that MCL1 is involved in other key cancer cell processes such as cell cycle regulation, DNA repair, metabolism and calcium homoeostasis[8]. In addition, MCL1 may play a role in the development of resistance to androgen deprivation therapy (ADT) and DNA damage-inducing chemotherapies in PC[9,10], although further studies are required in this space. Moreover, MCL1 is thought to promote cell survival in therapy-induced senescent PC cells, a phenotype that contributes to tumour proliferation and metastasis via the Senescence-Associated Secretory Phenotype and has recently been reported to play a key role in endocrine treatment resistance[11,12]. Taken together, MCL1 holds promise as a therapeutic target to induce apoptotic cell death, rather than growth inhibition in mCRPC, potentially mitigating the risk of acquired therapeutic resistance.

✉ e-mail: johann.de-bono@icr.ac.uk; adam.sharp@icr.ac.uk

Importantly, small molecule BH3 mimetics have been developed to inhibit the anti-apoptotic BCL2 proteins, by binding to their hydrophobic groove[5]. Although developing high-affinity MCL1-specific BH3 mimetics has been challenging, there are now multiple agents under investigation in first-in-human clinical trials, with a focus on hematopoietic cancers. Despite these advances, preclinical studies have shown limited anti-tumour activity for single protein targeting, potentially due to functional redundancies among the anti-apoptotic BCL2 family members, including BCLXL[13–15]. Indeed, it has been shown that PC is co-dependent on MCL1 and BCLXL and that both proteins must be inhibited to induce apoptosis[16,17], which suggests that targeting pathways regulating the expression or activity of BCLXL, in combination with an MCL1 inhibitor, may drive apoptotic cell death. Taken together, the identification of predictive biomarkers that enrich for response to single agent MCL1 inhibition in PC, and the discovery of rational MCL1 inhibitor combination strategies that drive apoptosis in PC, would address an urgent unmet clinical need.

Here, we show that *MCL1* copy number gain and amplification occur early in lethal PC evolution and associate with worse clinical outcomes. We demonstrate that single agent MCL1 inhibition has marked anti-tumour activity in PC models with *MCL1* copy number gain, and that MCL1 targeting (directly through BH3 mimetics or indirectly through CDK9 inhibition) synergises with AKT inhibition to deliver cancer-specific killing in PTEN-loss/PI3K-activated PC models, with ongoing anti-cancer activity in PC models with acquired AKT resistance. These data support the evaluation of therapeutic strategies targeting MCL1, as a single agent for PC with *MCL1* copy number gains, and in combination with AKT inhibition in PTEN loss/PI3K-activated PC, within 'proof of concept, proof of mechanism' early phase clinical trials.

## Results

### *MCL1* copy number gains are common in lethal prostate cancer, occur early in tumour evolution, and increase with castration-resistance

To investigate the genomic status of *MCL1* in PC, we examined gene copy number alteration data from multiple PC biopsy cohorts. *MCL1* is located on chromosome 1q21 and falls within a commonly amplified multi-gene cassette (Fig. 1A, B). *MCL1* amplification (high level copy number gain) was rarely seen in primary PC from the The Cancer Genome Atlas (TCGA) cohort (1%), the majority of which are cured after radical treatment (Fig. 1A, B). In advanced CRPC biopsies from the Stand Up To Cancer/Prostate Cancer Foundation (SU2C/PCF) cohort, *MCL1* amplification was more common (14%) (Fig. 1A, B). Analysis of matched same-patient CSPC and CRPC biopsies from the Royal Marsden Hospital (RMH) cohort showed a similar frequency of *MCL1* amplification (CSPC 11.36%, CRPC 13.64%), suggesting this event occurs early in tumour evolution and associates with CRPC development (Fig. 1A, B). Despite this, a higher proportion of CRPC biopsies (34.09%) exhibited *MCL1* copy number gain when compared to matched, same patient, CSPC biopsies (15.91%) (Fig. 1A, B), suggesting this might be part of an adaptive response emerging with treatment resistance. In addition, consistent with treatment resistance emergence, androgen receptor (*AR*) locus (Xq12) amplification was almost exclusively observed in CRPC biopsies (36%) when compared to CSPC biopsies from the same patient (Fig. 1A, B).

### *MCL1* copy number gains associate with worse clinical outcome

Having shown that genomic gains of *MCL1* are common in mCRPC, we evaluated their association with prognostic markers and clinical outcome. In primary PC (TCGA cohort), *MCL1* copy number gain/amplification was associated with significantly shorter progression-free interval (PFI) (HR 1.93, CI 1.05-3.53, $p = 0.03$) (Fig. 1C), a higher Gleason score and pathological tumour stage, and an increased chance of lymph node involvement at diagnosis (Fig. 1D). Furthermore, in a phase II trial evaluating 6 months of neoadjuvant ADT and enzalutamide prior to radical prostatectomy in men with localised intermediate- or high-risk PC[18], *MCL1* somatic copy number gains at baseline were associated with an inferior pathologic response to therapy. Four out of 37 tumours (11%) had *MCL1* copy number gain/amplification, all of which were classified as pathologic incomplete or non-responders, providing evidence that gains in *MCL1* may be associated with worse clinical outcome (Fig. 1E, F). In mCRPC, analysis of plasma cell-free DNA from patients enrolled in the FIRSTANA and PROSELICA clinical trials showed that subjects whose tumours have *MCL1* copy number gain/amplification have shorter overall survival (HR 1.35, CI 0.95–1.90, $p = 0.09$) (Fig. 1G). *MCL1* copy number gain/amplification was also associated with prior exposure to second-generation hormonal therapies (abiraterone and/or enzalutamide), whereas no significant differences were observed in terms of prior radical prostatectomy or radiotherapy (Supplementary Fig. 4A). *MCL1* copy number gain/amplification did not, however, associate with taxane response highlighting its potential role as a prognostic, rather than predictive biomarker, in this specific context (Supplementary Fig. 4B). When considering prognostic markers in CRPC, *MCL1* copy number gain/amplification associated with higher PSA and lower haemoglobin (Fig. 1H), but not with the other putative prognostic biomarkers analysed herein (Supplementary Fig. 4C–E). Overall, these data suggest that *MCL1* copy number changes as PC progresses to mCRPC and is associated with worse clinical outcome.

### MCL1 RNA and protein are highly expressed in mCRPC

Transcriptomic analysis revealed an association between *MCL1* copy number gain/amplification and increased RNA expression in the TCGA and SU2C/PCF cohorts, as well as in the rapid autopsy University of Washington/Fred Hutchinson (UW/FH) CRPC cohort which also demonstrated highly concordant intra-patient, inter-tumour, *MCL1* expression[19] (Fig. 1I, J). The same trend was observed in the RMH cohort although not statistically significant ($p = 0.301$) (Fig. 1I). These data underline the functional significance of *MCL1* copy number gain/amplification. However, a substantial number of tumours with unaltered *MCL1* copy number exhibit high RNA expression, suggesting that additional mechanisms can also drive *MCL1* overexpression (Fig. 1I, J). To investigate the correlation between *MCL1* RNA and protein levels, Immunohistochemistry (IHC) was performed on 30 mCRPC biopsies from the RMH mCRPC cohort (Fig. 1K). MCL1 staining exhibited a cytoplasmic and granular pattern, which is consistent with its mitochondrial localisation (Fig. 1L). *MCL1* RNA expression was found to be positively correlated with MCL1 protein cytoplasmic optical density values ($n = 30$, Spearman $r = 0.482$, $p = 0.007$) (Fig. 1K, L), suggesting that elevated RNA levels led to functionally relevant protein expression. While a range of expression levels was observed, none of the tumours exhibited loss of MCL1 protein or RNA (Supplementary Fig. 5A). Cytoplasmic MCL1 H-score was found to be correlated with MCL1 cytoplasmic optical density levels (Supplementary Fig. 5B). In addition, higher tumour MCL1 expressions (H-Score ≥100) was associated with shorter overall survival in CRPC patients (HR 2.09, 95% CI 0.85–5.12, $p = 0.11$) (Supplementary Fig. 6). These data confirm that *MCL1* copy number changes associate with MCL1 RNA and protein, which are broadly expressed in advanced PC, further underlying its potential importance in PC biology.

### Response to MCL1 inhibition is enriched in PC models with MCL1 copy number gain

Following the demonstration that *MCL1* copy number was gained in a subset of mCRPC patients and linked to poor prognosis, we explored MCL1 targeting as a therapeutic strategy, hypothesising that *MCL1* copy number gain would indicate dependency and enrich response to MCL1 inhibition, as suggested in other tumour types[7,20,21]. We first characterised seven prostate-derived cell lines and found that 22Rv1

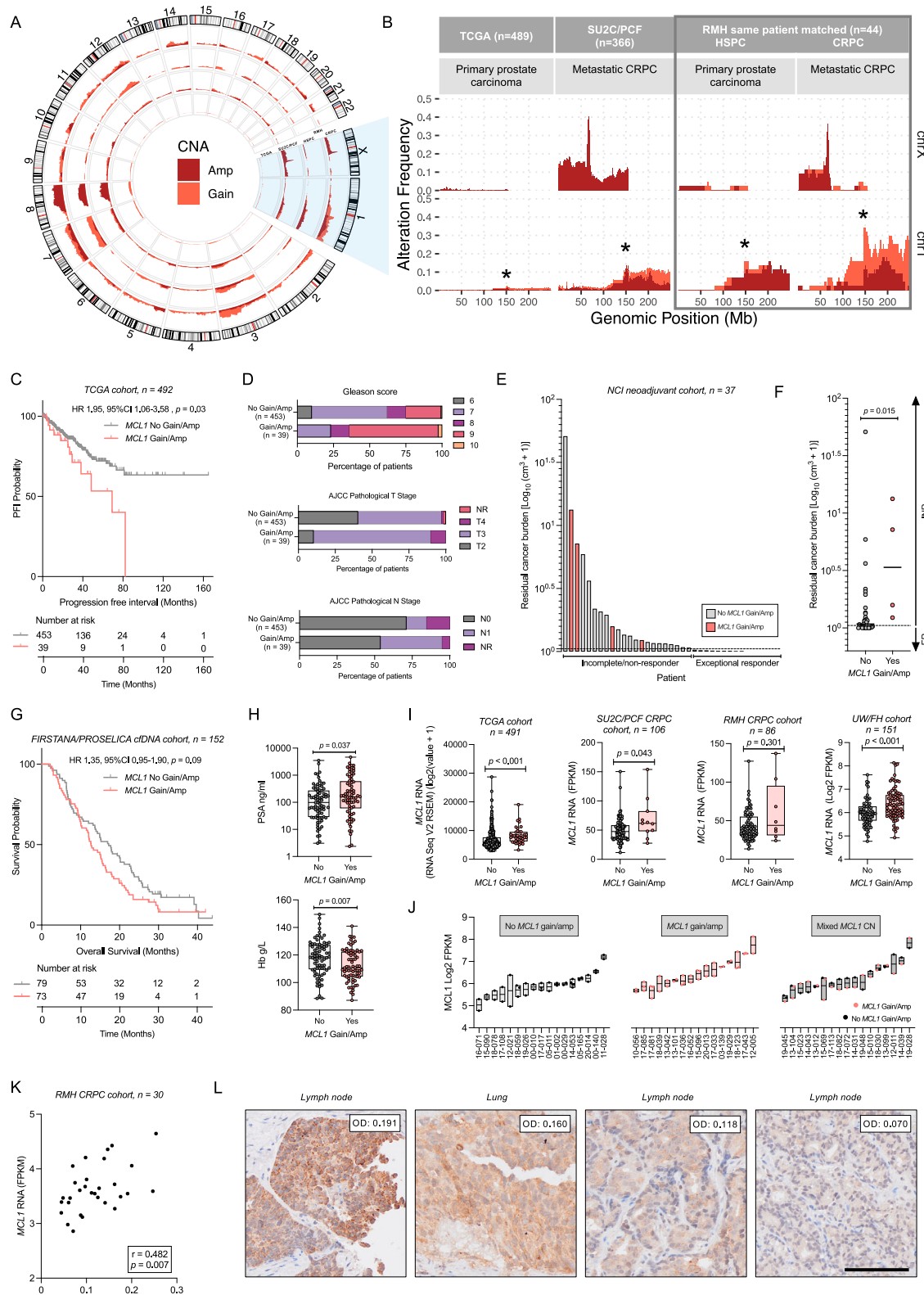

and DU145 harbour MCL1 copy number gain; however, only 22Rv1 exhibited markedly elevated MCL1 protein levels, suggesting additional mechanisms regulate MCL1 protein levels in DU145 (Fig. 2A–C). Among the tested cell lines, 22Rv1 showed the highest sensitivity to MCL1 inhibition, as evidenced by increased caspase 3/7 activity and reduced cell viability at 24 h and 6 days (Fig. 2D–F). LNCaP95 displayed a comparable response, indicating that sensitivity to MCL1 inhibition can be influenced by factors beyond basal MCL1 copy number or

protein levels (Fig. 2D-F). The lack of response in DU145 may stem from its reported apoptosis resistance due to BAX deficiency[22] (Fig. 2D–F). Next, we evaluated the apoptotic response to MCL1 inhibition of five CRPC PDX-O models with varying MCL1 copy numbers and protein levels (Fig. 2G–K). Both AZD5991 and S63845 increased caspase activity (6 h) and reduced organoid viability (at 96 h) in the MCL1 copy number gain models CP50c and CP267c, which exhibited high MCL1 protein expression (Fig. 2G–K and Supplementary Fig. 7). In contrast,

**Fig. 1 | MCL1 (1q21) copy number gain/amplification is common in CRPC, occurs early in tumour evolution, and associates with poor prognosis. A, B** Copy number alterations in TCGA (primary PC), SU2C/PCF (CRPC), and RMH (matched HSPC and CRPC) cohorts. **A** Chromosomes 1 (MCL1) and X (AR) are highlighted; **B** MCL1 locus (1q21) marked with an asterisk. *CNA* copy number alterations. *Gain*: copy number gain; *Amp* amplification. **C** Kaplan–Meier progression-free interval curves in TCGA cohort (*n* = 492) comparing patients with MCL1 gain/amplification vs those without. Hazard ratio (HR) with 95% confidence intervals and *p*-value from log-rank test shown. **D** Comparison of Gleason score, American Joint Committee on Cancer (AJCC) pathological tumour (T) and node (N) stages between patients with MCL1 gain/amplification vs those without in TCGA (*n* = 492). **E, F** Residual cancer burden in patients from the NCI neoadjuvant cohort (*n* = 37), treated with 6 months ADT and enzalutamide, stratified by MCL1 copy number (gain/amplification vs no gain/amplification). **F** Two-tailed Mann-Whitney U test; median indicated. *INR* Incomplete/non-responder, *ER* exceptional responder. **G** Kaplan–Meier overall survival for patients from FIRSTANA and PROSELICA trials (*n* = 152), stratified by MCL1 gain/amplification on baseline ctDNA. HR with 95% confidence intervals and *p*-values from log-rank test shown. **H** Baseline PSA and Hb in patients from FIRSTANA and PROSELICA trials (*n* = 152), stratified by MCL1 gain/amplification on baseline ctDNA; Two-sided Mann-Whitney *U* test used. Box shows the interquartile range (IQR), line indicates the median, and whiskers the min/max values. **I** MCL1 RNA expression by copy number status across TCGA (*n* = 491), SU2C/PCF (*n* = 106), RMH (*n* = 86), and UW/FH (*n* = 151) cohorts. Box (IQR), line (median), whiskers (min/max). Two-tailed Mann-Whitney *U* test. **J** *MCL1* RNA expression in UW/FH cohort split by homogeneous (with and without *MCL1* copy number gain/amplification) and heterogeneous copy number (*n* = 115 tumours from 50 patients with >1 tumour). Mean with min–max floating bars shown. *NA* Not available. **K** MCL1 IHC in 30 mCRPC biopsies from RMH cohort with RNA-seq data. Spearman correlation between MCL1 RNA and protein (OD) levels. **L** Representative IHC images from the 30-biopsy set. Scale bar = 100 μm. Source data are provided as a Source Data file.

models with unaltered MCL1 copy number and lower protein levels (CP142c, CP253c and CP336c) were minimally impacted (Fig. 2G–K and Supplementary Fig. 7). The effects of MCL1 inhibition demonstrated herein reflect short-term apoptosis; thus, the potential impact on PDX-O proliferation may require extended incubation times. These data suggest that, at least a subset of CRPC cells with high MCL1 levels may be sensitive to single agent MCL1 inhibition.

## MCL1 inhibition synergises with PI3K inhibition to trigger apoptosis in CRPC cells

In order to enhance and/or broaden the effect of MCL1 inhibition, we conducted a screen testing 166 FDA-approved drugs (at 5 μM), which included AR inhibitors (apalutamide, darolutamide and enzalutamide), as single agents and in combination with AZD5991 (1 μM). We evaluated the impact on apoptosis induction (6 h) and cell viability (24 h) in the C4-2 and LNCaP95 CRPC models, as well as in the immortalised prostate epithelium cell line PNT2. Specifically, PI3K pathway inhibitors (copanlisib, idelalisib, duvelisib and alpelisib) showed synergistic effects, decreasing cell viability, and inducing apoptosis in C4-2 and LNCaP95, but not in PNT2, suggesting that this combination selectively kills tumour cells (Fig. 3A, B and Supplementary Fig. 8). These results were validated using copanlisib and buparlisib at four different concentrations, revealing a dose-dependent induction of caspase 3/7 activity (6 h), accompanied by a dose-dependent decrease in cell viability (24 h) when combined with AZD5991 (Supplementary Fig. 9A, B). Taken together, these data demonstrate that MCL1 and PI3K pathway inhibitors synergise to induce cancer cell specific death in PC models.

## MCL1 inhibition synergises with AKT inhibition to trigger apoptosis in CRPC cells

Given that AKT inhibition has shown anticancer activity in mCRPC clinical trials, we next tested whether direct inhibition of AKT could also elicit a robust synergistic effect when combined with MCL1 inhibition[23,24]. The ZIP synergy scores obtained in response to the co-inhibition of AKT (capivasertib and ipatasertib) surpassed those observed with PI3K inhibitors (copanlisib and buparlisib) when combined with the MCL1 inhibitor AZD5991 in LNCaP95 and C4-2 cells (Fig. 3C, D and Supplementary Fig. 10). These results were further validated independently in an all vs all screen of 42 agents with 30 distinct mechanisms of action (861 different combinations), carried out at the National Cancer Institute (NCI). Specifically, the MCL1 inhibitor S63845 synergised with the PI3K inhibitors copanlisib and paxalisib, as well as with the AKT inhibitors capivasertib and ipatasertib to decrease cell viability at 48 h in the LNCaP95 cells (Supplementary Fig. 11A, B). The induction of apoptosis in response to the combination of AZD5991 with capivasertib and ipatasertib was further validated by protein analyses at 6 h, with cleavage of PARP, caspase 7 (determined by western blot) and caspase 3 (determined by both western blot and IHC) in LNCaP95 and

C4-2 cells (Fig. 3E, F and Supplementary Fig. 12). These results were orthogonally validated using siRNAs to target AKT1/2/3 and MCL1, confirming "on-target" pharmacological mechanism of action (Supplementary Fig. 13). Overall, these findings demonstrate that, beyond the synergy observed with MCL1 and PI3K inhibition, the combined impact of MCL1 and AKT inhibition is more pronounced, leading to a heightened induction of apoptosis in PC models.

## The synergistic anti-tumour activity of AKT and MCL1 inhibition is mediated through alterations in BAD-BCLXL and BIM-MCL1 interactions

Given the essential role of BH3-only proteins in the control of the intrinsic apoptosis pathway, through the regulation of the activity of anti-apoptotic BCL2 family members[25], we conducted a pro-apoptotic BCL2 family siRNA screen in C4-2 and LNCaP95 cells in response to capivasertib, ipatasertib, AZD5991, and their combinations. Specifically, individual silencing of the BH3-only proteins BIM and BAD, and the mitochondrial pore forming effector BAK partially prevented the induction of apoptosis (assessed by caspase 3/7 assay, and cleaved caspase 3 and cleaved PARP protein levels) and the reduction in cell viability (assessed by CellTiter-Glo) induced by the combined treatments in LNCaP95 and C4-2 cells (Supplementary Fig. 14 and Supplementary Fig. 15). When silencing LNCaP95 and C4-2 cells with a combination of BIM, BAD and BAK siRNAs, a near-complete rescue was achieved (Fig. 4A, B), suggesting that BIM, BAD and BAK are co-operative in the molecular mechanism underlying the synergistic effects driven by AKT and MCL1 co-inhibition. Furthermore, western blot analyses showed no significant changes in BAK, BAD, or BIM protein levels in response to capivasertib, ipatasertib, AZD5991, or their combinations (Fig. 4C). However, phosphorylation of BAD at Ser136 decreased with AKT inhibitors (capivasertib and ipatasertib) and combined treatments (Fig. 4C). Co-immunoprecipitation experiments revealed that ipatasertib increased BAD-BCLXL and BIM-MCL1 binding, with the latter being more evident when BIM was pulled down, in both LNCaP95 and C4-2 lines (Fig. 4D). Additionally, AZD5991 (as a single treatment or combined with ipatasertib) disrupted BIM-MCL1 and BAK-MCL1 binding (Fig. 4D), in keeping with its mechanism of action[26]. These results were orthogonally validated by PLA in C4-2 cells (Supplementary Fig. 16). Taken together, these data suggest that the induction of apoptosis by MCL1 and AKT inhibition is mediated by BAD-driven sequestration of BCLXL and the displacement of BIM from MCL1.

## AKT and MCL1 co-inhibition triggers BAK-dependent apoptosis in PTEN-loss/PI3K-activated CRPC cells

To identify potential predictive biomarkers, we assessed phospho-BAD, total BAD, BIM and BAK levels and examined responses to AKT and MCL1 co-inhibition in seven cell lines with different molecular backgrounds (Fig. 5A, B and Supplementary Fig. 17A, B). Synergistic

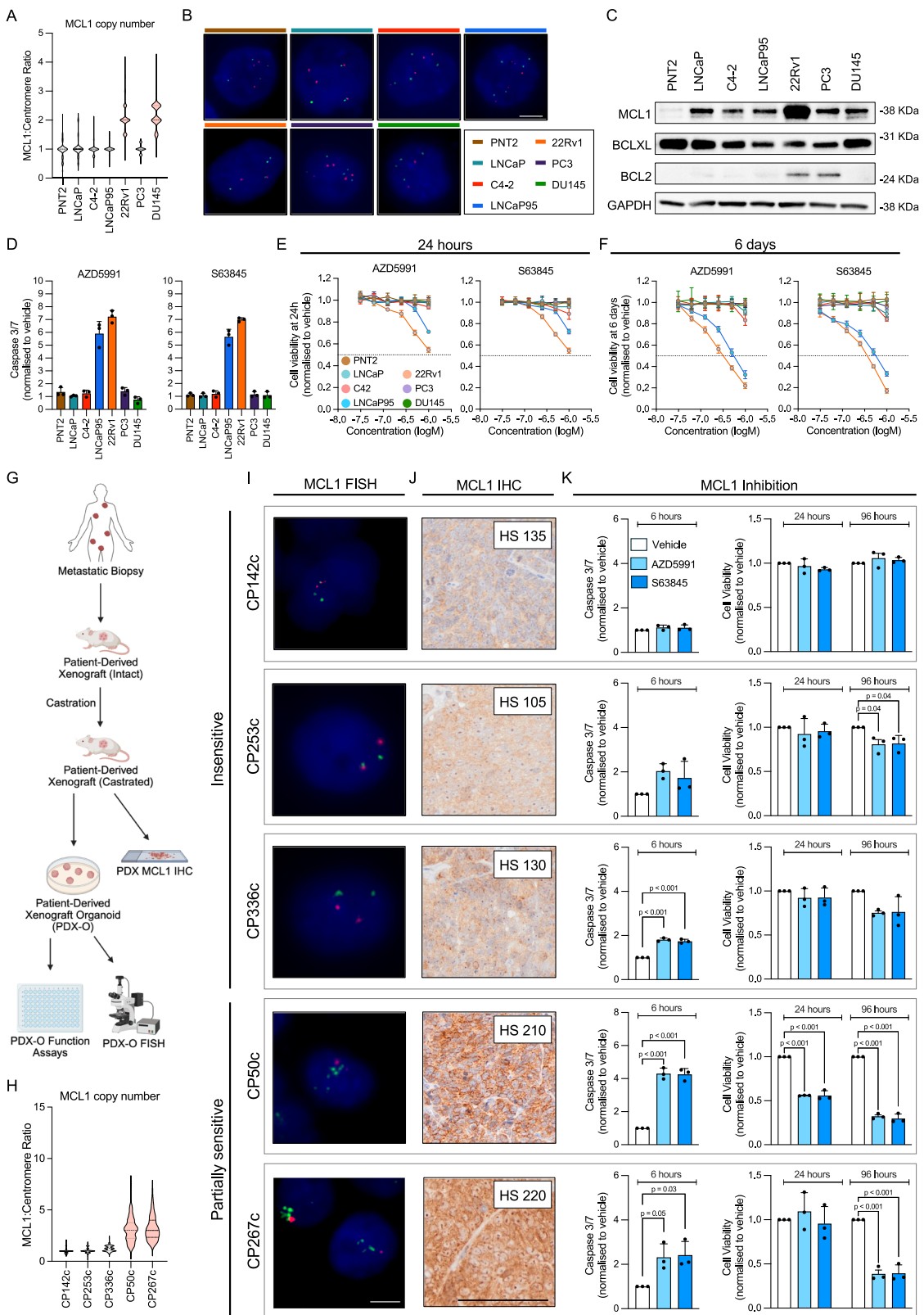

caspase 3/7 activation and reduced viability were observed in PTEN loss lines (LNCaP, C4-2, LNCaP95) and the PIK3CA-mutated line 22Rv1, which exhibited higher levels of phospho-BAD, total BAD, BIM, and BAK (Fig. 5A, B and Supplementary Fig. 17A, B). In contrast, no synergy was observed in PNT2, DU145 (both PTEN WT) nor in PC-3 despite harbouring PTEN loss (Fig. 5B and Supplementary Fig. 17A, B). Specifically, PC-3 cells have low levels of phospho-BAD, BIM and BAK, and

additionally we showed that they rely on BAX rather than BAK (as opposed to the responder LNCaP95) to undergo apoptosis (Fig. 5B and Supplementary Fig. 17C, D). This suggests that undergoing apoptosis in a BAK-dependent manner is required for PC cells to be sensitive to AKT and MCL1 co-inhibition.

We next analysed CRPC PDX-Os with PI3K/AKT pathway aberrations, including CP50c (AKT1 amplification), which shows the highest

**Fig. 2 | Prostate cancer models with MCL1 copy number gain are sensitive to MCL1 inhibition. A** Violin plot of *MCL1*:Centromere (1p12) ratio in PNT2, LNCaP, C4-2, LNCaP95, 22Rv1, PC-3, and DU145 cell lines (*n* = 100 cells each). **B** Representative MCL1 FISH images (1 biological replicate; *n* = 100 cells). Scale bar = 10 μm. **C** MCL1, BCLXL, and BCL2 protein levels assessed by western blot in the same cell lines. GAPDH used as loading control. **D** Caspase 3/7 activity following AZD5991 (1 μM, left) or S63845 (1 μM, right) for 6 h using Caspase-Glo 2D. Mean ± Standard error (SEM) from three biological replicates, each with three technical replicates. Cell viability assessed by CellTiter-Glo 3D after 24 h (**E**) and 6 days (**F**) following treatment with six concentrations (1 μM highest, two-fold dilutions) of AZD5991 (left) or S63845 (right). Mean ± SEM from three biological replicates, each with three technical replicates is shown. Data depicted as fold change relative to vehicle (DMSO). **G** Workflow for mCRPC patient-derived xenograft (PDX) organoids

(PDX-Os). Created in BioRender. Jimenez Vacas (2025) https://BioRender.com/7s2gcp5. **H** Violin plot of *MCL1*:Centromere (1p12) ratios in CP253c, CP336c, CP50c, CP267c (*n* = 100 cells each) and CP142c (*n* = 50). Median and IQR shown. One-way ANOVA with Tukey's post-hoc test was performed. **I** Representative MCL1 FISH images of single PDX-O cells (1 biological replicate). Median *MCL1*:Centromere ratio shown. Scale bar = 5 μm. **J** Representative MCL1 IHC images from PDX tissues (single passage). Cytoplasmic H-score shown. Scale bar = 100 μm. **K** Caspase 3/7 activity (6 h) and organoid viability (24 and 96 h) in PDX-Os treated with AZD5991 or S63845 (1 μM), measured with Caspase-Glo 3D and CellTiter-Glo 3D. Mean ± SEM from three biological replicates, each with five technical replicates. Data shown as fold change relative to vehicle (DMSO). One-way ANOVA with Dunnett post-hoc test was performed. Source data are provided as a Source Data file.

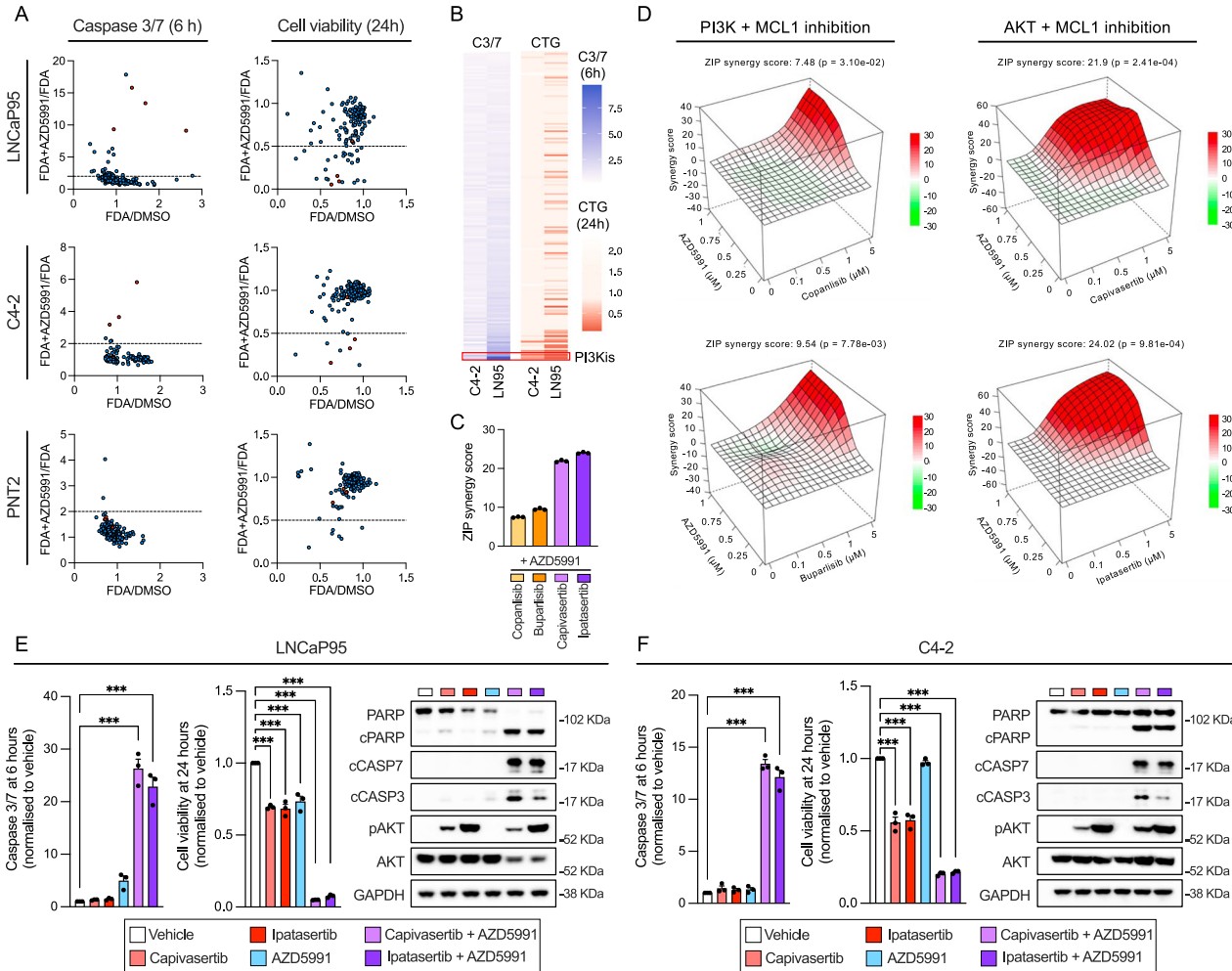

**Fig. 3 | PI3K/AKT pathway inhibition drives apoptosis when combined with MCL1 inhibition in LNCaP95 and C4-2 cell lines. A** FDA-approved drug screen (*n* = 166 drugs) in LNCaP95, C4-2, and PNT2 cells, in the presence and absence of AZD5991 (1 μM). FDA approved drugs used at 5 μM. Caspase 3/7 activity (6 h; Caspase-Glo 2D) and cell viability (24 h; CellTiter-Glo 2D) were assessed. The x-axis shows the effect of each FDA drug alone (FDA/DMSO), and the y-axis shows the added effect of combining AZD5991 with each drug (FDA + AZD5991/FDA). Experiment performed as a single biological replicate in technical singlets. PI3K inhibitors are highlighted in red. **B** Heatmaps showing the combined effect of AZD5991 with FDA drugs on caspase activation (6 h) and cell viability (24 h) in LNCaP95 and C4-2, normalised to the effect on the normal-like prostate cell line PNT2. Blue and red indicate increased or decreased caspase 3/7 activity or cell viability, respectively. **C** Zero Interaction Potency (ZIP) synergy scores comparing

combinations of AZD5991 with PI3K inhibitors (copanlisib, buparlisib) versus AKT inhibitors (capivasertib, ipatasertib) in LNCaP95 cells. Data are mean ± SEM from three biological replicates (each with three technical replicates). One-way ANOVA with Tukey's post-hoc test. **D** ZIP synergy surface plots for AZD5991 combined with PI3K (copanlisib, buparlisib; left) or AKT (capivasertib, ipatasertib; right) inhibitors in LNCaP95. Caspase 3/7 activity (6 h), cell viability (24 h), and protein expression of cleaved PARP, cleaved caspase 3, and phosphorylated AKT (Ser473) following treatment with DMSO, AZD5991 (1 μM), capivasertib (1 μM), ipatasertib (1 μM), or combinations in LNCaP95 (**E**) and C4-2 (**F**). Bars show mean ± SEM of three biological replicates (three technical replicates). Statistical comparisons performed using one-way ANOVA with Tukey's post-hoc test. Asterisks indicate statistical significance between groups (***p < 0.001). Source data and exact *p* values are provided are provided as a Source Data file.

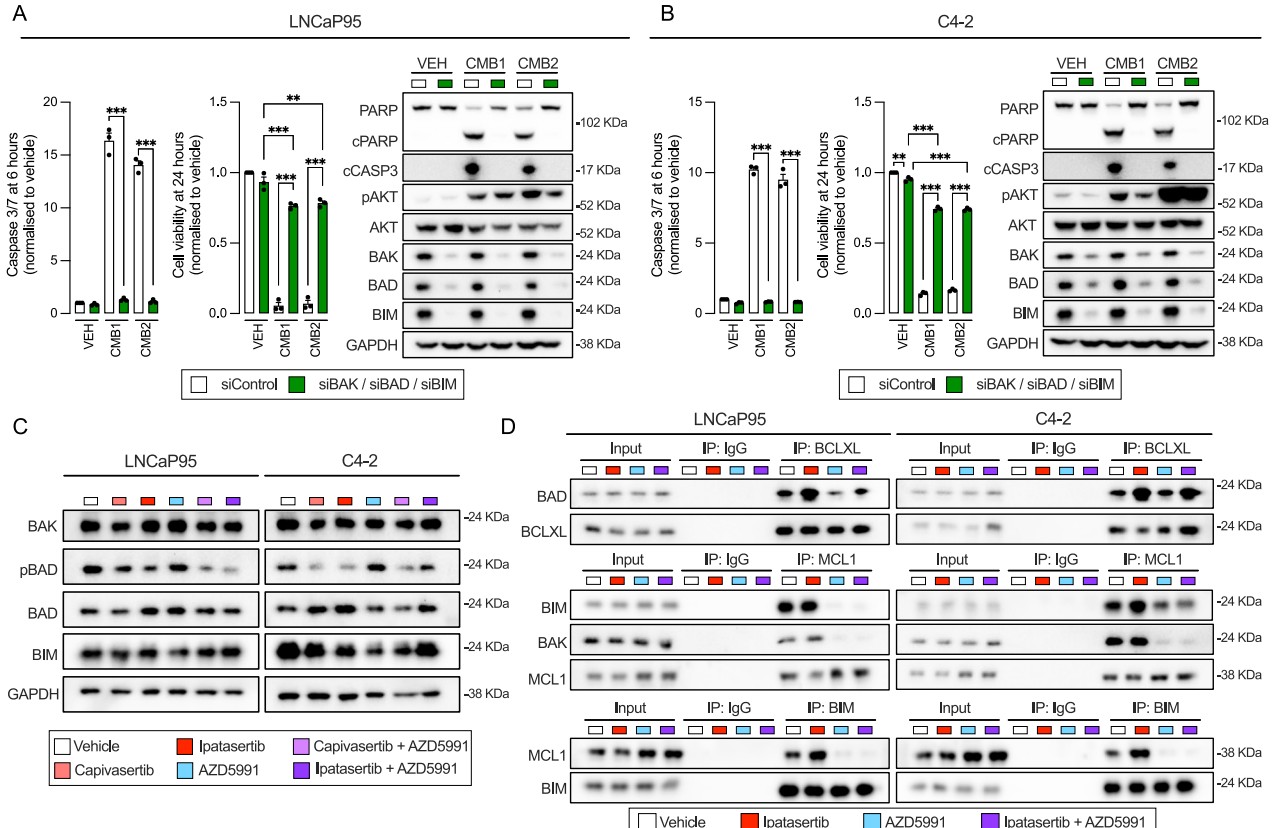

**Fig. 4 | AKT and MCL1 co-inhibition triggers apoptosis through dysregulation of BAD, BIM and BAK interactions in LNCaP95 and C4-2 cell lines.** Impact of combined treatment 1 [CMB1; capivasertib (1 μM) and AZD5991 (1 μM)], and combined treatment 2 [CMB2; ipatasertib (1 μM) and AZD5991 (1 μM)] upon triple knockdown of BAK, BAD, and BIM (using specific OnTarget siRNAs at 25 nM for 72 h) or OnTarget Control siRNA (75 nM, 72 h) on caspase 3/7 (6 h; left panel), cell viability (24 h; middle panel) and protein expression (6 h; right panel) in LNCaP95 (**A**) and C4-2 (**B**). Caspase 3/7 activity, cell viability, and protein expression were measured by Caspase-Glo 3/7 2D, CellTiter-Glo 2D and western blot, respectively. GAPDH was used as a housekeeping. Data represent three biological replicates, each with three technical replicates. Bars indicate the mean ± SEM. Statistical

significance was assessed using one-way ANOVA with Tukey's post-hoc test. Asterisks indicate statistical significance between groups (\*\**p* < 0.01; \*\*\**p* < 0.001). **C** Protein levels of BAK, phosphorylated BAD (Ser136), total BAD and BIM were measured by western blot in LNCaP95 and C4-2 cells treated with DMSO, AZD5991 (1 μM), capivasertib (1 μM), ipatasertib (1 μM), or their combinations. Experiment performed as a single biological replicate. **D** Immunoprecipitation of BCLXL (top), MCL1 (middle), and BIM (bottom) in LNCaP95 and C4-2 cells treated with DMSO, ipatasertib (1 μM), AZD5991 (1 μM), or the combination. BAD, BCLXL, BIM, BAK and MCL1 were blotted. Input and IgG pulldown controls included. Experiment performed as a single biological replicate. Source data and exact *p* values are provided as a Source Data file.

PI3K-AKT pathway score among the SU2C/PCF samples[27], and CP253c, CP267c, and CP336c (PTEN loss by IHC), as well as the PTEN WT model CP142c (Supplementary Fig. 18 and 19). CP142c has significantly lower PI3K/AKT pathway activity than CP50c, CP253c, CP267c and CP336c, which was inferred by the Reactome PI3K/AKT signalling in cancer score and AKT downstream markers including phospho-GSK3B (Ser9) and phospho-PRAS40 (Thr246) (Fig. 5C and Supplementary Fig. 19). Among the five models, CP50c, CP253c, and CP142c were AR-FL and AR-V7 positive, with CP50c and CP253c also ERG positive, indicating TMPRSS2:ERG fusion[28] (Supplementary Fig. 19). Synergistic caspase 3/7 activation and reduced viability were observed in response to ipatasertib combined with MCL1 inhibitors (AZD5991 and S63845) in CP50c, CP253c, CP267c, and CP336c, all of which exhibited higher basal phospho-BAD levels than CP142, where no synergy was observed (Fig. 5D–G). In contrast, BAD, BIM, and BAK levels varied across models (Fig. 5D and Supplementary Fig. 19).

To confirm that PTEN loss is a key determinant of sensitivity to AKT and MCL1 co-inhibition, we treated PC Preclinical Mouse Modelling Platform organoids (ProMPt-Os) with different genomic backgrounds. Only PTEN loss models (PTEN^KO/TP53^WT/MYC^OE and PTEN^KO/TP53^KO/MYC^OE) responded synergistically to the combination of AKT and MCL1 inhibitors, as evidenced by a more pronounced increase in caspase 3/7 activity and reduced viability compared to single-agent

treatments, while no synergy was observed in the PTEN WT model (PTEN^WT/TP53^KO/MYC^OE) (Fig. 6A–D). Notably, both PTEN loss models exhibited higher levels of phospho-BAD (Ser136), BAK, and BIM than the PTEN WT model (Fig. 6E). Basal phosphorylated BAD levels strongly correlated with the degree of response to AKT and MCL1 co-inhibition across all models, suggesting its potential as a predictive biomarker for sensitivity to this combination therapy (Fig. 6F, G). These data demonstrate that co-inhibition of AKT and MCL1 is synergistic and induces BAK-dependent cancer-specific killing in PTEN-loss/PI3K-activated PC models, supporting the clinical evaluation of this therapeutic strategy in a molecularly defined patient population.

**Synergistic inhibition of tumour growth by AKT and MCL1 co-inhibition in vivo**

To evaluate AKT and MCL1 co-inhibition in vivo, we assessed CP253c, a PTEN loss, AR-FL/AR-V7 positive and ERG rearranged PDX model derived from a lymph node biopsy of an mCRPC patient previously treated with abiraterone (Supplementary Fig. 3 and Supplementary Fig. 19), in response to ipatasertib [50 mg/kg, p.o., 5 days/week (Monday–Friday)], S63845 [25 mg/kg, i.v., 3 days/week (Monday, Wednesday, and Friday)], combined treatment, or vehicle. The combined treatment led to a significant suppression of tumour growth,

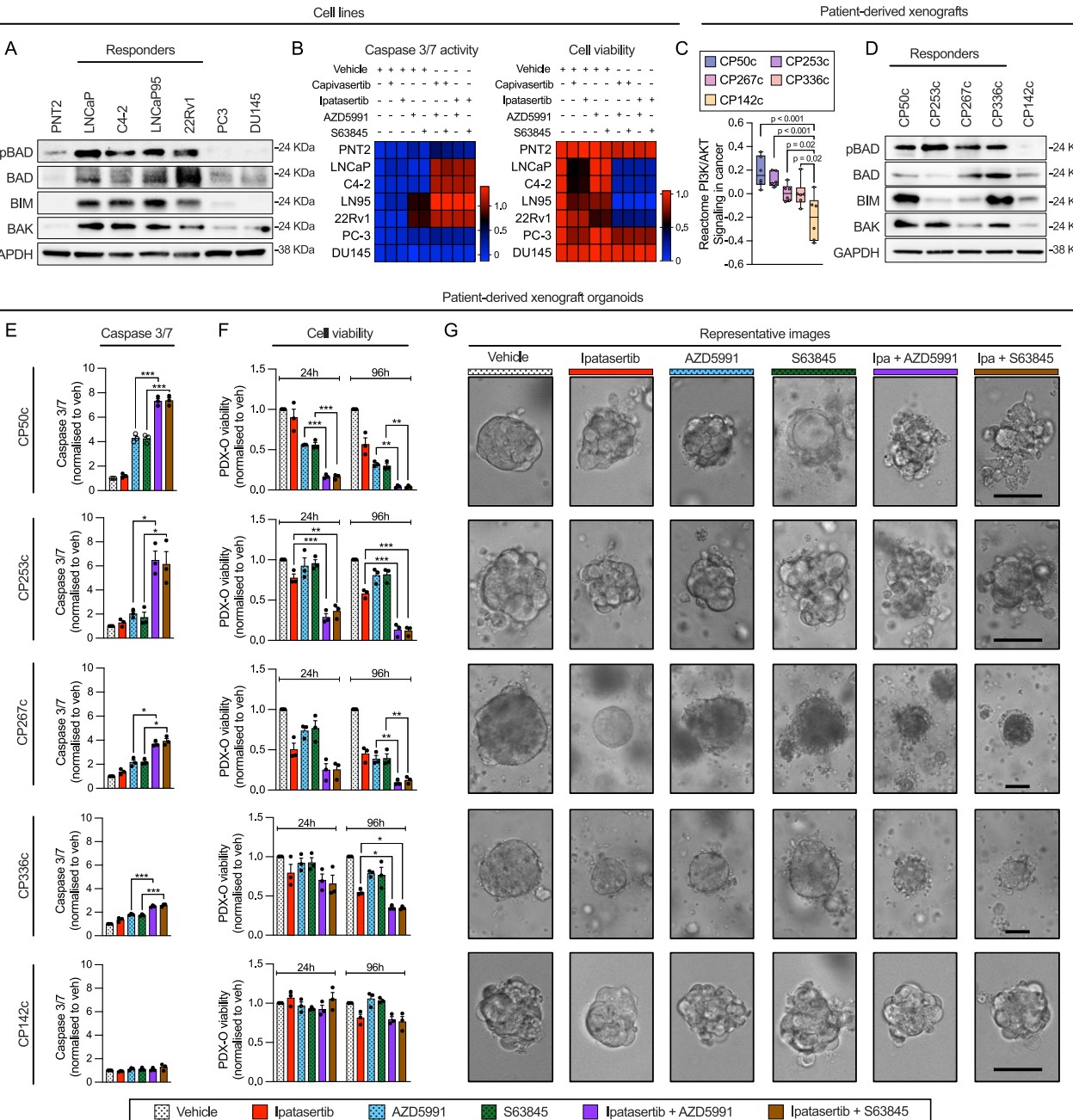

**Fig. 5 | AKT and MCL1 co-inhibition triggers apoptosis in prostate cancer patient-derived xenograft organoids harbouring PI3K/AKT pathway hyperactivating aberrations. A** The protein levels of phosphoBAD (Ser136; pBAD); total BAD, BIM, and BAK in PNT2, LNCaP, C4-2, LNCaP95, 22Rv1, PC3, and DU145 cell lines, with GAPDH as a loading control, were detected by western blot. This experiment was performed as a single biological replicate. **B** Heatmaps showing the effects of capivasertib (1 μM), ipatasertib (1 μM), AZD5991 (1 μM), S63845 (1 μM), and their combinations on caspase 3/7 activity (left panel, 6 h, measured by Caspase-Glo 3/7 assay) and cell viability (right panel, 24 h, measured by CellTiter-Glo 2D assay) in prostate-derived cell lines. Caspase 3/7 activity and cell viability are expressed as fold changes relative to vehicle (DMSO), with caspase data logarithmically transformed (log₁₀). **C** PI3K/AKT pathway activity was assessed by calculating the Reactome PI3K/AKT signalling pathway score from RNAseq data in CP50c, CP253c, CP267c, CP336c, and CP142c patient-derived xenograft (PDX) models. Data represent six biological replicates. The box shows the interquartile range, the line

indicates the median, and the whiskers represent the minimum and maximum values. Statistical analysis was performed using one-way ANOVA with Tukey's posthoc test. **D** Protein levels of phosphoBAD (Ser136; pBAD), total BAD, BIM, and BAK were detected by western blot, with GAPDH as a loading control, in the PDX models. Caspase 3/7 activity at 6 h (**E**) and cell viability at 24 and 96 h (**F**) in response to vehicle (DMSO), ipatasertib (1 μM), AZD5991 (1 μM), S63845 (1 μM), and their combinations in CP50c, CP253c, CP267c, CP336c, and CP142c PDX-derived organoids (PDX-Os). Statistical analysis was performed using one-way ANOVA with Tukey's post-hoc test. The vehicle, AZD5991, and S63845 arms (represented by dotted-pattern bars) were previously shown in Fig. 2K (same experiment). Asterisks indicate statistical significance between groups (*$p < 0.05$; **$p < 0.01$; ***$p < 0.001$). **G** Representative microscopy images of the PDX-Os on day 4 after treatment. The scale bar indicates a length of 50 μm. All the experiments were performed in three biological replicates and the standard error of the mean is shown. Source data and exact $p$ values are provided as a Source Data file.

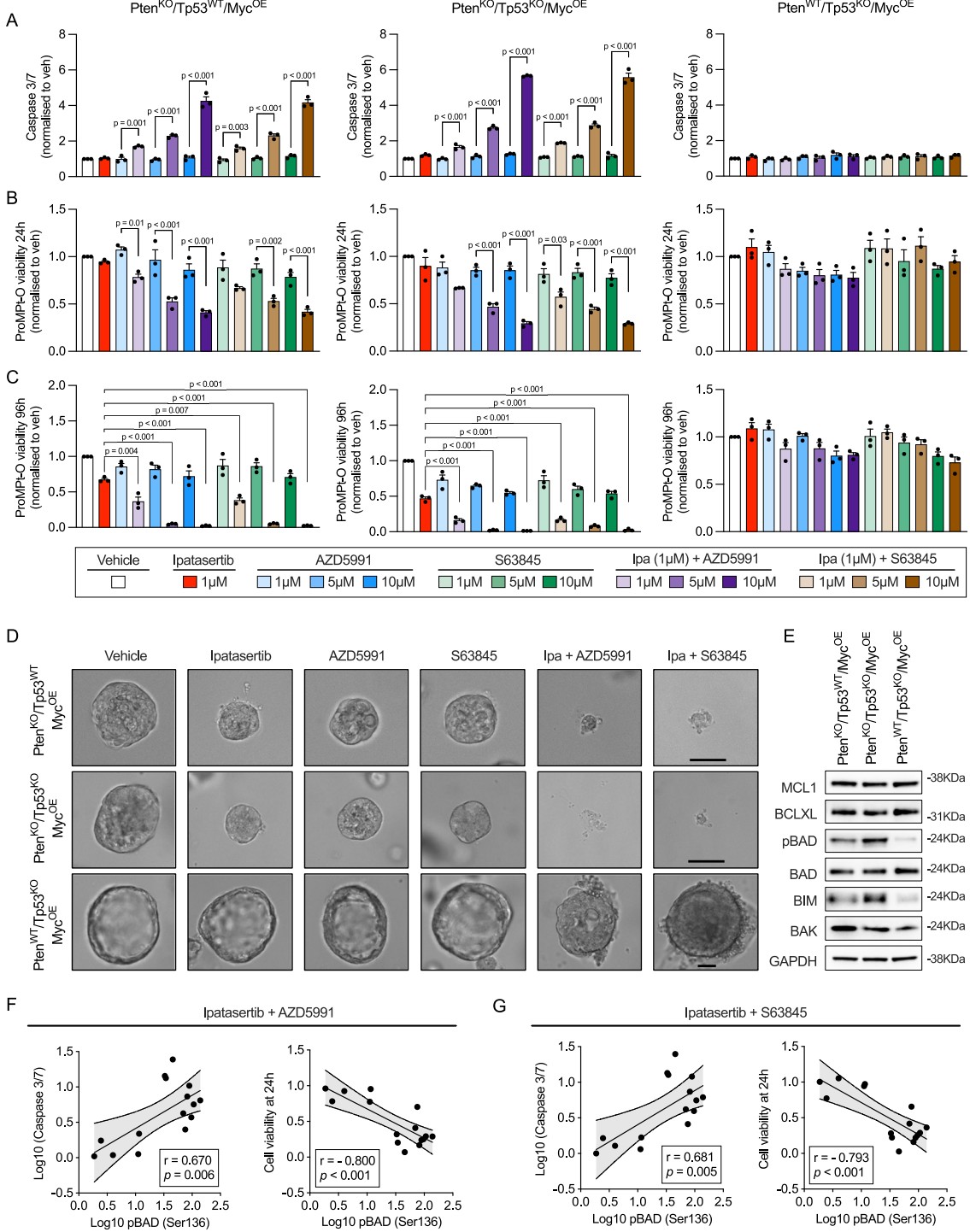

**Fig. 6 | AKT and MCL1 co-inhibition triggers apoptosis in PTEN-deficient mouse prostate cancer organoids.** Prostate Cancer Preclinical Mouse Modelling Platform organoids (ProMPt-Os) were treated with vehicle (DMSO), ipatasertib (1 μM), AZD5991 (1, 5, and 10 μM), S63845 (1, 5, and 10 μM), and their combinations to assess caspase 3/7 activity at 6 h (**A**), cell viability at 24 h (**B**), and cell viability at 96 h (**C**). Caspase 3/7 activity and cell viability were assessed by Caspase-Glo 3/7 3D and CellTiter-Glo 3D, respectively. Data represent the mean ± SEM of three independent biological replicates, each performed with three technical replicates. Statistical analysis was performed using one-way ANOVA with Tukey's post-hoc test. **D** Representative microscopy images of the ProMPt-Os on day 4 after treatment. This experiment was repeated three times with similar results. The scale bar indicates a length of 200 μm. **E** Protein levels of MCL1, BCLXL, pBAD (Ser136), BAD, BIM, and BAK were analysed by western blot with GAPDH as the loading control in

ProMPt-Os. This experiment was performed as a single biological replicate. Spearman correlation between caspase 3/7 activity (left panels) or cell viability at 24 h (right panels) in response to ipatasertib + AZD5991 (**F**) or ipatasertib + S63845 (**G**) and basal pBAD (Ser136) levels in all cell lines, PDX-O, and ProMPt-O models. Simple linear regression is depicted, with the line representing the mean and the shaded area indicating the 95% CI. Caspase 3/7 activity is expressed as log10 fold change relative to vehicle, cell viability as fold change relative to vehicle, and pBAD levels as log10 of pBAD intensity normalised to GAPDH intensity. Spearman correlation coefficients and two-sided p-values are shown. Caspase 3/7 activity was measured using Caspase-Glo assays (2D for cell lines and 3D for PDX-Os/ProMPt-Os), and cell viability was measured using CellTiter-Glo assays (2D for cell lines and 3D for PDX-Os/ProMPt-Os). Source data are provided as a Source Data file.

while single-agent treatments showed no statistically significant effect, indicating a synergistic interaction between the two drugs in vivo (Fig. 7A–C). No mice exhibited weight loss exceeding 20% of their baseline or showed any signs of discomfort or distress Supplementary Fig. 20). However, this should be taken with caution as S63845 (and other MCL1 inhibitors) show less affinity for mouse MCL1 than human MCL1[29]. Histopathological examination of terminal tumours revealed a significant increase in cleaved caspase 3 in response to the combined treatment, and a decrease in KI67 with ipatasertib treatment, suggesting that the combined treatment inhibited the growth of CP253c by triggering apoptosis (Fig. 7D). Furthermore, downstream markers of the PI3K/AKT pathway (i.e., pGSK3B and pPRAS40) were downregulated in response to both ipatasertib single treatment and the combined treatment with S63845 (Fig. 7D). These data orthogonally validate our in vitro studies, confirming that AKT and MCL1 co-inhibition demonstrates marked anti-tumour activity in a PTEN loss patient-derived model in vivo, with associated pharmacodynamic biomarker modulation.

### CRPC cells with acquired resistance to capivasertib remain sensitive to AKT and MCL1 co-inhibition, unlike those with acquired resistance to MCL1 inhibitors

To mimic acquired resistance to AKT inhibition seen in mCRPC patients during clinical trials[23,24], we generated capivasertib-resistant 'Capi-R' C4-2 and LNCaP95 cell lines. We first validated the models, demonstrating a >10-fold higher capivasertib IC$_{50}$ in Capi-R compared with parental cells (Fig. 8A, B). Both C4-2 and LNCaP95 Capi-R cells exhibited a slower growth rate compared to their parental counterparts (Fig. 8A, B). The IC$_{50}$ of capivasertib significantly decreased in the presence of AZD5991 in both LNCaP95 and C4-2 Capi-R cells (Fig. 8C, D), suggesting sensitivity to this combined treatment. Moreover, a synergistic increase in caspase 3/7 activity was observed in response to the combination of capivasertib and AZD5991 in LNCaP95 and C4-2 Capi-R cells (Fig. 8E, F). Furthermore, western blot analysis of cleaved PARP and caspase 3 orthogonally validated apoptosis induction (Fig. 8G, H). Apoptosis induction was more pronounced in LNCaP95 Capi-R compared to C4-2 Capi-R cells (Fig. 8E, F, G, H). BAK, BAD, and BIM were downregulated in Capi-R cells, with a more pronounced decrease in C4-2 compared to LNCaP95 Capi-R cells (Fig. 8I, J and Supplementary Fig. 21A, B), which may contribute to their reduced sensitivity to the combined treatment. Noteworthy, phosphorylation levels of BAD (Ser136) decreased with capivasertib in both C4-2 and LNCaP95 Capi-R cells (Fig. 8K, L and Supplementary Fig. 21C, D).

We next generated AZD5991- and S63845-resistant LNCaP95 cells and assessed their response (Supplementary Fig. 22A–C). While AKT inhibition alone continued to reduce cell viability, the combination of AKT and MCL1 inhibitors no longer increased caspase 3/7 activity or synergistically decreased viability (Supplementary Fig. 22D–F). The MCL1 inhibitor-resistant cell lines exhibited lower levels of phospho-BAD, BIM, and BAK compared to parental LNCaP95 cells (Supplementary Fig. 22G).

These results suggest that co-inhibition of AKT and MCL1 induces apoptosis, retaining efficacy in PC cells with acquired resistance to AKT inhibitors (as long as AKT inhibition continues to dephosphorylate BAD), while this synergy is lost in cells with acquired resistance to MCL1 inhibitors.

### Targeting MCL1 through CDK9 inhibition recapitulates the phenotype of MCL1 inhibition in combination with AKT blockade in CRPC cells

CDK9 inhibition rapidly downregulates MCL1 protein levels in several tumour-types[30–32]. We thus hypothesised that CDK9 inhibitors could also trigger acute apoptosis when combined with AKT inhibitors, in CRPC cells with PI3K/AKT hyperactivation. First, fadraciclib significantly decreased MCL1 protein levels after 6-h of incubation in LNCaP95 and C4-2 cells (Fig. 9A, B). The co-inhibition of AKT (using capivasertib and ipatasertib) and CDK9 (using fadraciclib and AZD4573), synergistically induced apoptosis and reduced cell viability (24 h) (Fig. 9A, B and Supplementary Fig. 23–24). Consistent with the mechanism underlying the synergistic effects of AKT and MCL1 co-inhibition, the caspase 3/7 activity induction and cell viability reduction observed in response to the combination of fadraciclib with AKT inhibitors, were prevented when cells were pre-treated with a combination of BIM, BAD, and BAK siRNAs (Fig. 9C). Furthermore, the response profile to AKT inhibitors and fadraciclib combinations mirrored that of AKT and MCL1 co-inhibition, with the same cell lines, PDX-Os and ProMPt-Os exhibiting sensitivity to both strategies (Fig. 9D–F and Supplementary Fig. 25). In vivo analysis of CP253c PDX showed that the combination of ipatasertib with fadraciclib significantly reduced tumour growth, whereas single treatments showed no significant impact (Fig. 10A). No mice exhibited a weight loss exceeding 20% of their baseline or showed any signs of discomfort or distress (Supplementary Fig. 26). Subsequent examination of terminal tumours by IHC revealed that cleaved caspase 3 induction was observed only in response to the combined treatment, while both single treatments decreased KI67, suggesting that single treatments impacted cell growth, whereas combined treatment triggered apoptosis in vivo (Fig. 10D). Additionally, downstream markers of AKT activity (pGSK3B and pPRAS40) were decreased in response to both ipatasertib alone and when combined with fadraciclib, and MCL1 protein levels were decreased by fadraciclib (Fig. 10D). Finally, while AKT inhibitors combined with fadraciclib continued to synergistically induce apoptosis and cell death in C4-2 and LNCaP95 Capi-R cells, this effect was no longer observed in LNCaP95 cells with acquired resistance to MCL1 inhibitors (Supplementary Fig. 27–28).

These data demonstrate that CDK9-mediated MCL1 downregulation, combined with AKT inhibition, replicated the in vitro and in vivo findings of direct MCL1 inhibition, providing further opportunities to clinically translate this therapeutic strategy.

## Discussion

MCL1 supports cell survival by sequestering pro-apoptotic BCL2 proteins, preventing mitochondrial outer membrane permeabilization and the initiation of the intrinsic apoptosis pathway. Herein, we characterise MCL1 status at genomic, RNA and protein level in multiple independent mCRPC cohorts, including in matched same-patient CSPC and CRPC biopsies. We show that *MCL1* amplification is a truncal event occurring early in lethal PC evolution, whilst *MCL1* copy number gain occurs at diagnosis but also emerges with treatment resistance, probably representing an adaptive response. Interestingly, these genomic changes associate with worse clinical outcomes and inferior responses to neoadjuvant ADT and enzalutamide. Indeed, previous studies have shown MCL1 upregulation confers resistance to ADT in PC cells[9], therefore we suggest that the selection of subclones harbouring MCL1 copy number gains or amplifications is a potential mechanism contributing to tumour progression and therapy resistance. However, no association was observed between MCL1 gain/amplification and taxane response, which could potentially be explained by the ability of taxanes to induce MCL1-independent mitotic catastrophe in these tumours[33,34]. These data support the urgent development of molecularly stratified therapeutic strategies in MCL1 overexpressing mCRPC to improve the outcome for patients with this lethal disease.

We show that CRPC models with MCL1 copy number gain respond to MCL1-targeting BH3 mimetics, likely due to their increase reliance on MCL1 for survival, with inhibition inducing apoptosis, as previously observed in other tumour-types[7,20,21]. A limitation of our study is the lack of in vivo validation testing the anti-tumour effects of MCL1 inhibition in models with MCL1 copy number gain. Despite this, we believe that our data support single-agent MCL1 inhibition as an

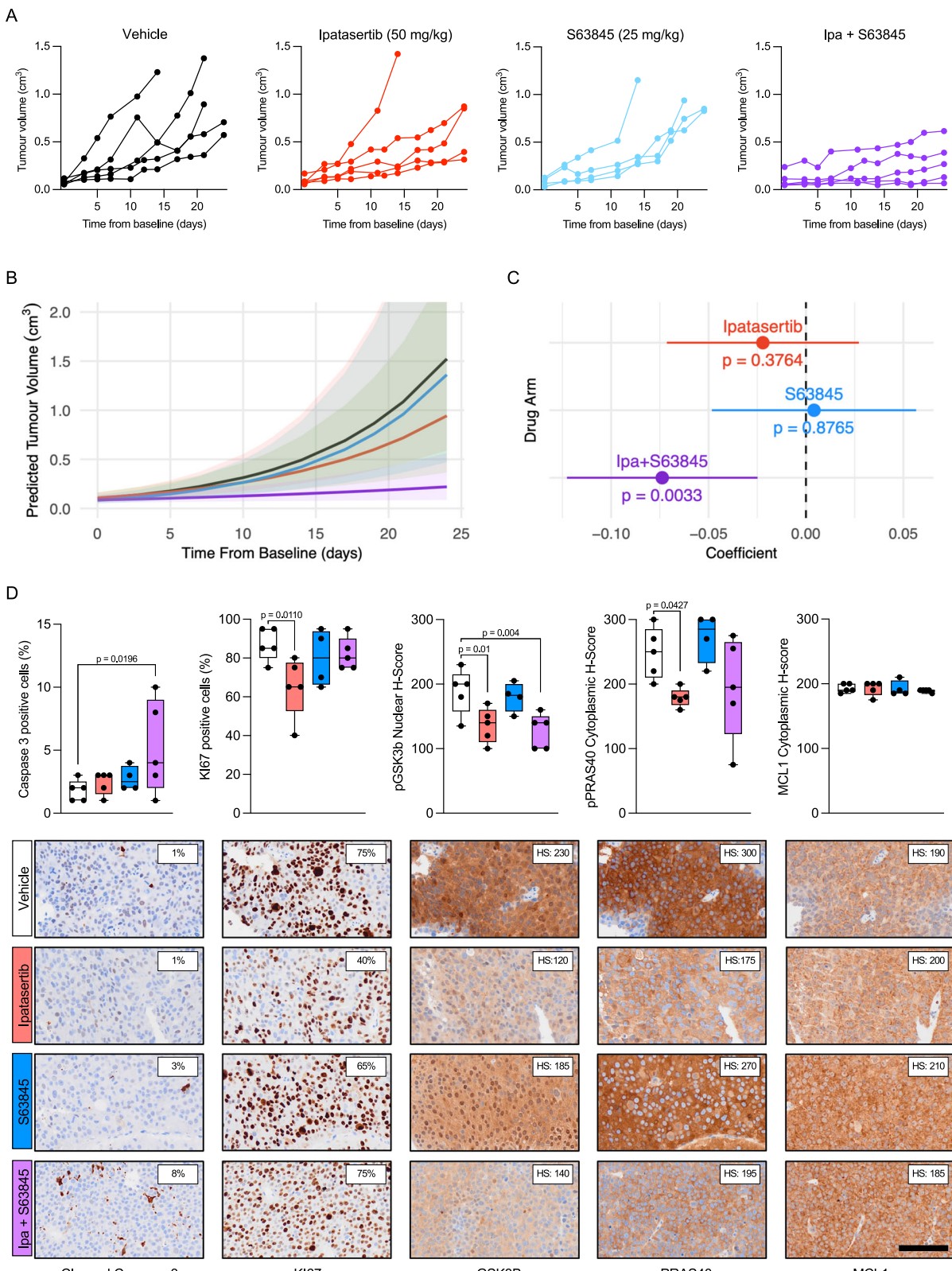

effective anti-cancer strategy for CRPC with *MCL1* copy number gains in a molecularly stratified approach. Previous studies have reported limited anti-tumour activity for single agent MCL1 inhibitors in PC models, but this is likely due to the selection of models with unaltered *MCL1* copy number status as well as redundancy with other BCL2 family members including BCLXL[15,16]. Therefore, the major challenge for drug development in this space, is to identify therapeutic strategies that can inhibit both MCL1 and BCLXL to drive cancer cell specific cell death, whilst maintaining tolerability.

Taking this important point into consideration, and to broaden the effect of MCL1 inhibition in CRPC without MCL1 copy number gains, we explored the efficacy of MCL1 inhibition in combination with FDA-approved oncological drugs with a view to discover synergistic combinations that could be rapidly translated to the

**Fig. 7 | The combination of ipatasertib and S63845 has anti-tumour activity in the CRPC PDX CP253c in vivo. A** Individual tumour volumes of CP253c treated with ipatasertib (50 mg/kg; *n* = 5), S63845 (25 mg/kg; *n* = 4), combined treatment (*n* = 5) and vehicle (*n* = 5). **B** Predicted tumour growth using a linear mixed-effect model of CP253c treated with ipatasertib (50 mg/kg; *n* = 5), S63845 (25 mg/kg; *n* = 4), combined treatment (*n* = 5) and vehicle (*n* = 5). **C** Forest plots showing the results from the longitudinal mixed effect model for log-transformed tumour volume. Estimates and *p*-values refer to the interaction term of treatment arm and time indicating the difference in tumour volume growth rate between each treatment arm and the vehicle arm. Points represent the random effect coefficients for each drug group and error bars the 95% confidence intervals. The black dotted line indicates the vehicle (reference level). **D** Comparison of protein expression for cleaved caspase 3 (% positive cells), Ki67 (% positive cells) and pGSK3B, pPRAS40 and MCL1 H-score for the different treatment arms. The box shows the interquartile range, the line indicates the mean, and the whiskers represent the minimum and maximum values. One-way ANOVA with post-hoc Fisher's LSD test was performed. Representative IHC micrographs for end-of-treatment CP253c tumours are also shown (bottom panel). The scale bar indicates a length of 100 μm. Source data are provided as a Source Data file.

clinic. This led to the discovery that MCL1 inhibition synergises with AKT blockade to induce BAK-dependent apoptosis by modulating BAD-BCLXL and BIM-MCL1 interactions. This therapeutic combination has been investigated in other tumour types, including myeloma[35], leukaemia[36], and breast cancer[37]. However, our study highlights its therapeutic potential in CRPC and offers significant advancements in understanding the underlying molecular mechanisms. Specifically, our study demonstrates that AKT inhibition dephosphorylates, and subsequently activates the BH3-only protein BAD, as previously reported[38]. This activation leads to increased binding of BAD to BCLXL, inhibiting its anti-apoptotic function, and therefore triggering apoptosis when combined with MCL1 inhibition[39]. We showed that AKT inhibition also increases BIM-MCL1 binding, likely in response to the displacement of BIM from BCLXL due to the competitive binding of BAD. This might contribute to increasing dependency on MCL1 in the context of AKT inhibition (and subsequent BCLXL blockade), following the direct activation model[40]. Altogether, these mechanistic data are consistent with previous findings showing that PC models are co-dependent on MCL1 and BCLXL, with dual blockade needed to induce apoptosis[16].

One potential challenge linked to MCL1 and BCLXL dual inhibition is the anticipated clinical toxicity[41,42]. Critically, this is less likely to be the case for co-inhibition of MCL1 and AKT, as we observed cancer cell killing only in models that harbour PI3K/AKT pathway activating genomic aberrations (such as *PTEN* loss and *AKT1* amplification), which is likely attributed to their significant dependency on phosphorylated BAD for survival[38,39,43]. Indeed, our data suggest that high basal levels of phosphorylated BAD (Ser136) may be a more sensitive predictive biomarker for this combination therapy than genomic PI3K/AKT pathway aberrations (e.g., PTEN loss). Although single agent MCL1 demonstrated promising activity in PC with *MCL1* copy number gains, this remains a small subset when compared with activation of the PI3K/AKT pathway, which accounts for around 50% of mCRPC, and is associated with worse clinical outcome[44,45]. Taken together, these data demonstrate that co-inhibition of MCL1 and AKT deliver tumour specific killing in PTEN-loss/PI3K-activated lethal PC with high levels of phospho-BAD, providing a molecularly driven therapeutic strategy for a common subtype of lethal PC with poor prognosis[45].

AKT inhibitors have shown promise in clinical trials for patients with mCRPC, but have not yet met the threshold for regulatory approval[23,24]. As such, the development of drug combination strategies to enhance anti-tumour activity or reverse resistance to AKT inhibition is an unmet clinical need. In this space, it should be noted that the co-inhibition of MCL1 and AKT not only drove apoptotic cell death in PC cells sensitive to AKT inhibition but also exerted anti-cancer activity in PC models that had acquired resistance to single agent AKT blockade. This is consistent with the reported sensitivity of AKT inhibition resistant breast cancer cell lines (developed by CRISPR-Cas9 KO techniques, as opposed to acquired resistance) to MCL1 blockade[37]. Furthermore, we show that dual blockade of AKT and MCL1 is effective in mCRPC models characterised by biomarkers associated with resistance to AR signalling inhibitors and PC progression, including AR negativity, AR-V7 expression and *TMPRSS2:ERG* fusion[46–52]. In conclusion, AKT and MCL1 co-inhibition demonstrates efficacy in overcoming resistance to AKT blockade and provides a targeted therapeutic strategy for common mCRPC lethal subtypes with potential for significant clinical impact. Importantly, we observed that cells with acquired resistance to MCL1 inhibitors no longer responded to dual AKT/MCL1 inhibition. This suggests that combining AKT and MCL1 blockade upfront, rather than after progression on MCL1 inhibitors, may be a more effective treatment strategy.

CDK9 inhibitors rapidly downregulate MCL1 protein levels and have shown anti-tumour activity with manageable toxicity in early-phase clinical trials for haematological malignancies[30–32,53–55]. Here, we demonstrate that co-inhibition of CDK9 and AKT mirrors the effects of MCL1 and AKT co-inhibition, inducing cell death in PTEN-loss/PI3K-activated PC cells with high phospho-BAD levels. While CDK9 is a broad transcriptional regulator affecting multiple genes beyond MCL1, some studies suggest that MCL1 depletion is a key driver of the apoptosis-related anti-tumour effects of CDK9 inhibitors[30,56,57]. Our data support that CDK9-dependent acute MCL1 downregulation contributes, at least in part, to the synergy observed with CDK9 and AKT co-inhibition. Specifically, the synergistic effects observed with both CDK9/AKT and MCL1/AKT co-inhibition strategies require the same BH3-only proteins (BIM, BAD, and BAK), and cells with acquired resistance to MCL1 inhibitors exhibit resistance to both combinations. However, we acknowledge that additional factors acutely regulated by CDK9 inhibition may also contribute to this synergy. Despite this, our findings support CDK9 inhibition combined with AKT blockade as an alternative, promising therapeutic strategy for PTEN-loss/PI3K-activated PC. It is important to acknowledge the challenges associated with direct MCL1 inhibition, particularly due to on-target cardiotoxicity[58,59]. In this context, future strategies, including antibody-drug conjugates and proteolysis-targeting chimeras (PROTACs), similar to those developed for other BCL2 family members, could enable PC-specific MCL1 targeting while minimising the risk of cardiotoxicity[60,61].

Critically, our data support the urgent evaluation of therapeutic strategies targeting MCL1, directly or indirectly, as a single agent for PC harbouring *MCL1* copy number gains, and in combination with AKT blockade in PTEN-loss/PI3K-activated lethal PC, within 'proof of concept, proof of mechanism' early phase clinical trials.

## Methods
### Patients and sample collection
**Royal Marsden Hospital cohorts.** All patients had mCRPC, were treated at the RMH, provided written informed consent, and were enroled in protocols approved by the RMH ethics review committee. For the matched same-patient cohort, treatment-naïve diagnostic samples were retrieved from referring hospitals. CRPC tissue was collected from mCRPC patients. Clinical data and demographics were collected retrospectively from the RMH electronic patient record system. Clinical characteristics and sample disposition are as previously described [RMH matched same-patient cohort[62], RMH CRPC cohort[63]]. For the RMH matched same-patient cohort[62], 44 patients were included with copy number analysis data by low pass whole

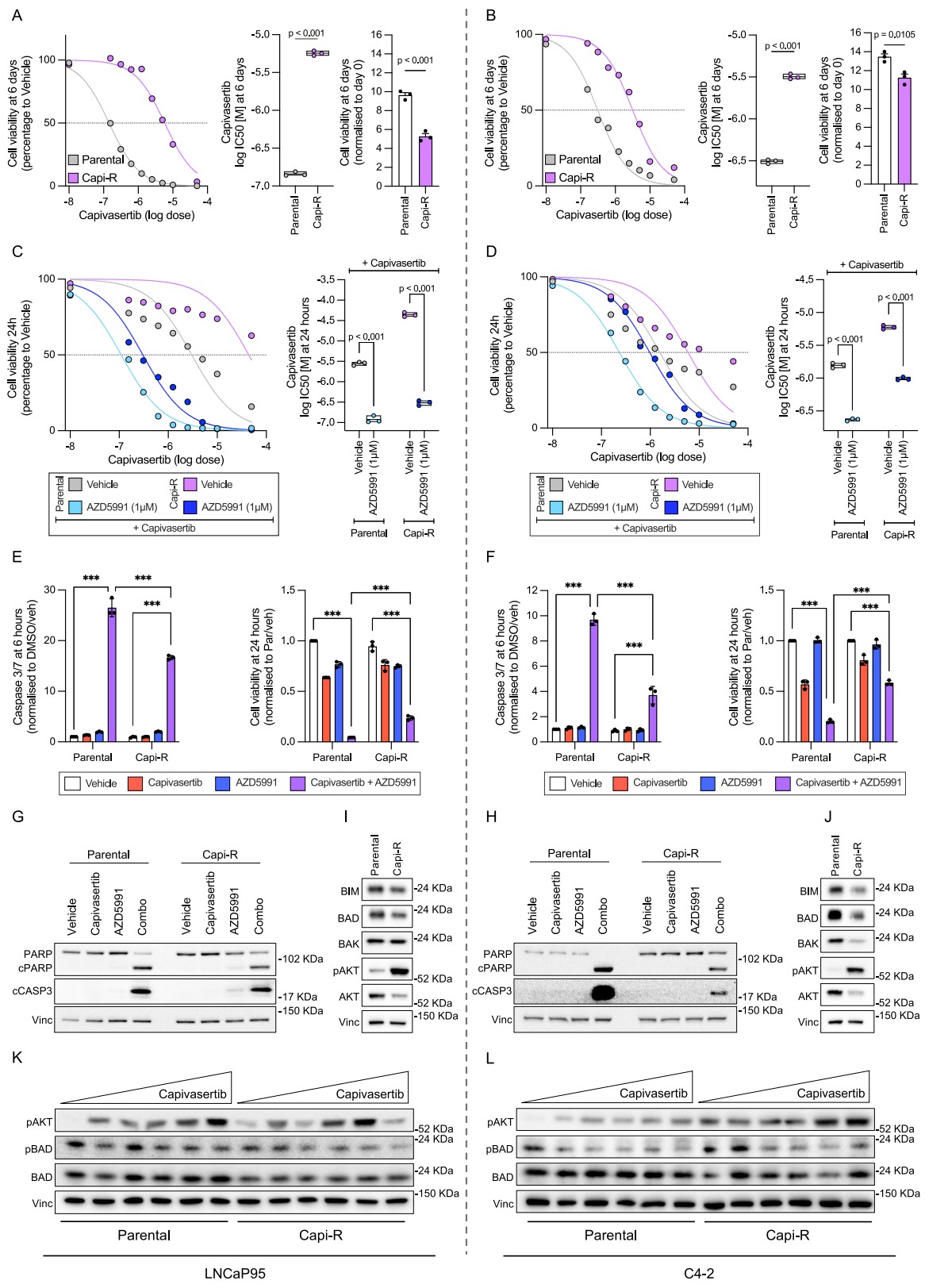

genome sequencing of tumour biopsy DNA. For the RMH CRPC cohort, 86/100 patients had RNA-sequencing data available and were included for this study.

**FIRSTANA and PROSELICA cohort.** All patients had mCRPC and participated in the FIRSTANA or PROSELICA prospective, randomised, open-label, international phase III trials. In FIRSTANA, patients were randomised to receive cabazitaxel (20 mg or 25 mg/m²) or docetaxel

(75 mg/m²), all with prednisone (10 mg/day) as first-line chemotherapy for mCRPC[64]. In PROSELICA, patients were randomised to receive either cabazitaxel 20 mg/m² or cabazitaxel 25 mg/m², each with prednisone 10 mg/day as second-line chemotherapy after docetaxel. All patients included in our study consented to allow for the collection of plasma. Plasma cell-free DNA low-pass whole-genome sequencing data were obtained from baseline samples. Samples with a tumour fraction >5% were assessed for genomic copy number burden (FIRSTANA

**Fig. 8 | CRPC cells with acquired resistance to capivasertib remain sensitive to AKT and MCL1 co-inhibition. A, B** Comparison of capivasertib dose-response curves (left panel), capivasertib IC50 (central panel) and basal cell growth (fold change to day 0; right panel) between Capi-R and parental LNCaP95 (**A**) and C4-2 (**B**) cells. A two-sided unpaired t-test assuming normal distribution was performed. Comparison of capivasertib dose-response curves (left panel) and capivasertib IC50 (right panel) in presence and absence of AZD5991 (1 μM, 24 h) between Capi-R and parental cells in LNCaP95 (**C**) and C4-2 (**D**). Floating bars (min to max) with line at mean are depicted. One-way ANOVA with Tukey's post-hoc test was performed. Caspase 3/7 activity at 6 h (left panel) and cell viability at 24 h (right panel) in parental and Capi-R LNCaP95 (**E**) and C4-2 (**F**) cells treated with vehicle (DMSO), capivasertib (1 μM), AZD5991 (1 μM) and combined treatment. Two-way ANOVA with Tukey's post-hoc test was performed. Cell viability and caspase 3/7 activity were determined using Caspase-Glo 3/7 2D assay and CellTiter-Glo 2D, respectively. Protein expression of cleaved PARP and cleaved caspase 3 (at 6 h; determined by western blot) in LNCaP95 (**G**) and C4-2 (**H**) treated with vehicle (DMSO), capivasertib (1 μM), AZD5991 (1 μM) and combined treatment (combo). Comparison of protein levels of BIM, BAD, BAK, total and p-AKT$^{Ser473}$ between parental and Capi-R cells in LNCaP95 (**I**) and C4-2 (**J**). Protein levels of total and p-BAD$^{Ser136}$, total BAD, and p-AKT$^{Ser473}$ in response to vehicle (DMSO) and varying capivasertib concentrations (0.1 μM 0.5 μM, 1 μM, 5 μM) between parental and Capi-R cells in LNCaP95 (**K**) and C4-2 (**L**). Vinculin was used as a housekeeping protein. All the experiments were performed in three biological triplicates (apart from the western Blot shown in (**H** and **J**); $n = 1$) and technical singlets. The standard error of the mean is shown. Asterisks (*$p < 0.05$; **$p < 0.01$; ***$p < 0.001$) indicate statistically significant differences between groups. Source data and exact p values are provided as a Source Data file.

$n = 79$, PROSELICA $n = 73$). OS was defined as the time from randomisation to death of any cause. Clinical characteristics are presented in Supplementary Table 1.

**Stand Up To Cancer/Prostate Cancer Foundation cohort.** Copy number variation ($n = 336$) and transcriptome ($n = 159$) data from mCRPC biopsies, generated by the International SU2C/PCF, were downloaded and re-analysed[45,65].

**The Cancer Genome Atlas cohort.** Clinical data and associated genomic data ($n = 494$) were downloaded from the TCGA-Clinical Data Resource (CDR) and Progression Free Interval (PFI) was used to evaluate outcome[66]. PFI was defined as the time from the date of diagnosis until the date of the first occurrence of a new tumour event, which includes progression of the disease, locoregional recurrence, distant metastasis, new primary tumour, or death with tumour.

**National Cancer Institute neoadjuvant cohort.** For the NCI neoadjuvant cohort, tissue collection and analysis from patients with intermediate- and high-risk locally advanced PC treated with neoadjuvant ADT and enzalutamide prior to radical prostatectomy. This trial was approved by the Institutional Review Board of the Center for Cancer Research, NCI, Bethesda, MD (ClinicalTrials.gov identifier: NCT02430480).

**University of Washington/Fred Hutchinson (UW/FH) cohort.** For UW/FH cohort, tissue was obtained from patients who had died of mCRPC and had signed written informed consent for a rapid autopsy performed within 6 h of death, under the aegis of the PC Donor Programme at the University of Washington. All procedures involving human subjects in the rapid autopsy programme were approved by the Institutional Review Board of the University of Washington and of the Fred Hutchinson Cancer Center.

## Bioinformatic analyses

**Copy number analysis.** Copy number variations data were processed and analysed as previously described in biopsies from RMH[62], SU2C/PCF[45,65]. In short, exome-sequencing reads were aligned to GRCh37 using BWA MEM. GATK realignment and recalibrate base scores were applied. Germline variance and somatic variance were called using GATK and Mutect2 by comparing tumour-normal paired comparison. ASCAT were used for copy number analysis. TCGA CNA segment data was download from cBioportal ($n = 489$). Copy number analysis for FIRSTANA/PROSELICA[64] cohorts were performed using low pass whole genome sequencing from plasma DNA. Sequencing reads were aligned to GRCh37 using BWA MEM. Aligned reads were quantified by HMMcopy readCounter, and copy number and the tumour fraction were calculated using ichorCNA. For the National Cancer Institute neoadjuvant cohort, exome sequencing of tumour foci and processing of data to derive gene-level somatic copy number estimates was described previously[67].

**Transcriptome analysis.** TCGA transcriptomic data were downloaded from cBioPortal[66,68]. RMH and SU2C/PCF RNA sequencing dataset analysis (TopHat2 pipeline): CRPC transcriptomes generated by the SU2C/PCF Dream Team, were downloaded and reanalysed[45]. Paired-end transcriptome sequencing reads for each of the SU2C/PCF ($n = 106$) and RMH ($n = 86$) cohorts, comprising samples with matched RNA and copy number data, were aligned to the human reference genome (GRCh37/hg19) using Tophat2 (v2.0.7)[63]. Gene expression, Fragments Per Kilobase of transcript per Million mapped reads (FPKM), was calculated using Cufflinks. The top expressed genes ($n = 15000$) were analysed for each cohort respectively. The Spearman's rank correlation coefficient (Spearman's rho) between each gene's expression (FPKM) and *MCL1* expression (FPKM) was calculated, and subsequently used for pathway analysis. Pathway analysis was performed using the Gene Set Enrichment Analysis (GSEA) Pre-Ranked algorithm from GSEA software (v4.1.0). GSEA Pre-Ranked results were obtained using the H collection of Hallmark gene sets (MSigDB v7.0), with default parameters. UW/FH mCRPC RNA sequencing: RNA isolation and sequencing of mCRPC tumours were performed as described previously. Sequencing reads were mapped to the hg38 human genome using STAR.v2.7.3a. All subsequent analyses were performed in R. Gene level abundance was quantitated using GenomicAlignments and transformed to log2 FPKM using edgeR. Boxplots by patient were filtered to include only patients with at least two tumours profiled and matched copy number data ($n = 115$ tumours from 50 patients) and are ordered by per-patient median log2 FPKM gene expression. PI3K/AKT pathway score was calculated by accumulated z-score using genes of the REACTOME_PI3K_AKT_SIGNA-LING_IN_CANCER pathway obtained from the molecular signature database (MSigDB)[69].

## Immunohistochemistry

IHC was performed on freshly sectioned tissue blocks where adequate material was present (≥100 tumour cells, reviewed by histopathologist B.G). BIM, BAD, and BAK antibodies were validated for specificity and optimised for IHC (Supplementary Figs. 1 and 2), while MCL1 antibody validation was reported previously[70]. Details on the IHC assays are available in Supplementary Table 2. Protein quantification was determined for by a histopathologist (B.G) blinded to clinical data using H score method; [(% of absent staining x 0) + (% of weak staining x 1) + (% of moderate staining x 2) + (% of strong staining x 3)], resulting in a score between 0 and 300[71]. In addition, a pathologist-supervised machine learning algorithm (HALO AI, Indica Labs) was trained to calculate cytoplasmic MCL1 protein levels in patient samples. First, PC cells were differentiated from benign stroma. Colour deconvolution for DAB and haematoxylin staining was performed, and cell recognition and nuclear segmentation optimised. The analysis algorithm was then adjusted to provide continuous data on the intensity of MCL1 cytoplasmic signal intensity normalised per cell (accounting for cell size), averaged across the number of tumour cells identified per

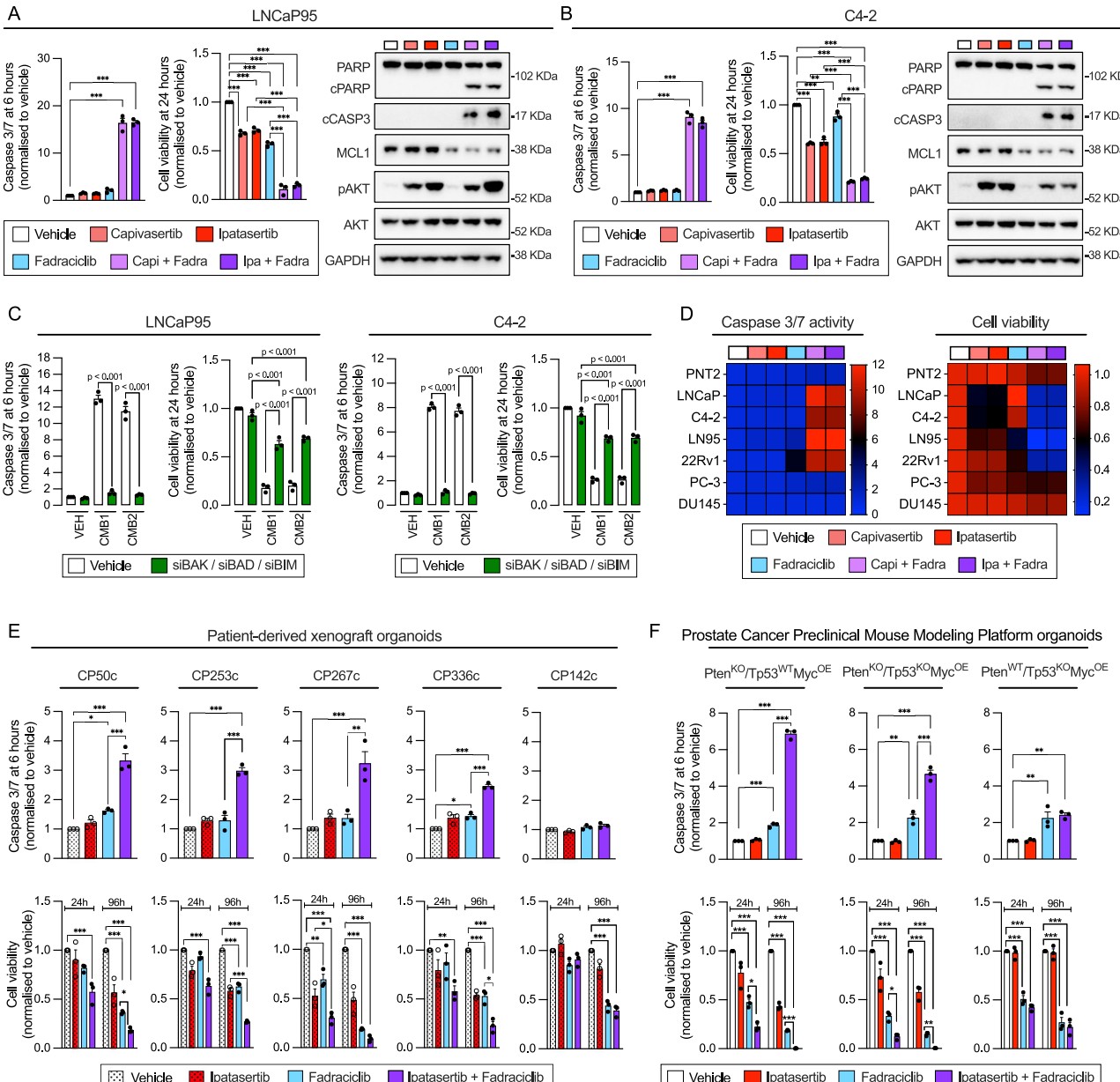

**Fig. 9 | AKT and CDK9 co-inhibition recapitulates the effects of AKT and MCL1 co-inhibition in vitro.** Caspase 3/7 activity (6 h), cell viability (24 h), and protein expression (cleaved PARP, cleaved caspase 3, MCL1, total and phospho-AKT Ser473; 6 h) were assessed in LNCaP95 (**A**) and C4-2 (**B**) cells treated with vehicle (DMSO), capivasertib (1 μM), ipatasertib (1 μM), fadraciclib (1 μM), and their combinations. GAPDH was used as a loading control. **C** Caspase 3/7 activity (6 h), cell viability (24 h), and protein expression (6 h) were evaluated in LNCaP95 (left) and C4-2 (right) cells following treatment with either capivasertib + AZD5991 or ipatasertib + fadraciclib, in the context of triple knockdown of BAK, BAD, and BIM (OnTarget siRNAs, 25 nM each for 72 h) or control siRNA (75 nM, 72 h). **D** Heatmaps show the effects of capivasertib, ipatasertib, fadraciclib, and their combinations on caspase 3/7 activity (6 h) and cell viability (24 h) in PNT2, LNCaP, C4-2, LNCaP95, 22Rv1, PC-3, and DU145 cells, shown as fold change relative to vehicle. **E** Caspase 3/7 activity (6 h) and organoid viability (24 h and 96 h) were assessed in CP50c, CP253c,

CP267c, CP336c, and CP142c PDX-derived organoids (PDX-Os) treated with vehicle, ipatasertib (1 μM), fadraciclib (1 μM), or the combination. Vehicle and ipatasertib data (dotted bars/symbols) are from the same experiment shown in Fig. 4D. **F** Caspase 3/7 activity (6 h) and cell viability (24 h and 96 h) were measured in Prostate Cancer Preclinical Mouse Modelling Platform organoids (ProMPt-Os) treated with vehicle, ipatasertib (1 μM), fadraciclib (1 μM), or the combination. All data represent three biological replicates with three (cell lines, ProMPt-Os) or five (PDX-Os) technical replicates. Bars show mean ± SEM. Statistical analysis was performed using one-way ANOVA with Tukey's post-hoc test. Caspase 3/7 activity and viability were assessed using Caspase-Glo 3/7 and CellTiter-Glo in 2D (cell lines) or 3D (PDX-Os, ProMPt-Os) formats. Asterisks indicate statistical significance between groups (*p < 0.05; **p < 0.01; ***p < 0.001). Source data and exact p values are provided as a Source Data file.

sample, providing a value between 0 (no staining) and 1 (black) called optical density (OD).

## Fluorescence in situ hybridisation
Fluorescence In Situ Hybridisation (FISH) was performed in cell lines and patient-derived xenograft-organoids (PDX-Os) as previously

described[72]. Briefly, fresh cell lines and organoids were resuspended in 10 mL of phosphate-buffered saline and treated with a fixative solution (FS) containing a 3:1 mixture of methanol and acetic acid (600 μL). The suspension was vigorously vortexed and subsequently centrifuged at 500 × g for 5-min. The supernatant was then carefully decanted, and the cell pellet was resuspended in 15 mL of FS.

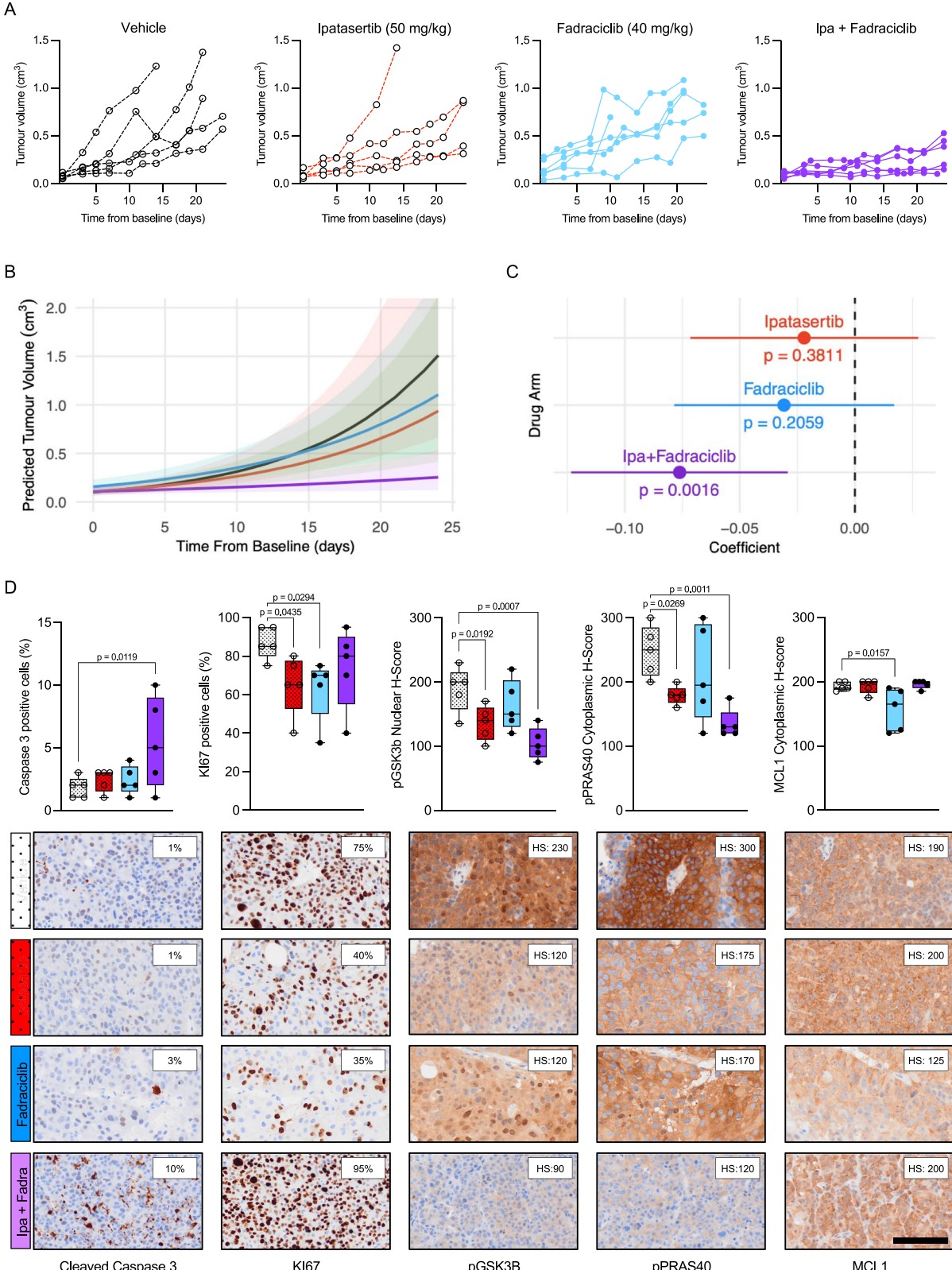

Following another round of centrifugation and removal of the supernatant, this process was repeated to ensure thorough fixation. A final 2 mL of FS was added to resuspend the cells. To facilitate optimal cell spreading for subsequent analysis, cells were released from a height of 20 cm onto positively charged slides. A dual-colour FISH assay was optimised using a cocktail of two FISH probes commercially available (ZytoLight SPEC *MCL1*/1p12 Dual Colour Probe, Z-2173-200, Bio SB). The *MCL1*/1p12 ratio ≥2.0 as previously reported[21].

## Drug compounds

AZD4573 and CYC065 (fadraciclib) were kind gifts from AstraZeneca and Cyclacel respectively. AZD5363 (capivasertib) and AZD5991 were (in part) a kind gift from AstraZeneca, with further stocks purchased from MedChemExpress. GDC-0068 (ipatasertib), ABT-263 (navitoclax), S63845, copanlisib and buparlisib were purchased from MedChemExpress. AZD5991 and S63845 specificity and selectivity are reported elsewhere[26,29]. All drugs were prepared as 10 mM stock

**Fig. 10 | AKT and CDK9 co-inhibition recapitulates the effects of AKT and MCL1 co-inhibition in vivo. A** Individual tumour volumes of CP253c treated with ipatasertib (50 mg/kg; $n = 5$), fadraciclib (40 mg/kg; $n = 6$), combined treatment ($n = 6$) and vehicle ($n = 5$). **B** Predicted tumour growth using a linear mixed-effect model of CP253c treated with ipatasertib (50 mg/kg; $n = 5$), fadraciclib (40 mg/kg; $n = 5$), combined treatment ($n = 5$) and vehicle ($n = 5$). **C** Forest plots showing the results from the longitudinal mixed effect model for log-transformed tumour volume. Estimates and p-values refer to the interaction term of treatment arm and time indicating the difference in tumour volume growth rate between each treatment arm and the vehicle arm. Points represent the random effect coefficients for each drug group and error bars the 95% confidence intervals. The black dotted line indicates the vehicle (reference level). **D** Comparison of protein expression for cleaved caspase 3 (% positive cells), Ki67 (% positive cells) and pGSK3B, pPRAS40 and MCL1 H-score for the different treatment arms. Necrotic samples were not analysed. The box shows the interquartile range, the line indicates the mean, and the whiskers represent the minimum and maximum values. One-way ANOVA with post-hoc Fisher's LSD test was performed. Representative IHC micrograph for end-of-treatment CP253c tumours are also shown (bottom panel). The scale bar indicates a length of 100 μm. Source data are provided as a Source Data file.

solution in DMSO and subsequently diluted in the appropriate media to achieve a final DMSO assay concentration of ≤0.1%.

## Cell lines
PC cell lines (LNCaP, C4-2, 22Rv1, PC-3, and DU145) were purchased from American Type Culture Collection (ATCC) and normal-like prostate cell line PNT2 from Sigma-Aldrich. LNCaP95 cells were generously provided by Drs Alan K. Meeker and Jun Luo (Johns Hopkins University, Baltimore, MD). The cell culture conditions are depicted in Supplementary Table 3. Cell lines underwent regular Short Tandem Repeat (STR) profiling and mycoplasma testing.

## Patient-derived xenograft (PDX) organoids (PDX-Os)
PDX-Os were generated from PDXs derived from human CRPC biopsies using methods described previously[27,73,74]. The patient treatment histories are detailed in Supplementary Fig. 3. Organoids were formed in Matrigel® matrix (356231, Corning) domes in 24-well plates and then re-seeded into 96-well plates for drug experiments.

## Prostate Cancer Preclinical Mouse Modeling Platform (ProMPt) organoids (ProMPt-Os)
The mouse organoids utilised in this study are part of the ProMPt. The mouse strains include PtenLoxP[75]; p53LoxP (JAX stock: #008462)[76]; MYCStopFL (JAX stock #020458)[77]; and the UBC-Cre-ERT2 (JAX stock #007001)[78]. Organoids were derived from mouse prostate tissue as previously described[79]. Briefly, mouse prostate organoids were established from prostates isolated from 3-month-old mice. Luminal and basal cells from mouse prostate tissue were separated by double antibody staining for CD24 and CD49f markers. Sorted luminal cells were plated in 30 μl Matrigel drops on 6-well plates and covered with complete mouse prostate organoid medium (1x B27, 1.25 mM N-acetyl-l-cysteine, 50 ng/ml EGF, 200 nM A83-01, 100 ng/ml Noggin, 500 ng/ml R-spondin, 1 nM dihydrotestosterone, 10 μM Y-27632 dihydrochloride in adDMEM/F12 containing penicillin/streptomycin, 10 mM HEPES and 2 mM GlutaMAX). To induce Cre-ERT2-mediated excision of all LoxP exons and STOP cassettes, organoids were treated for two passages with 4-hydroxytamoxifen. Successful CRE-mediated excision was confirmed using quantitative PCR and western blot analysis.

## Viability and apoptosis assays
PC cells, PDX-Os and ProMPt-Os were treated two days after seeding, or three days after small interfering RNA (siRNA) transfection, with BH3 mimetics, CDK9 inhibitors, or vehicle. PDX-Os and ProMPt-Os were cultured in full media without Y-27632 (ROCK inhibitor) for 24 h prior to treatment and maintained in the same media throughout the experiment. Cell line viability was evaluated 24 h post-treatment, and PDX-Os/ProMPt-Os viability at both 24 and 96 h after treatment using CellTiter-Glo 2D (G9241; Promega) and CellTiter-Glo3D (G9681; Promega), respectively. The in vitro caspase activity was evaluated 6 h post-treatment using the 2D Caspase-Glo 3/7 Luminescent assay (G8093; Promega) for cell lines and the 3D Caspase-Glo 3/7 Luminescent assay (G8983; Promega) for PDX-Os/ProMPt-Os, in accordance with the manufacturer's instructions.

## Food and drug administration-approved drug screen
The National Cancer Institute/National Institutes of Health (NCI/NIH) Food and Drug Administration (FDA)-approved anti-cancer Oncology Set was obtained from the NCI/NIH DTP Open Chemical Repository. The screen included 166 different compounds, delivered in DMSO at 10 mmol/L (Supplementary Table 4). C4-2, LNCaP95 and PNT2 cells were seeded in 384 well plates at 2000 cells per well. 48 h after seeding, cells were treated with the FDA approved compounds at 5 μM in absence and presence of AZD5991 at 1 μM with a final DMSO concentration in the wells ≤0.01%. Cell viability and apoptosis induction were assessed 24 h and 6 h post-treatment, using CellTiter-Glo 2D and Caspase-Glo 3/7 2D, respectively. The screen was performed as a single biological experiment ($n = 1$) without technical replicates.

## High-throughput combination matrix drug screening
For pairwise drug-combination assessments in matrix format, compounds were acoustically dispensed into a 1536-well white solid bottom tissue culture-treated plate (EWB041000A, Aurora Microplates, Whitefish, MT, USA) with an Echo 550 acoustic liquid handler (Labcyte, San Jose, CA, USA) using previously described methods[80,81]. A 9-point custom concentration range with 1:2 dilution between points was used for each drug-pair tested. Bortezomib (final concentration 20.3 μM) was used as a positive control for cell cytotoxicity. Cells were seeded into compound-containing plates at a density of 500 cells/well, in a final volume 5 μL of growth media by using a Multidrop Combi dispenser (Thermo Fisher Scientific). Plates were covered by a stainless-steel gasketed lid to prevent evaporation and incubated for 48 h in a humidified CO2 incubator. At the 48 h time point, 3 μL of Cell Titer Glo (Promega) was added to each well using a BioRAPTR® (Beckton Coulter, Brea, CA, USA) and plates were incubated at room temperature for 15 min with the stainless-steel lid in place. Luminescence readings were taken using a Viewlux reader (PerkinElmer) with a 2 s exposure time per plate. Viability of compound treated wells was normalised to DMSO and empty well controls present on each plate, and combination-response plotting was automatically performed for each individual drug+drug combination.

## siRNA transfection
Cell lines were transfected with ON-TARGETplus siRNA (SMARTpool, mixture of 4 siRNAs) (Dharmacon, Horizon) using 0.2% Lipofectamine RNAiMAX Reagent (Thermo Fisher Scientific) as per manufacturer's guidelines. For the pro-apoptotic BCL2 family siRNA screen, 3000–5000 cells were seeded in 96-well plates and treated with AZD5991 (1 μM) after 72 h of siRNA exposure. Cell viability (assessed with CellTiter-Glo 2D) and caspase activity (assessed with 2D Caspase-Glo 3/7) were determined 6- and 24 h post-treatment, respectively. Concentration of the siRNAs ranged from 25–75 nM and is specified on figure legends. Detailed information on the siRNAs used in this study is provided in Supplementary Table 5.

## Western blot
Cells were plated in 6-well plates at a seeding density of 400,000 cells per well and allowed to attach and grow for 48 h. After a 6-h treatment

period, cells were lysed and harvested using Pierce RIPA buffer containing a Pierce Protease and Phosphatase inhibitor tablet (Thermo Fisher Scientific). Following sonication (10 s), the protein lysates were centrifuged for 15-min at 4 °C. Protein concentrations were then determined using the Pierce BCA Protein Assay Kit (Thermo Fisher Scientific). A total of 25 μg of protein were loaded on 4–12% NuPAGE Bis-Tris gels (Invitrogen), separated via electrophoresis and subsequently transferred (30 V, overnight) to Immobilon-P PVDF membranes (pore size 0.45 μm). The membranes were incubated with primary and secondary antibodies (Supplementary Table 6) in 5% milk in TBS supplemented with 0.1% Tween20 (Sigma-Aldrich). Chemiluminescence was detected using the ChemiDoc Touch Imaging System (Bio-Rad).

### Co-immunoprecipitation

Cells were seeded in 100 mm dishes at a density of 6 million cells, with three dishes per experimental condition. Cells were treated 48 h after seeding. Following a 6-h treatment, cells were washed with cold PBS and lysed in HMKEN buffer (10 mM HEPES, pH 7.2, 5 mM $MgCl_2$, 142 mM KCl, 2 mM EGTA, 0.2% IGEPAL® CA-630, protease, and phosphatase inhibitors). The lysate samples were triturated, and then preclearing was performed by incubating with 1:1 protein A:G Dynabeads (Life Technologies) on a rotator at 4 °C for 1-h. Primary antibodies [BCLXL (#2764, Cell Signaling), MCL1 (16225-1-AP, Proteintech) and BIM (#2933, Cell Signaling Technology)] at a concentration of 0.45 μg were added to the precleared lysates and incubated on a rotator at 4 °C overnight. Subsequently, samples were incubated with 1:1 protein A:G Dynabeads for 2 h at 4 °C. Beads underwent five washes in HMKEN buffer, were resuspended in 50 μL of HMKEN buffer supplemented with reducing and loading buffer. Samples were processed for western blotting.

### Proximity ligation assay

Duolink® proximity ligation assay (PLA) was performed on formalin-fixed paraffin-embedded sections from C4-2 cells treated with ipatasertib (1 μM), AZD5991 (1 μM), combined treatment and control (DMSO). Antigen retrieval was performed on a Bond RX autostainer (Leica Biosystems) for 10-min with Bond ER2 solution (AR9640, Leica Biosystems). The following interactions were studied using the red fluorescence assay (DUO92008, Sigma-Aldrich): BAD (1:100, ab32445, abcam)-BCLXL (1:50, ab77571, abcam) and BIM (1:250, #2933, Cell Signaling Technology)-MCL1 (1:100, sc-69838, Santa Cruz). Antibody cocktails were incubated at room temperature for 1-h and the remaining protocol was performed according to manufacturer's instructions (Sigma–Aldrich). Duolink® in situ mouse PLUS and rabbit MINUS probes were used. The omission of one antibody from an interaction was used as negative control. At the end of the procedure the slides were mounted using Duolink® In Situ Mounting Medium which contains DAPI for nuclear staining. Imaging was performed and PLA dots were counted using pathologist supervised HALO™ image analysis software (Indica Labs).

### In vivo analyses with the CP253C CRPC PDX model

All procedures involving mice were conducted in compliance with the Institute of Cancer Research guidelines and approved by the ICR Animal Welfare and Ethical Review Body, adhering to the UK Animals (Scientific Procedures) Act 1986. Animals were housed in a pathogen-free facility in ventilated cages, at 20–24 °C and 40–60% relative humidity, with a 12 h light/dark cycle. CP253c PDX tumours were implanted into the left flank of NSG mice (7 weeks old at implantation; $n = 31$). Sex was considered in the study design and only male mice were used in the experiment given the disease aetiology. Upon reaching a tumour size of 50–300 mm³, mice were randomly assigned to one of six treatment arms: arm 1 received the vehicle, arm 2 was administered ipatasertib (50 mg/kg), arm 3 received S63845

(25 mg/kg), arm 4 was treated with fadraciclib (40 mg/kg), arm 5 received a combination of ipatasertib (50 mg/kg) and S63845 (25 mg/kg), and arm 6 received a combination of ipatasertib (50 mg/kg) and fadraciclib (40 mg/kg). Ipatasertib and fadraciclib were reconstituted in a solution of 0.5% methylcellulose and 0.2% Tween 80 in water, while S63845 was reconstituted in a solution containing 25 mM HCl and 20% hydroxypropyl-beta-cyclodextrin in water. Ipatasertib and fadraciclib were administered orally 5 days/week (Monday–Friday), while S63845 was delivered intravenously 3 days/week (Monday, Wednesday, and Friday). Tumour measurements and body weight were taken every 2–4 days, grouped in 3-day intervals. Mice were monitored for potential signs of discomfort or distress, including changes in physical appearance (facial expression, piloerection, hunched posture), behaviour (reduced activity), and physiological signs (abnormal breathing or altered food/water intake). The experiment concluded either 24 days after the initial treatment or earlier if tumours approached 14 mm diameter. None of the tumours exceeded the approved maximal size/burden allowed by our license (17 mm diameter). Mice were euthanized 6-h after the final dose, and tumour samples were collected for IHC analysis.

### Development of drug-resistant cell lines

LNCaP95 and C4-2 cells were gradually exposed to increasing concentrations of capivasertib to develop capivasertib-resistant (Capi-R) cells, while LNCaP95 cells were similarly treated with AZD5991 or S63845 to generate AZD5991- and S63845-resistant (AZD5991-R and S63845-R) cells. Parental cell lines were maintained in parallel with 0.01% DMSO as a vehicle control. Resistance was defined after 6 days of drug exposure: Capi-R cells showed a >10-fold increase in capivasertib IC50, while AZD5991-R and S63845-R cells showed no significant change in growth when treated with 1 μM AZD5991 or S63845, respectively. Mycoplasma testing and STR profiling confirmed cell line identity and integrity.

### Statistical analyses

All statistical analyses were performed using R Statistical Software version 4.1.3 (R Core Team, Vienna, Austria) or GraphPad Prism version 10 (GraphPad Software Inc., San Diego, California, USA). OS and PFI analyses were performed using Kaplan-Meier methodology and the log-rank test. The Mann-Whitney $U$ test was used to compare pathologic tumour response in the NCI neoadjuvant cohort and to compare RNA expression, between tumours with *MCL1* copy number gain/amplification and those without. Correlation analyses were performed with Spearman's rank correlation coefficient. The linear mixed effects model was employed to compare the growth rate between the different treatment arms, and the vehicle, in the CP253c in vivo experiment and was performed using the R package nlme. Estimates and p-values refer to the interaction term of treatment arm and time indicating the difference in tumour volume growth rate between each treatment arm and the vehicle arm. Zero Interaction Potency (ZIP) synergy score was calculated, and ZIP surface plots were downloaded, using SynergyFinder+ web application[82,83]. Statistical comparisons of cell viability or caspase 3/7 activity between the control (or scramble) and treatments (or siRNAs) were conducted using one-way ANOVA. Post-hoc Tukey tests were employed for comparisons among all groups, whereas post-hoc Dunnett tests were used for comparisons against the control group. Mean and standard deviation were presented unless specified otherwise. A two-sided $p$ value ≤ 0.05 was deemed to be statistically significant. All experiments were performed in biological triplicates ($n = 3$) unless stated otherwise.

### Reporting summary

Further information on research design is available in the Nature Portfolio Reporting Summary linked to this article.

## Data availability

Source Data files are provided with this paper. All raw data are available in the Source Data. Transcriptomic and copy number variation datasets used in this study have been previously published and made available[19,45,62,65–68]. The datasets used in this study are publicly available from the following sources: The TCGA data can be accessed in cBioPortal. The SU2C/PCF cohort data are available in cBioPortal and on GitHub under the code prad_su2c_2019. The RMH cohort data are available in the European Nucleotide Archive under accession number PRJEB32038. Finally, the NCI neoadjuvant cohort data have been deposited in the Database of Genotypes and Phenotypes (dbGaP) under accession number phs001938.v2.p1. No new omics datasets were generated. Further reasonable data access requests can be submitted to the corresponding authors. Source data are provided with this paper.

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

## Acknowledgements

The authors gratefully acknowledge the patients and the families of patients who contributed to this study. This work was supported by Prostate Cancer UK (TLD-PF19-006 to J.M.J.-V.; Career Acceleration Fellowship to J.T.; Research Funding to J.S.dB.), the Department of Defence Prostate Cancer Research Programme (Early Investigator Research Award to J.M.J-V. and S.W.; Impact Awards to A.G.S. and S.P.B.; Idea Development Award to A.G.S.), Cancer Research UK (Clinical Research Training Fellowship to D.W. and N.W.; Radiation Research Network Seed Funding to A.Sharp; Convergence Centre Grant funding, Centre Programme and Experimental Cancer Medicine Centre grants to J.S.dB.), MINECO (FPU18/02485 to A.J.M-H.; PID2022-138185OB-I00 to R.M.L) National Cancer Institute/National Institutes of Health (R50CA274336, P50CA0907186, P01CA163227, and R21CA277368 to I.C. and P.S.N.; the Intramural Research Programme of the Center for Cancer Research, Research Funding to W.D.), Royal Society of Medicine (International Exchanges Award IES\R3\213131 to A.Sharp and N.A.L.), Terry Fox Research Institute New Frontiers Programme (F22-00589 to N.A.L.), Prostate Cancer Foundation Canada grant-in-aide (to N.A.L.), American Association for Cancer Research (AACR Clinical/Translational Cancer Research Fellowship 18-40-11-BEZZ to M.B.), Medical Research Council (Research Funding to J.S.d.b.), the Prostate Cancer Foundation (Young Investigator Award to S.W.; Challenge Award to S.P.B., J.S.dB. and A.S.), the Movember Foundation through the London Movember Centre of Excellence (CEO13 2-002 to J.S.d.B.), the National Institute for Health and Care Research Biomedical Research Centre (pump priming award to A.Sharp), the Wellcome Trust (Clinical Research Career Development Fellowship to A.Sharp). Portions of this work was supported by the NCI (HHSN261200800001E), the Intramural Research Programme of the NCI, NIH, and the NIH National Center for Advancing Translational Sciences. Portions of this work utilised the computational resources of the NIH HPC Biowulf cluster. The authors thank the patients, families and investigators, and the Institute for Prostate Cancer Research for supporting the University of Washington Tissue Acquisition Programme. The content of this publication does not necessarily reflect the views or policies of the Department of Health and Human Services or the Department of Defence, nor does mention of trade names, commercial products, or organisation imply endorsement by the U.S. Government. J.S.d.B. is a National Institute for Health Research (NIHR) Senior Investigator. The views expressed in this article are those of the author(s) and not necessarily those of the National Health Service, the NIHR, or the Department of Health, nor does mention of trade names, commercial products, or organisation imply endorsement by the UK Government.

## Author contributions

J.M.J.-V., D.W., J.S.d.b. and A.Sharp. conceived and designed the study. J.M.J.-V., D.W., I.F., A.DH.B., A.P., B.G., C.B., S.M., M.L., A.J.M.-H., I.C., I.P.L.Y., L.B., W.Z., A.J.N., J.W., R.P., F.G., N.P., A.F., M.C., R.R., S.D., J.T., N.W., E.H., M.V., J.N., I.B., K.L., T.P., P.G., S.W., S.Y.T., F.K., C.H.C., E.L.B., X.Z., C.K.-T. and A.V. performed experiments and/or collected data. J.M.J.-V., D.W., W.Y., G.S., D.B. and J.R. analysed the data and/or performed statistical analyses. J.M.J.-V., D.W., R.M.L., A.S., F.R., N.A.L., C.J.T., G.H., W.D.F., M.B., A.G.S., P.S.N., S.C., S.P.B., J.S.d.B. and A.Sharp. interpreted the results and/or contributed to data visualisation. J.M.J.-V., D.W., J.S.d.B. and A.Sharp draughted the manuscript. All authors provided advice on, reviewed, edited and approved the manuscript. J.S.d.B. and A.Sharp served as sponsors and monitored the study.

## Competing interests

J.M.J.-V., D.W., I.F., A.d.H.V., A.P., W.Y., G.S., D.B., B.G., C.B., S.M., L.B., W.Z., A.J.N., J.W., J.R., R.P., A.F., M.C., R.R., S.D., J.T., E.H., M.V., J.N., K.L., A.S., F.R., S.C., J.S.d.B. and A.Sharp are employees of the ICR, which has a commercial interest in abiraterone, PARP inhibition in DNA repair defective cancers, and PI3K/AKT pathway inhibitors (no personal income). A.G.S. reports that the National Cancer Institute (NCI) has a Cooperative Research and Development Agreement (CRADA) with Astellas. Resources are provided by this CRADA to the NCI. A.G.S. gets no personal funding from this CRADA but is the primary investigator of the CRADA. J.S.d.B. has served on advisory boards and received fees from many companies, including Amgen, Astra Zeneca, Bayer, Bioxcel Therapeutics, Daiichi, Genentech/Roche, GSK, Merck Serono, Merck Sharp & Dohme, Pfizer, and Sanofi Aventis. He is an employee of the ICR, which has received funding or other support for his research work from AZ, Astellas, Bayer, Cellcentric, Daiichi, Genentech, Genmab, GSK, Janssen, Merck Serono, MSD, Menarini/Silicon Biosystems, Orion, Sanofi Aventis, Sierra Oncology, Taiho, Pfizer, Vertex. J.S.d.B. was named as an inventor, with no financial interest, for patent 8,822,438, submitted by Janssen, that covers the use of abiraterone acetate with corticosteroids. J.S.d.B. has been the CI/PI of many industry-sponsored clinical trials. A.Sharp has received travel support from Sanofi, Roche-Genentech and Nurix, and speaker honoraria from Astellas Pharma and Merck Sharp & Dohme. He has served as an advisor to DE Shaw Research, CHARM Therapeutics, Ellipses Pharma and Droia Ventures. A.Sharp has been the CI/PI of industry-sponsored clinical trials. The remaining authors declare no conflicts of interest.

## Additional information

Juan M. Jiménez-Vacas[1,11], Daniel Westaby[1,2,11], Ines Figueiredo [1], Alexis De Haven Brandon[1], Ana Padilha[1], Wei Yuan [1], George Seed [1], Denisa Bogdan[1], Bora Gurel [1], Claudia Bertan[1], Susana Miranda [1], Maryou Lambros[1], Antonio J. Montero-Hidalgo [1,3], Ilsa Coleman[4], Ivan Pak Lok Yu[5], Lorenzo Buroni[1], Wanting Zeng [1], Antje J. Neeb[1], Jon Welti[1], Jan Rekowski [1], Roberta Paravati[1], Florian Gabel[1], Nicole Pandell[1], Ana Ferreira[1], Mateus Crespo [1], Ruth Riisnaes[1], Souvik Das[1], Joe Taylor[1], Nick Waldron[1], Emily Hobern[1], Melanie Valenti[1], Jian Ning[1], Ilona Bernett[1], Kate Liodaki[1], Thomas Persse[4], Patricia Galipeau [4], Scott Wilkinson [6], Shana Y. Trostel[6], Fatima Karzai [6], Cindy H. Chau [6], Erica L. Beatson[6], Xiaohu Zhang[7], Carleen Klumpp-Thomas [7], Andreas Varkaris[8], Raul M. Luque[3], Amanda Swain [1], Florence Raynaud [1], Nathan A. Lack [5,9], Craig J. Thomas [7,10], Gavin Ha [4], William D. Figg [6], Marco Bezzi[1], Adam G. Sowalsky [6], Peter S. Nelson[4], Suzanne Carreira [1], Steven P. Balk [8], Johann S. de Bono [1,2,12] ✉ & Adam Sharp [1,2,12] ✉

[1]The Institute of Cancer Research, London, UK. [2]The Royal Marsden Hospital, London, UK. [3]Maimonides Institute for Biomedical Research of Cordoba (IMIBIC); Department of Cell Biology, Physiology, and Immunology, University of Cordoba; Reina Sofia University Hospital; CIBER Physiopathology of Obesity and Nutrition (CIBERobn), Cordoba, Spain. [4]Divisions of Human Biology and Clinical Research, Fred Hutchinson Cancer Center, Seattle, WA, USA. [5]Vancouver Prostate Centre, University of British Columbia, Vancouver, BC, Canada. [6]Genitourinary Malignancies Branch, Center for Cancer Research, National Cancer Institute, National Institutes of Health, Bethesda, MD, USA. [7]Division of Preclinical Innovation, National Center for Advancing Translational Sciences, National Institute of Health, Rockville, MD, USA. [8]Hematology-Oncology Division, Beth Israel Deaconess Medical Center, Boston, MA, USA. [9]Department of Medical Pharmacology, School of Medicine, Koc University, Istanbul, Turkey. [10]Lymphoid Malignancies Branch, Center for Cancer Research, National Cancer Institute, National Institutes of Health, Bethesda, MD, USA. [11]These authors contributed equally: Juan M. Jiménez-Vacas, Daniel Westaby. [12]These authors jointly supervised this work: Johann S. de Bono, Adam Sharp. ✉e-mail: johann.de-bono@icr.ac.uk; adam.sharp@icr.ac.uk

