## [Transparent Peer Review file · Nature Communications]

Elucidating molecularly stratified single agent, and combination, therapeutic strategies targeting MCL1 for lethal prostate cancer

Corresponding Author: Dr Adam Sharp

Version 0:

Reviewer comments:

Reviewer #1

(Remarks to the Author)

This study by Jiménez-Vacas and colleagues investigates a role for anti-apoptotic MCL-1 in prostate cancer. The authors show increased expression of MCL1 (copy number gain) tracks with lethal PC, the authors then demonstrate single agent activity of MCL1 inhibition in patient derived organoids (and other models), extending this (on the back of resistance to single-agent MCL-1 inhibitors is likely), the authors then rationally apply AKT inhibitors alongside MCL-1 inhibition demonstrating very potent synergistic effects as well as describing the underlying mechanism of synergy. Finally, they perturb MCL-1 expression through CDK9 inhibition, this again supporting the thesis that MCL-1 represents a potentially important therapeutic target in prostate cancer. This study is a tour-de-force, uses many different approaches to investigate MCL-1 activity (and its validity as a drug target) in prostate cancer – the data supports the authors' conclusion that inhibition of MCL-1 may be a valid therapeutic approach in prostate cancer, particularly in combination with targeting additional pro-survival checkpoints. My comments to be considered for revision are below.

- the authors predominantly correlate effects of MCL-1 on PC progression/lethality by measuring RNA levels/copy number variation. MCL-1 expression is heavily regulated at the level of protein stability, hence (if anything) the authors' approach may be underestimating the importance of MCL-1 in PC – have the authors done corresponding IHC analysis on some of their patient samples to measure total MCL-1 level and (if so), how does this analysis look like? – in the absence of such analysis, some discussion of further work investigating MCL-1 protein levels in PC should be included.

- a potential elephant in the room is the apparent (on-target) cardiotoxicity of MCL-1 inhibitors, which has made their clinical progression challenging. While I'm not suggesting the authors solve this, they should at least discuss this problem and potential ways to circumvent this (e.g. target PC specific regulators of MCL-1 expression).

- the combination of MCL-1 inhibitor (S683) alongside AKT inhibitor shows no apparent toxicity in mice (Supp Figure 15), important to note (and include discussion here) that S683 (and other MCL-1 inhibitors) show less affinity for mouse MCL-1 than human MCL-1, thus potentially underestimating toxic effects in mice.

Reviewer #2

(Remarks to the Author)

This manuscript reports that MCL1 is frequently amplified and overexpressed in metastatic prostate cancers and MCL1 inhibitors are active prostate cancer models, including organoids from PDX models (PDX-O) with MCL1 amplification/overexpression. Also, combinations of MCL1 inhibitors with PI3K and AKT inhibitors were tested showing a marked synergy with the combined approaches in prostate cell lines and PDX-O. Overall, the main observations presented here have been reported before in previous manuscripts and in many tumor types, including prostate and other types of solid tumors in addition to hematological malignancies. There are also concerns regarding the models, experimental design, and data (see specific comments).

Most data are not novel and there are no relevant new insights gained by this work, particularly on the relevance of MCL1 in prostate cancer progression to castration resistance and the feasibility and safety of implementing MCL1 inhibitors for treating prostate cancer patients. Re-analysis of public databases shows that MCL1 is amplified and overexpressed in metastatic prostate cancers. However, the interesting link between MCL1 and castration resistance here is not further examined. The activity of MCL1 inhibitors in prostate cancer has been examined before in previous publications. MCL1 inhibitors have limited efficacy in prostate cancer and other solid tumors, a finding that has hindered their clinical development and a question that is not solved by this work. Combinations of MCL1 inhibitors with targeted inhibitors of signaling pathways have been widely tested in many tumor types, including prostate cancer. The finding that PI3K and AKT inhibitors can synergize with MCL1 inhibitors is not novel and particularly relevant.

Specific comments

Analysis of public patients' datasets confirms the frequent amplification, copy number gain and upregulation of mRNA and protein level of MCL1 in PC, CSPC and CRPC. This is a very comprehensive analysis. However, the findings do not add new insights regarding the potential involvement of MCL1 in prostate cancer progression and castration resistance. It is only descriptive work without any mechanistic insight.

MCL1 inhibitors like AZD5991 are initially assessed in four PDX-O models with and without MCL1 amplification and overexpression. Only PDX-Os with amplification of MCL1 appear to respond to the MCL1 inhibitors. No information is reported on specificity and selectivity toward MCL1 of these inhibitors. Measurements of caspase activation and cell viability are done within a limited time frame (6, 24, and 96 hours). This is a short time frame to see the effects on tumor cell viability and proliferation in organoid cultures. Could the low MCL1 expressing PDX-O models respond with longer exposure times? It is not clear why the combinations of MCL1, PI3K and AKT inhibitors were run in LNCAP 95 and C4-2 cell lines. Do they have amplification or overexpression of MCL1? Testing in the PDX-O models would have been more informative. The combinations of MCL1 inhibitors with PI3K and AKT inhibitors are of limited interest and novelty, also considering the limited efficacy and therapeutic index of the single agents. The synergy of these combinations has been described before. These data confirm the potential of the combined therapies in prostate cancer models and the expected mechanism related to the disruption of BAD-BCLXL or BIM-MCL1 interactions.

Based on the analysis of a limited set of cell lines and PDX-Os the authors conclude that PTEN loss increases the sensitivity to the combination of MCL1 and AKT inhibitors. However, this seems an overstatement. Comparing two cell lines with and without PTEN loss is not sufficient to support this conclusion, considering the multiple different genomic alterations acquired by these tumor cell lines. Furthermore, only PDX-Os with PTEN pathway alterations were included in the experiment.

Additional in vivo experiments would be required to evaluate the antitumor efficacy of MCL1 inhibitors alone and in combinations with AKT inhibitors. Only the CP253c PDX (with minimal MCL1 expression) was used here. It would be more informative to test additional models, including the high MCL1 expressing PDXs, to see whether their increased sensitivity in vitro translate in more relevant responses in vivo to single or combined treatment. The number of mice in the PDX experiment is also minimal (n=4-5) considering the variability in individual tumor growth.

Toxicity is a relevant issue given the limited therapeutic index of MCL1 and AKT inhibitors and should be monitored carefully by assessing multiple relevant parameters. A drop of 10-20% in the body weight, as seen with the combinations, suggests potentially relevant toxicity, unlike the authors' statement.

Evaluating the response of MCL1 inhibitors in cells resistant to AKT inhibitors is a nice addition to this body of work, but it seems of limited relevance in this context. Resistant cells respond to MCL1 inhibitors like the parental cells. Is this not expected? What about cells with innate or acquired resistance to MCL1 inhibitors?

CDK9 inhibitors act as broad transcriptional inhibitors and induce very complex responses in tumor cells. One cannot claim they are blocking MCL1. There is no specificity for MCL1, and their effects are likely dependent on many other unrelated mechanisms. The finding of a synergistic interaction between CDK9 and MCL1 inhibitors is interesting but there is no mechanistic rationale or compelling supporting evidence. The set of data does not add to the main message, and the conclusions are highly speculative and not supported by data.

Reviewer #3

(Remarks to the Author)

Manuscript: NCOMMS-24-14228-T

Title: Elucidating molecularly stratified single agent, and combination, therapeutic strategies targeting MCL1 for lethal prostate cancer

Summary

Prostate cancer is a very common cancer in elderly men. The 5-year relative survival rate for men diagnosed with local or regional disease is almost 99%, but it drops to less than 30% for patients with distant disease. The identification of new therapeutic strategies for the treatment of advanced stages of prostate cancer is a major clinical need.

Here, Juan M. Jiménez-Vacas, Daniel Westaby and colleagues define the anti-apoptotic factor MCL1 as a valuable molecular target for the treatment of advanced prostate cancer.

MCL1 belongs to the BCL-2 family of antiapoptotic factors and is one of the most frequently amplified genes in cancer. Downregulation, degradation or inhibition of MCL-1 have already been demonstrated a promising strategy to overcome tumor resistance to standard and targeted therapies. Potent MCL-1i are now tested in clinical trials, however identification of molecular mechanisms and signaling pathways favoring or limiting their anti-tumor activity will be crucial to capitalize on

MCL-1 targeting.

Leveraging on a broad spectrum of in vitro and in vivo prostate cancer models, the authors demonstrate the efficacy of the pharmacological inhibition of MCL1 in prostate tumors characterized by MCL1 gene amplification and PI3K/AKT overactivity. The synergistic effect of combining MCL1 inhibitors with AKT inhibitors promotes apoptosis in both therapy-naïve and therapy-resistant PCa models, in vitro as well as in vivo. Mechanistically, synergy converges on the interactions between BCL2 and BH3 proteins, promoting the formation of BAD-BCLXL and MCL1-BIM complexes and, consequently, unlocking the apoptotic process.

The manuscript is a solid demonstration of the relevance of combining molecular targeting and cancer stratification based on genetic and molecular criteria. It is well written (although some words such as orthogonal and synergy are used too frequently), the data presented are convincing and support the conclusions.

Below are a few suggestions that can improve the work.

These are mainly minor revisions, with the exclusion of points 11, 14, 15 and 16 for which I would ask the authors to pay particular attention.

1. The manuscript is too dense with data that sometimes weigh down the reading and the reasoning. I would suggest streamlining the work by removing some of the data or moving it to supplementary.
2. I would stratify the 4 patients with MCL1 gain/amplification in two distinct categories, partially sensitive (n=2) and insensitive (n=2).
3. Supplementary Figure 1 and 2 are not mentioned in the manuscript, Supplementary Figure 4 is mentioned before Supplementary Figure 3.
4. Absence of MCL1 gain/amplification association with Taxane response is interesting and should be discussed. Is the mechanism of mitotic catastrophe the possible reason?
5. Mito localization of MCL1 should be tested by immunofluorescence with a specific marker of mitochondria. Optical density data are not convincing. This analysis should be normalized for cell size. Metastatic cells in the lymph node appear smaller than those in the lung.
6. R value for the correlation of MCL1 with RELA and STAT3 transcripts is minimal. The association between MCL1 transcript amounts and TNF α /IL6-STAT signaling via SASP is speculative and not supported by scientific evidence. Section 3.4 MCL1 RNA associates with key signaling pathways in mCRPC is poorly informative, I would suggest removing this part from the results.
7. Nucleolar staining of MCL1 in PDX-O is surprising. Is it specific?
8. Caspase assay in CP267c shows Caspase 3/7 activity similar to those measured in CP336c and CP253c, however cell viability after 96 hours of MCL1 inhibition is substantially different. Authors should explain this inconsistency. Moreover, why inhibition of an anti-apoptotic factor such as MCL1 should be sufficient to trigger CRPC cell death?
9. Why did the authors use cell lines for FDA-approved drug screening instead of using PDX-Os? Differently by PDX-Os, prostate cancer cell lines respond little (LNCaP95) or not at all (LNCaP, C4-2, 22RV1, PC3, Du145) to MCL1 inhibitors administered as a single treatment (Figure 4 and Supplementary Figure 7 Vehicle plus MCL1 inhibitor). Definition of MCL1 status (gene/mRNA/protein) in PCa cell lines would help in the interpretation of the results obtained.
10. The analyses shown in Figure 2 do not demonstrate a synergistic effect of concomitant inhibition of PI3K and MCL1. Figure 3A formally demonstrates synergy.
11. Western blot analysis shows less cleaved Caspase 3 in the combo lpa + AZ compared to Cap + AZ in all triplicates of both cell lines (Suppl. Figure 8C). This is nicely recapitulated by siMCL1 plus lpa versus siMCL1 plus Cap in Suppl Figure 10. Nevertheless, PARP cleavage and cells viability are comparable in cells treated with lpa+AZ or Cap+AZ. Did the authors check for the activation of Caspase 7? Is active Caspase 7 more similar in PCa cells treated with the different combo?
12. AKT competitive inhibitors are known to induce AKT phosphorylation at both T308 and S473 sites. This is clearly shown in WB of C4-2 (Suppl Figure 10F) and LNCaP95 (10E). In LNCaP95, however, phosphorylation increases exceptionally when AKT inhibitors are combined with MCL1 knockdown. This is an interesting aspect that should be considered, especially since it does not occur when AKT inhibitors are combined with AZ (Figure 3B).
13. Authors should highlight with a mark and corresponding explanation in the figure legend the technical issues that occurred in the WB of Suppl. Fig. 12 (e.g., LNCaP95-A: lpa+AZ/BAD siRNA, BAK levels are strongly reduced; LNCaP95-B: lpa+AZ/BAD siRNA, BIM levels are strongly reduced; LNCaP95-C: Veh/Scr the amount of BAK is too low compared to the amount of BAK in BIM and BAD siRNA lines).
14. As expected, AKTi increases the BAD/BCLXL interaction. In contrast, the increase in MCL1/BIM interaction is not equally evident in the IP experiments reported in panel 3E. The PLA data described in Suppl. Fig. 13 support the minimal, if

any, change in MCL1/BIM interaction in the presence of lpa. This is a critical point, as these data may represent the molecular explanation for the synergy between AKT and MCL1 inhibition. The authors should produce more convincing data to support their conclusions. Moreover, because BAK is a main target of MCL1 antiapoptotic activity, the authors should analyze the interaction between MCL1 and BAK in PCa cells untreated and in presence of lpa, AZ, or their combination.

15. PTEN IHC in Figure 4C should be accompanied with pSer473-AKT staining (Cell Signaling antibodies against pAKT work wonderfully on FFPE sections). A PDX with no alteration in PTEN and PI3K pathway should be used to demonstrate AKT hyper-activation in CP50c, CP253c, CP267c, CP336c.

16. The plot in Figure 5D describes Cleaved Caspase 3 quantification in PDX treated with vehicle, lpa, S63, or their combo. lpa+S63 treatment shows heterogeneous results with three of the five samples characterized by similar amounts of Cleaved Caspase 3-positive cells compared to Veh, lpa or S63. Similar variability is evident for the immunoscore of pPRAS40. If GSK3b is more sensitive to minimal AKT inhibition than PRAS40, could the results suggest reduced lpa activity in non-responsive PDX treated with the combination? The authors should comment on this variability.

17. The sentences "Caspase 3/7 was induced" or "a synergistic increase in caspase 3/7 levels" are incorrect. The authors estimated the activity (or activation) of caspase 3/7, or the presence of the active form of the protein.

Version 1:

Reviewer comments:

Reviewer #1

(Remarks to the Author)

The authors have comprehensively addressed all comments I raised.

Reviewer #2

(Remarks to the Author)

The author's response to my comments is satisfactory.

They have performed additional experiments including multiple models.

They have also amended the test when necessary to clarify all the issues that I have raised.

I found that the manuscript is improved.

Reviewer #3

(Remarks to the Author)

The authors responded thoroughly to all my comments and suggestions.

Reviewer 1

This study by Jiménez-Vacas and colleagues investigates a role for anti-apoptotic MCL-1 in prostate cancer. The authors show increased expression of MCL1 (copy number gain) tracks with lethal PC, the authors then demonstrate single agent activity of MCL1 inhibition in patient derived organoids (and other models), extending this (on the back of resistance to single-agent MCL-1 inhibitors is likely), the authors then rationally apply AKT inhibitors alongside MCL-1 inhibition demonstrating very potent synergistic effects as well as describing the underlying mechanism of synergy. Finally, they perturb MCL-1 expression through CDK9 inhibition, this again supporting the thesis that MCL-1 represents a potentially important therapeutic target in prostate cancer. This study is a tour-de-force, uses many different approaches to investigate MCL-1 activity (and its validity as a drug target) in prostate cancer – the data supports the authors' conclusion that inhibition of MCL-1 may be a valid therapeutic approach in prostate cancer, particularly in combination with targeting additional pro-survival checkpoints. My comments to be considered for revision are below.

Authors' response: We are grateful to the reviewer for the positive feedback and hope the below actions have satisfactorily addressed their comments.

Reviewer's comment 1: *The authors predominantly correlate effects of MCL-1 on PC progression/lethality by measuring RNA levels/copy number variation. MCL-1 is expression is heavily regulated at the level of protein stability, hence (if anything) the authors' approach may be underestimating the importance of MCL-1 in PC – have the authors done corresponding IHC analysis on some of their patient samples to measure total MCL-1 level and (if so), how does this analysis look like ? – in the absence of such analysis, some discussion of further work investigating MCL-1 protein levels in PC should be included.*

Authors' response: As suggested by the reviewer, we evaluated whether MCL1 protein levels are associated with PC progression and lethality. In our IHC cohort of 30 CRPC samples, 29 had linked overall survival data. Patients with tumours having MCL1 H-Score ≥ 100 had a shorter overall survival (HR 2.09, 95% CI 0.85–5.12, $p=0.11$) compared to those with H-Score < 100 , although these differences were not statistically significant, likely due to the small sample size. This trend aligns with our hypothesis that high MCL1 levels correlate with worse outcomes.

These new data have been included in the revised manuscript as **Supplementary Figure 6**. Additionally, we have included the following information in the Results section "*In addition, CRPC patients with higher MCL1-expressing tumours (H-Score ≥ 100) were associated with shorter overall survival (HR 2.09, 95% CI 0.85–5.12, $p=0.11$) (Supplementary Figure 6)*" (lines 504-506).

Supplementary Figure 6: Association of MCL1 protein expression with overall survival in CRPC patients. (Left) Dot plot representing MCL1 protein levels (H-Score) measured in a cohort of 29 CRPC patient samples. Each dot represents an individual patient, with red dots indicating tumours with MCL1 H-Score ≥ 100 and grey dots indicating tumours with H-Score < 100 . (Right) Kaplan-Meier survival curves showing overall survival from biopsy, stratified by MCL1 protein expression levels. Patients with MCL1 H-Score ≥ 100 are represented by the red line, and those with H-Score < 100 by the grey line. Hazard ratio (HR) with 95% confidence intervals and p-value for log-rank test are shown. HS: H-Score.

Reviewer's comment 2: *A potential elephant in the room is the apparent (on-target) cardiotoxicity of MCL-1 inhibitors, which has made their clinical progression challenging. While I'm not suggesting the authors solve this, they should at least discuss this problem and potential ways to circumvent this (e.g. target PC specific regulators of MCL-1 expression).*

Authors' response: We agree with the Reviewer and have now included the following text in the discussion: *"It is important to acknowledge the challenges associated with direct MCL1 inhibition, particularly due to on-target cardiotoxicity (80, 81). In this context, future strategies, including antibody-drug conjugates and proteolysis-targeting chimeras (PROTACs), similar to those developed for other BCL2 family members, could enable PC-specific MCL1 targeting while minimising the risk of cardiotoxicity (82, 83)"* (lines 831-835).

Reviewer's comment 3: *The combination of MCL-1 inhibitor (S683) alongside AKT inhibitor shows no apparent toxicity in mice (Supp Figure 15), important to note (and include discussion here) that S683 (and other MCL-1 inhibitors) show less affinity for mouse MCL-1 than human MCL-1, thus potentially underestimating toxic effects in mice.*

Authors' response: We thank the reviewer for the valuable comment. We have now discussed this point in the result section of the manuscript as follows *"No mice exhibited weight loss*

exceeding 20% of their baseline or showed any signs of discomfort or distress (Supplementary Figure 20). However, this should be taken with caution as S63845 (and other MCL1 inhibitors) show less affinity for mouse MCL1 than human MCL1 (31)" (lines 651-654). The newly included experiments with mouse organoids support the reviewer's point and confirm an on-target effect of the MCL1 inhibitors used. Synergy with AKT inhibitors was observed only at MCL1 inhibitor concentrations of 5 μ M and 10 μ M, consistent with the sixfold lower affinity of these inhibitors for mouse MCL1 compared to human MCL1. These results have been included as Figure 4H-J.

Figure 4. AKT and MCL1 co-inhibition triggers apoptosis through dysregulation of BAD, BIM and BAK interactions in prostate cancer harbouring PI3K/AKT pathway hyperactivating aberrations. (H-J) Prostate Cancer Preclinical Mouse Modelling Platform organoids (ProMPt-Os) were treated with vehicle (DMSO), ipatasertib (1 μ M), AZD5991 (1, 5, and 10 μ M), S63845 (1, 5, and 10 μ M), and their combinations to assess caspase 3/7 activity at 6 hours (H), cell viability at 24 hours (I), and cell viability at 96 hours (J). Caspase 3/7 activity and cell viability were assessed by Caspase-Glo 3/7 3D, and CellTiter-Glo 3D, respectively. Data represent the mean \pm standard error (SEM) of three independent biological replicates, each performed with three technical replicates. Asterisks (* p < 0.05; ** p < 0.01; *** p < 0.001) indicate statistically significant differences between groups.

Reviewer 2

This manuscript reports that MCL1 is frequently amplified and overexpressed in metastatic prostate cancers and MCL1 inhibitors are active prostate cancer models, including organoids from PDX models (PDX-O) with MCL1 amplification/overexpression. Also, combinations of MCL1 inhibitors with PI3K and AKT inhibitors were tested showing a marked synergy with the combined approaches in prostate cell lines and PDX-O.

Overall, the main observations presented here have been reported before in previous manuscripts and in many tumor types, including prostate and other types of solid tumors in addition to hematological malignancies. There are also concerns regarding the models, experimental design, and data (see specific comments).

Most data are not novel and there are no relevant new insights gained by this work, particularly on the relevance of MCL1 in prostate cancer progression to castration resistance and the feasibility and safety of implementing MCL1 inhibitors for treating prostate cancer patients. Re-analysis of public databases shows that MCL1 is amplified and overexpressed in metastatic prostate cancers. However, the interesting link between MCL1 and castration resistance here is not further examined. The activity of MCL1 inhibitors in prostate cancer has been examined before in previous publications. MCL1 inhibitors have limited efficacy in prostate cancer and other solid tumors, a finding that has hindered their clinical development and a question that is not solved by this work. Combinations of MCL1 inhibitors with targeted inhibitors of signaling pathways have been widely tested in many tumor types, including prostate cancer. The finding that PI3K and AKT inhibitors can synergize with MCL1 inhibitors is not novel and particularly relevant.

Authors' Response: We sincerely thank the Reviewer for the thoughtful and constructive comments and suggestions, which have guided us in significantly improving our study. In response to the reviewer's comments, we have included six additional models, including one CRPC patient-derived xenograft organoid, three mouse organoids, and two drug-resistant cell lines (AZD5991- and S63845-resistant LNCaP95 cells). Additional experiments have been performed to better understand MCL1 biology in advanced prostate cancer (PC). We provide detailed answers below and hope our responses thoroughly address the questions raised.

Reviewer's comment 1: *Analysis of public patients' datasets confirms the frequent amplification, copy number gain and upregulation of mRNA and protein level of MCL1 in PC, CSPC and CRPC. This is a very comprehensive analysis. However, the findings do not add new insights regarding the potential involvement of MCL1 in prostate cancer progression and castration resistance. It is only descriptive work without any mechanistic insight.*

Authors' Response: We appreciate the Reviewer's comment and their acknowledgment of the comprehensiveness of our analysis. While our findings build on existing literature indicating that MCL1 is upregulated in response to androgen deprivation in PC cells via a cell cycle-dependent mechanism, conferring resistance to apoptosis (PMID: 25749045), we believe that our study provides novel insights into the role of MCL1 in PC progression.

Specifically, to our knowledge, this is the first study to demonstrate that MCL1 amplification is an early event in PC evolution and that it is associated with the development of CRPC (Figure 1A-B). Furthermore, our data reveal that MCL1 copy number gains become increasingly

prevalent as tumours progress under AR signalling inhibitors (Figure 1A-B and Supplementary Figure 4A).

We therefore suggest that the selection of subclones harbouring MCL1 copy number gains or amplification is a potential mechanism driving therapy resistance. We believe this represents a significant advancement in understanding the molecular dynamics of MCL1 in the context of PC and CRPC. These points have been addressed in detail in the discussion section of our manuscript as follows: *“Indeed, previous studies have shown MCL1 upregulation confers resistance to ADT in PC cells (9), therefore we suggest that the selection of subclones harbouring MCL1 copy number gains or amplifications is a potential mechanism contributing to tumour progression and therapy resistance”* (lines 739-742).

Reviewer’s comment 2: *MCL1 inhibitors like AZD5991 are initially assessed in four PDX-O models with and without MCL1 amplification and overexpression. Only PDX-Os with amplification of MCL1 appear to respond to the MCL1 inhibitors. No information is reported on specificity and selectivity toward MCL1 of these inhibitors. Measurements of caspase activation and cell viability are done within a limited time frame (6, 24, and 96 hours). This is a short time frame to see the effects on tumor cell viability and proliferation in organoid cultures. Could the low MCL1 expressing PDX-O models respond with longer exposure times?*

Authors’ response: We appreciate the reviewer’s input and the opportunity to address these important points.

AZD5991 and S63845 are highly selective MCL1 inhibitors with subnanomolar binding affinities (AZD5991: $K_d = 170$ pM; S63845: $K_d = 190$ pM) and minimal interaction with other BCL-2 family proteins such as BCL-2 and BCL-XL (PMID: 30559424, PMID: 27760111). This information has been included in the methods section of the manuscript as follows: *“AZD5991 and S63845 specificity and selectivity are reported elsewhere (31, 32)”* (lines 261-262).

Regarding the duration of the Caspase-Glo 3/7 3D and CellTiter-Glo 3D assays in our PDX-O models, we acknowledge that the short time frame may not capture longer-term effects on PDX-O viability and proliferation. To address this, we have emphasized in the results section that our focus is on the acute apoptotic response, stating: *“Next, we evaluated the apoptotic response to MCL1 inhibition of five CRPC PDX-O models with varying MCL1 copy numbers and protein levels (Figure 2G-K)”* (lines 523-525), and *“The effects of MCL1 inhibition demonstrated herein reflect short-term apoptosis; thus, the potential impact on PDX-O proliferation may require extended incubation times”* (lines 529-531). Additionally, our in vivo experiment using the CP253c model showed that MCL1 inhibition did not impact tumour growth over 24 days, suggesting a limited effect on long-term proliferation.

Prompted by the reviewer’s comment, we evaluated both the acute apoptotic response (caspase 3/7 activity at 6 hours and cell viability at 24 hours) and longer-term effects (cell viability at 6 days) of all cell lines treated with MCL1 inhibitors. Our results showed that only two cell lines (22Rv1 and LNCaP95) were sensitive to MCL1 inhibition, and importantly, non-responding cells at 24 hours remained unresponsive after 6 days of incubation. These findings have been included as Figure 2D-F and are described in the results section as follows: *“Among the tested cell lines, 22Rv1 showed the highest sensitivity to MCL1 inhibition, as evidenced by*

increased caspase 3/7 activity and reduced cell viability at 24 hours and 6 days (Figure 2D-F). LNCaP95 displayed a comparable response, indicating that sensitivity to MCL1 inhibition can be influenced by factors beyond basal MCL1 copy number or protein levels (Figure 2D-F). The lack of response in DU145 may stem from its reported apoptosis resistance due to BAX deficiency (48) (Figure 2D-F)” (lines 518-523).

Figure 2. PC models with MCL1 copy number gain are sensitive to MCL1 inhibition. (D) The impact of AZD5991 (1 μ M; left-hand graph) and S63845 (1 μ M, right-hand graph) on caspase 3/7 activity was determined using the Caspase-Glo 3/7 2D (at 6 hours) in PNT2, LNCaP, C4-2, LNCaP95, 22Rv1, PC3, and DU145 cell lines. **(E-F)** Cell viability in PNT2, LNCaP, C4-2, LNCaP95, 22Rv1, PC3, and DU145 cell lines after six doses (highest at 1 μ M, decreasing two-fold) of AZD5991 (left) or S63845 (right) was assessed with CellTiter-Glo 3D at 24 hours **(E)** and 6 days **(F)**. The experiment was performed in biological triplicates with five three technical replicates, and the data are presented as fold change relative to the vehicle (DMSO).

Reviewer’s comment 3: It is not clear why the combinations of MCL1, PI3K and AKT inhibitors were run in LNCaP95 and C4-2 cell lines. Do they have amplification or overexpression of MCL1? Testing in the PDX-O models would have been more informative.

Authors’ response: We initially conducted the FDA-approved drug screen (166 drugs) with or without the MCL1 inhibitor AZD5991 in the representative CRPC cell lines C4-2 and LNCaP95 (Figure 3A-B). Our goal was to identify combined therapies that enhance MCL1 inhibition effects in sensitive cells (e.g., LNCaP95) and broaden its impact in resistant cells (e.g., C42) given that CRPCs with MCL1 copy number gains represent a relatively small subset of patients. This information is included in the discussion section as follows “Taking this important point into consideration, and to broaden the effect of MCL1 inhibition in CRPC without MCL1 copy number gains, we explored the efficacy of MCL1 inhibition in combination with FDA-approved oncological drugs with a view to discover synergistic combinations that could be rapidly translated to the clinic” (lines 763-766).

We did not use PDX-O models due to their limited availability and challenges in generating sufficient material required for this drug screen. However, after discovering that AKTi+MCL1i had a more pronounced synergistic effect than PI3Ki+MCL1i, we tested AKTi+MCL1i in seven cell lines (PNT2, LNCaP, C4-2, LNCaP95, 22Rv1, PC3, and DU145) as shown in Figure 4B and Supplementary Figure 17, five PDX-O models (C50c, CP253c, CP267c, CP336c, and the newly included CP142c) as depicted in Figure 4E-G, and three ProMpt-Os (Pten^{KO}/Tp53^{WT}/Myc^{OE}, Pten^{KO}/Tp53^{KO}/Myc^{OE}, and Pten^{WT}/Tp53^{KO}/Myc^{OE}) as shown in Figure 4H-L.

Regarding MCL1 status in C4-2 and LNCaP95 models, we have now included MCL1 copy number and protein levels for all cell lines. Our results indicate that C4-2 and LNCaP95 do not

have MCL1 copy number gain or overexpression, while 22Rv1 is the only cell line showing MCL1 copy number gain and high MCL1 protein levels (Figure 2A-C). This is detailed in the results section as follows "We first characterized seven prostate-derived cell lines and found that 22Rv1 and DU145 harbour MCL1 copy number gain; however, only 22Rv1 exhibited markedly elevated MCL1 protein levels, suggesting additional mechanisms regulate MCL1 protein levels in DU145 (Figure 2A-C)" (lines 514-517).

Figure 2. PC models with MCL1 copy number gain are sensitive to MCL1 inhibition. (A) Violin plot showing the MCL1:Centromere (1p12) ratio in PNT2, LNCaP, C4-2, LNCaP95, 22Rv1, PC-3 and DU145 cell lines (n=50 cells). **(B)** Representative images of MCL1 Fluorescence in situ hybridization (FISH) in prostate-derived cell lines. The median of MCL1:Centromere (1p12) ratio is depicted. The scale bar indicates a length of 5 µm. **(C)** The protein levels of MCL1, BCLXL, and BCL2 in PNT2, LNCaP, C4-2, LNCaP95, 22Rv1, PC3, and DU145 cell lines, with GAPDH as a loading control, were detected by western blot.

Reviewer's comment 4: *The combinations of MCL1 inhibitors with PI3K and AKT inhibitors are of limited interest and novelty, also considering the limited efficacy and therapeutic index of the single agents. The synergy of these combinations has been described before. These data confirm the potential of the combined therapies in prostate cancer models and the expected mechanism related to the disruption of BAD-BCLXL or BIM-MCL1 interactions.*

Authors' response: We fully acknowledge the importance of addressing concerns regarding the combination of MCL1 inhibitors with PI3K/AKT inhibitors.

Even though AKT inhibition has demonstrated superior anti-tumour activity when combined with abiraterone versus abiraterone alone in CRPC patients with PTEN-loss tumours in clinical trials, and is generally well tolerated, early resistance remains a significant challenge (PMID: 30037818; PMID: 34246347). This, we believe, underscores the potential clinical relevance of co-inhibiting AKT and MCL1. Specifically, AKT blockade primes PC cells for MCL1 inhibition, thus inducing rapid cell death and potentially delaying the emergence of resistant clones or hindering tumour adaptation when combined with MCL1 inhibitors. On the other hand, as the reviewer noted, the therapeutic index of MCL1 inhibitors might be compromised due to on-target toxicity. However, novel strategies already developed for other antiapoptotic BCL2 family members could be used to ensure specific targeting of MCL1 in PC cells. This has been acknowledged in the discussion, with the following addition: "It is important to acknowledge the challenges associated with direct MCL1 inhibition, particularly due to on-target cardiotoxicity (80, 81). In this context, future strategies, including antibody-drug conjugates and proteolysis-targeting chimeras (PROTACs), similar to those developed for other BCL2 family members, could enable PC-specific MCL1 targeting while minimizing cardiotoxicity risks (82, 83)" (lines 831–835).

Regarding the novelty of this combination, it should be noted that while a few previous studies have explored it, in myeloma (PMID: 32898245), leukaemia (PMID: 37222135), and breast cancer (PMID: 36241868), our study offers significant advancements in understanding its therapeutic potential and molecular mechanisms in CRPC. We have now included and acknowledged these studies in the discussion section as follows *“This therapeutic combination has been investigated in other tumour types, including myeloma (58), leukaemia (59), and breast cancer (60). However, our study highlights its therapeutic potential in CRPC and offers significant advancements in understanding the underlying molecular mechanisms”* (lines 768-771).

We incorporate clinically relevant models, including CRPC PDX-Os, mouse derived organoids, as well as AKT inhibitor, and MCL1 inhibitor-resistant cell lines. These models accurately mimic clinical scenarios, enhancing the translational impact of our findings. Mechanistically, we confirm that AKT inhibition leads to BAD dephosphorylation, priming PC cells to undergo apoptosis through MCL1 inhibition (PMID: 9346240, PMID: 24040284). However, our study also provides novel mechanistic insights into the synergy of AKT and MCL1 co-inhibition, demonstrating for the first time that AKT inhibition increases BIM-MCL1 interaction, further priming PC cells for MCL1 inhibition. This is included in the discussion section as follows *“To the best of our knowledge, we showed for the first time that AKT inhibition also increases BIM-MCL1 binding, likely in response to the displacement of BIM from BCLXL due to the competitive binding of BAD. This might contribute to increasing dependency on MCL1 in the context of AKT inhibition (and subsequent BCLXL blockade), following the “direct activation model”* (63) (lines 775-779).

The increase in the interaction between MCL1 and BIM in response to AKT inhibition has been now included in the final version of the manuscript as the bottom panel of Figure 3J, where we show the pulldown of BIM, and blot for MCL1, and in the results section as follows: *“Co-immunoprecipitation experiments revealed that ipatasertib increased BAD-BCLXL and BIM-MCL1 binding, with the latter being more evident when BIM was pulled down, in both LNCaP95 and C4-2 lines (Figure 3J)”* (lines 590-591).

Figure 3. AKT inhibition drives apoptosis when combined with MCL1 inhibition by altering the BAD-BCLXL and BIM-MCL1 interactions. Immunoprecipitation of BIM (bottom panels) in LNCaP95 (left panel) and C4-2 (right panel) treated with vehicle (DMSO), ipatasertib (1 μ M), AZD5991 (1 μ M) and combined treatment. Input (3.5%) and IgG pulldown are also depicted. BIM, and MCL1 were blotted.

Our study also addresses key gaps in the literature by identifying phospho-BAD (Ser136) levels as a potential predictive biomarker for response to AKT and MCL1 co-inhibition. This might be the reason why PC-3 (PTEN loss but low phospho-BAD levels) as well as other PTEN-deficient models in previous studies do not respond to this combination (PMID: 36241868). This information has been included in the results section *“Basal phosphorylated BAD levels strongly correlated with the degree of response to AKT and MCL1 co-inhibition across all*

models, suggesting its potential as a predictive biomarker for sensitivity to this combination therapy (Figure 4M)” (lines 636-638) and in the discussion section as follows “Indeed, our data suggest that high basal levels of phosphorylated BAD (Ser136) may be a more sensitive predictive biomarker for this combination therapy than genomic PI3K/AKT pathway aberrations (e.g., PTEN loss)” (lines 787-790).

Figure 4. AKT and MCL1 co-inhibition triggers apoptosis through dysregulation of BAD, BIM and BAK interactions in prostate cancer harbouring PI3K/AKT pathway hyperactivating aberrations. (M) Spearman correlation between caspase 3/7 activity (top panels) or cell viability at 24 hours (bottom panels) in response to ipatasertib + AZD5991 (left panels) or ipatasertib + S63845 (right panels) and basal pBAD (Ser136) levels in all cell lines, PDX-O, and ProMPT-O models. Caspase 3/7 activity is expressed as log10 fold change relative to vehicle, cell viability as fold change relative to vehicle, and pBAD levels as log10 of pBAD intensity normalized to GAPDH intensity. Spearman correlation coefficients and p-values are shown. Simple linear regression is depicted, with line (mean) and shaded area (95% CI). Caspase 3/7 activity was measured using Caspase-Glo 3/7 assays (2D for cell lines and 3D for PDX-Os/ProMPT-Os), and cell viability was measured using CellTiter-Glo assays (2D for cell lines and 3D for PDX-Os/ProMPT-Os).

Therefore, we sincerely believe that our study not only supports and confirms previous findings showing that AKT inhibition sensitizes cells to MCL1 inhibition but also expands our understanding of the underlying molecular mechanisms, thereby making a significant contribution to the field.

Reviewer’s comment 5: Based on the analysis of a limited set of cell lines and PDX-Os the authors conclude that PTEN loss increases the sensitivity to the combination of MCL1 and AKT inhibitors. However, this seems an overstatement. Comparing two cell lines with and without PTEN loss is not sufficient to support this conclusion, considering the multiple different genomic alterations acquired by these tumor cell lines. Furthermore, only PDX-Os with PTEN pathway alterations were included in the experiment.

Authors' response: We thank the reviewer for this insightful comment. To address this, we expanded our analysis to include mouse PC organoid models with and without PTEN loss. Specifically, only PTEN loss models ($Pten^{KO}/Tp53^{WT}/Myc^{OE}$ and $Pten^{KO}/Tp53^{KO}/Myc^{OE}$) responded to the combination of AKT and MCL1 inhibitors, while PTEN WT models ($Pten^{WT}/Tp53^{KO}/Myc^{OE}$) did not. PTEN loss models exhibited higher levels of phospho-BAD (Ser136), BAK, and BIM compared to PTEN WT models, all of which are required for the synergistic anti-tumour effects of the combination, as shown in the initial manuscript. These findings are now included in Figure 4H-L and added to the results section (lines 596-603) as follows: *“To confirm that PTEN loss is a key determinant of sensitivity to AKT and MCL1 co-inhibition, we treated Prostate Cancer Preclinical Mouse Modeling Platform organoids (ProMPt-Os) with different genomic backgrounds. Only Pten loss models ($Pten^{KO}/Tp53^{WT}/Myc^{OE}$ and $Pten^{KO}/Tp53^{KO}/Myc^{OE}$) responded synergistically to the combination of AKT and MCL1 inhibitors, as evidenced by a more pronounced increase in caspase 3/7 activity and reduced viability compared to single-agent treatments, while no synergy was observed in the Pten WT model ($Pten^{WT}/Tp53^{KO}/Myc^{OE}$) (Figure 4H-K). Notably, both PTEN loss models exhibited higher levels of phospho-BAD (Ser136), BAK, and BIM than the PTEN WT model (Figure 4L)”* (lines 628-635).

Figure 4. AKT and MCL1 co-inhibition triggers apoptosis through dysregulation of BAD, BIM and BAK interactions in prostate cancer harbouring PI3K/AKT pathway hyperactivating aberrations. (H-J) Prostate Cancer Preclinical Mouse Modelling Platform organoids (ProMPt-Os) were treated with vehicle (DMSO), ipatasertib (1 μ M), AZD5991 (1, 5, and 10 μ M), S63845 (1, 5, and 10 μ M), and their combinations to assess caspase 3/7 activity at 6 hours (H), cell viability at 24 hours (I), and cell viability at 96 hours (J). (L) Protein levels of MCL1, BCLXL, pBAD (Ser136), BAD, BIM, and BAK were analysed by Western blot with GAPDH as the loading control in ProMPt-Os.

Also, in response to the reviewer's comment, we included an additional human PDX-O model that does not harbour PTEN loss (CP142c), having significantly lower PI3K/AKT activity and phospho-BAD levels than the other PDX-Os tested herein. PI3K/AKT activity was inferred by "PI3K/AKT in cancer REACTOME pathway" score and AKT downstream markers including phospho-GSK3B (Ser9) and phospho-PRAS40 (Thr246). Importantly, AKT and MCL1 co-inhibition did not show synergistic effects in CP142c. These findings have been included as Figure 4C-G and Supplementary Figures 18 and 19, and added to the results section as follows *“We next analysed CRPC PDX-Os with PI3K/AKT pathway aberrations, including CP50c (AKT1 amplification), which shows the highest PI3K-AKT pathway score among the SU2C/PCF samples (34), and CP253c, CP267c, and CP336c (PTEN loss by IHC), as well as the PTEN WT*

model CP142c (Supplementary Figure 18 and 19). CP142c has significantly lower PI3K/AKT pathway activity than CP50c, CP253c, CP267c and CP336c, which was inferred by the “Reactome PI3K/AKT signalling in cancer” score and AKT downstream markers including phospho-GSK3B (Ser9) and phospho-PRAS40 (Thr246) (Figure 4C and Supplementary Figure 19). Among the five models, CP50c, CP253c, and CP142c were AR-FL and AR-V7 positive, with CP50c and CP253c also ERG positive, indicating TMPRSS2:ERG fusion (52) (Supplementary Figure 19). Synergistic caspase 3/7 activation and reduced viability were observed in response to ipatasertib combined with MCL1 inhibitors (AZD5991 and S63845) in CP50c, CP253c, CP267c, and CP336c, all of which exhibited higher basal phospho-BAD levels than CP142c, where no synergy was observed (Figure 4D-G). In contrast, BAD, BIM, and BAK levels varied across models (Figure 4D, Supplementary Figure 19)” (lines 613-626). The methods section has been updated accordingly as follows “PI3K/AKT pathway score was calculated by accumulated z-score using genes of the REACTOME_PI3K_AKT_SIGNALING_IN_CANCER pathway obtained from the molecular signature database (MSigDB) (26)” (lines 220-222).

Figure 4. AKT and MCL1 co-inhibition triggers apoptosis through dysregulation of BAD, BIM and BAK interactions in prostate cancer harbouring PI3K/AKT pathway hyperactivating aberrations. (C) Protein levels of phosphoBAD (Ser136; pBAD), total BAD, BIM, and BAK were detected by western blot, with GAPDH as a loading control, in CP50c, CP253c, CP267c, CP336c, and CP142c patient-derived

xenograft (PDX) models. **(D)** PI3K/AKT pathway activity was assessed by calculating the Reactome PI3K/AKT signalling pathway score from RNAseq data in the PDX models. Statistical analysis was performed using one-way ANOVA with post-hoc Tukey test. **(E-F)** Caspase 3/7 activity at 6 hours **(E)** and cell viability at 24 and 96 hours **(F)** in response to vehicle (DMSO), ipatasertib (1 μ M), AZD5991 (1 μ M), S63845 (1 μ M), and their combinations in CP50c, CP253c, CP267c, CP336c, and CP142c PDX-derived organoids (PDX-Os). Statistical analysis was performed using one-way ANOVA with post-hoc Tukey test. The vehicle, AZD5991, and S63845 arms (represented by dotted-pattern bars) were previously shown in Figure 2K (same experiment). **(G)** Representative microscopy images of the PDX-Os on day 4 after treatment. The scale bar indicates a length of 200 μ m. All the experiments were performed in three biological triplicates and the standard error of the mean is shown. Asterisks (* $p < 0.05$; ** $p < 0.01$; *** $p < 0.001$) indicate statistically significant differences between groups.

Supplementary Figure 14. Characterisation of CP50c, CP253c, CP267c, CP336c and CP142c CRPC PDXs. Micrographs showing phospho-GSK3B (Ser9) and phospho-PRAS40 (Thr246) levels by IHC. The scale bar indicates a length of 100 μ m.

These results reinforce our initial hypothesis and suggest that PTEN status and PI3K/AKT pathway activity play critical roles in determining the sensitivity of PC cells to AKT and MCL1 co-inhibition.

Reviewer's comment 6: *Additional in vivo experiments would be required to evaluate the antitumor efficacy of MCL1 inhibitors alone and in combinations with AKT inhibitors. Only the CP253c PDX (with minimal MCL1 expression) was used here. It would be more informative to test additional models, including the high MCL1 expressing PDXs, to see whether their increased sensitivity in vitro translate in more relevant responses in vivo to single or combined treatment. The number of mice in the PDX experiment is also minimal (n=4-5) considering the variability in individual tumor growth.*

Authors' response: In our study, we focused on CP253c, as it is representative of a commonly observed molecular subset of CRPC which might benefit from AKT and MCL1 co-inhibition. CP253c harbours PTEN loss, leading to high PI3K/AKT activation and elevated phospho-BAD (Ser163) levels, but lacks MCL1 copy number gain. Notably, we have no evidence suggesting that the proposed combination therapy would be ineffective in PI3K/AKT hyperactivated CRPCs with MCL1 copy number gain, provided they express BIM, BAK, and phosphorylated BAD. In fact, the combination may be even more effective in such tumours due to increased sensitivity to MCL1 inhibition (e.g., 22Rv1, CP50c models). Thus, CP253c aligns with the central conclusion of our study: CRPC patients with high PI3K/AKT activity (irrespective of MCL1 copy number) may benefit from co-targeting AKT and MCL1.

However, we agree with the reviewer that additional *in vivo* experiments testing MCL1 inhibition in models with MCL1 copy number gain would add valuable insights to our study. We acknowledge this as a limitation of our study and have explicitly stated it in the discussion section of the revised manuscript as follows “*A limitation of our study is the lack of in vivo validation testing the anti-tumour effects of MCL1 inhibition in models with MCL1 copy number gain*” (lines 752-753).

It is important to note that we have strengthened our findings by including an additional PDX-O model (CP142c) and three ProMPt-Os with different molecular backgrounds (Figure 4). Given the added patient derived and preclinical genetically engineered models, and the central focus of our study on the combined therapy, we kindly request that no further *in vivo* experiments be required for the acceptance of our manuscript.

Finally, we appreciate the concern regarding the number of mice used for the PDX experiment. We aimed to minimize the number of animals used, in accordance with the principles of the 3Rs while ensuring the experiments provided sufficient data to draw meaningful conclusions. While we did observe some intra-group inter-animal variability in tumour growth, especially in certain drug arms, the effects of the combined treatments (ipatasertib + S63845 and ipatasertib + CYC065) were generally consistent and robust, as evidenced in Figure 5 and 7G. These results support the efficacy of the combination therapies.

Reviewer’s comment 7: *Toxicity is a relevant issue given the limited therapeutic index of MCL1 and AKT inhibitors and should be monitored carefully by assessing multiple relevant parameters. A drop of 10-20% in the body weight, as seen with the combinations, suggests potentially relevant toxicity, unlike the authors’ statement.*

Authors’ response: We thank the reviewer for raising this important point. While we carefully monitored the mice for signs of discomfort or distress, including physical appearance (piloerection, hunched posture, poor grooming, facial expressions), behavioural changes, and physiological signs (breathing, food/water intake), we apologize for not including this information in the original manuscript. We have now added these details to the final version, specifically in the methods section as follows “*Mice were monitored for potential signs of discomfort or distress, including changes in physical appearance (facial expression, piloerection, hunched posture), behaviour (reduced activity), and physiological signs (abnormal breathing or altered food/water intake)*” (lines 403-406) and results section “*No mice exhibited weight loss exceeding 20% of their baseline or showed any signs of discomfort or distress (Supplementary Figure 20)*” (lines 651-654). This information has also been included for the combined treatment with ipatasertib and fadraciclib in lines 714-715.

To address body weight concerns, we have now represented the weight data as percentages relative to baseline and showing that no mouse experienced a drop exceeding 20% of initial body weight, which is the threshold we define as a humane endpoint, consistent with prior studies (PMID: 31665969). This new analysis has been included as Supplementary Figure 20B and 26B.

Supplementary Figure 20. Mouse body weights during the *in vivo* experiment. (A-B) Individual mouse body weights are shown for treatments with vehicle, ipatasertib, S63845, and their combination. Data are presented as raw body weight (in grams) (**A**) and as percentages relative to baseline (**B**). (**C**) Mean body weight per treatment group, with error bars representing the standard error of the mean (SEM).

Supplementary Figure 26. Mouse body weights during the in vivo experiment. (A-B) Individual mouse body weights are shown for treatments with vehicle, ipatasertib, fadraciclib, and their combination. Data are presented as raw body weight (in grams) **(A)** and as percentages relative to baseline **(B)**. **(C)** Mean body weight per treatment group, with error bars representing the standard error of the mean (SEM). Vehicle and ipatasertib arms (depicted as dotted lines and clear symbols) have been already shown in Supplementary Figure 20 (same experiment).

We therefore consider the drugs tolerable in this model, though it is worth noting that MCL1 inhibitors have lower affinity for mouse MCL1 than for human MCL1, which may contribute to the observed tolerability. This has been addressed in the discussion section “*However, this should be taken with caution as S63845 (and other MCL1 inhibitors) show less affinity for mouse MCL1 than human MCL1 (32)*” (lines 653-654). Additionally, we have included a discussion on alternative approaches, such as ADCs or PROTACs, for selectively targeting MCL1 in prostate cancer cells while sparing normal cells, as follows “*It is important to acknowledge the challenges associated with direct MCL1 inhibition, particularly due to on-target cardiotoxicity (80, 81). In this context, future strategies, including antibody-drug conjugates and proteolysis-targeting chimeras (PROTACs), similar to those developed for other BCL2 family members, could enable PC-specific MCL1 targeting while minimizing cardiotoxicity risks (82, 83)*” (lines 831-835).

Reviewer’s comment 8: *Evaluating the response of MCL1 inhibitors in cells resistant to AKT inhibitors is a nice addition to this body of work, but it seems of limited relevance in this context. Resistant cells respond to MCL1 inhibitors like the parental cells. Is this not expected? What about cells with innate or acquired resistance to MCL1 inhibitors?*

Authors’ response: We initially hypothesized that PC cells with acquired resistance to AKT inhibition would no longer respond to AKT and MCL1 co-inhibition, as we expected resistance to AKT inhibition would prevent BAD dephosphorylation, disrupting the synergy. Contrary to this, AKT and MCL1 co-inhibition continued to induce apoptosis in capivasertib-resistant cells, as AKT inhibition still led to BAD dephosphorylation, highlighting the importance of this mechanism for the observed synergy (Figure 6). These findings indicate that even after resistance to AKT inhibition, CRPCs may still respond to AKT and MCL1 co-inhibition, as long as BAD dephosphorylation is maintained (Figure 6K-L). This has important clinical implications, suggesting that patients progressing on AKT inhibitors may still benefit from the combination therapy.

To enhance clarity, we have added new plots (Figure 6C and 6D) comparing the capivasertib IC50 values, both alone and when combined with AZD5991 (1 μ M), in capivasertib-resistant and parental cell lines.

Figure 6. CRPC cells with acquired resistance to capivasertib remain sensitive to AKT and MCL1 co-inhibition. (C-D) Comparison of capivasertib IC₅₀ in presence and absence of AZD5991 (1 μM, 24 hours) between Capi-R and parental cells in LNCaP95 (C) and C4-2 (D). Statistical analysis was performed using one-way ANOVA with post-hoc Tukey test. Asterisks (* p < 0.05; ** p < 0.01; *** p < 0.001) indicate statistically significant differences between groups.

Regarding innate resistance to MCL1 inhibitors, our data show that the sensitivity to the AKT and MCL1 combination therapy is not determined by MCL1 inhibitor sensitivity. Both MCL1-sensitive and -resistant models respond to the combination, provided they have PI3K/AKT pathway hyperactivation and elevated phospho-BAD levels. For example, both parental C4-2 cells (innate resistance to MCL1 inhibition) and LNCaP95 cells (sensitive to MCL1 inhibition) showed synergy in response to the combination (Figure 2E-F and 3E-F).

In response to the reviewer's suggestion, we also generated AZD5991 and S63845-resistant LNCaP95 cells by exposing them to increasing concentrations of each drug. The resistant cells exhibited lower levels of phospho-BAD (Ser136), BIM, and BAK compared to parental cells and no longer respond to the AKT and MCL1 co-inhibition. These results highlight the risk of developing resistance to MCL1 inhibitors before administering the combination therapy, potentially limiting its efficacy, and have been included in the final version of the manuscript as Supplementary Figure 22 and in the results section as follows. "We next generated AZD5991- and S63845-resistant LNCaP95 cells and assessed their response (Supplementary Figure 22A-C). While AKT inhibition alone continued to reduce cell viability, the combination of AKT and MCL1 inhibitors no longer increased caspase 3/7 activity or synergistically decreased viability (Supplementary Figure 22D-F). The MCL1 inhibitor-resistant cell lines exhibited lower

levels of phospho-BAD, BIM, and BAK compared to parental LNCaP95 cells (Supplementary Figure 22G)” (lines 685-690).

We have also discussed these new finding in the discussion section as follows “*Importantly, we observed that cells with acquired resistance to MCL1 inhibitors no longer responded to dual AKT/MCL1 inhibition. This suggests that combining AKT and MCL1 blockade upfront, rather than after progression on MCL1 inhibitors, may be a more effective treatment strategy*” (lines 812-815). The methods section “development of drug-resistant cell lines” has been updated accordingly including MCL1-inhibitor resistant cell lines.

Supplementary Figure 22. Acquired resistance to MCL1 inhibitors abolishes the synergistic effects of AKT and MCL1 co-inhibition. (A) Schematic of how AZD5991- and S63845-resistant LNCaP95 cells were developed. **(B)** Dose-response curves for AZD5991 (left) and S63845 (right) in AZD5991-resistant

(AZD5991-R) and S63845-resistant (S63845-R) vs parental LNCaP95 cells. **(C)** Cell growth at day 6 (fold change from day 0) in AZD5991-R and S63845-R cells compared to parental cells. Statistical analysis was performed using unpaired t-tests. **(D-E)** Caspase 3/7 activity at 6 hours (left), cell viability at 24 hours (center), and 6 days (right) in AZD5991-R **(D)** and S63845-R **(E)** vs parental cells treated with vehicle (DMSO), capivasertib (1 μ M), ipatasertib (1 μ M), AZD5991 (1 μ M), or combination treatments. Two-way ANOVA with post-hoc Tukey tests was used for analysis. **(F)** Western blot comparing protein levels of phospho-BAD (Ser136), BAD, BIM, and BAK between parental, AZD5991-R, and S63845-R cells. GAPDH served as a loading control. All experiments were performed in triplicate except the western blot (n=1). Error bars represent the standard error of the mean (SEM). Statistically significant differences are indicated (***) $p < 0.001$.

We thank the reviewer for raising these important points, which prompted us to reanalyse and generate novel data that enhance the depth and quality of our study.

Reviewer's comment 9: *CDK9 inhibitors act as broad transcriptional inhibitors and induce very complex responses in tumor cells. One cannot claim they are blocking MCL1. There is no specificity for MCL1, and their effects are likely dependent on many other unrelated mechanisms. The finding of a synergistic interaction between CDK9 and MCL1 inhibitors is interesting but there is no mechanistic rationale or compelling supporting evidence. The set of data does not add to the main message, and the conclusions are highly speculative and not supported by data.*

Authors' response: We thank the reviewer for highlighting this important point. We fully agree that CDK9 inhibitors have broad transcriptional effects and are not specific to MCL1. However, our rationale for using CDK9 inhibitors stemmed from prior evidence showing their ability to downregulate MCL1 during short-term treatment (4 hours) (PMID: 31699827). Based on this, we hypothesized that short-term CDK9 inhibition could effectively downregulate MCL1 in CRPC cells and serve as an alternative strategy to trigger acute apoptosis when combined with AKT inhibition. We have included this information in the results section as follows *"CDK9 inhibition rapidly downregulates MCL1 protein levels in several tumour-types (47-49). We thus hypothesized that, CDK9 inhibitors could also trigger acute apoptosis when combined with AKT inhibitors, in CRPC cells with PI3K/AKT hyperactivation"* (lines 699-701).

Our findings support this hypothesis. Specifically, the acute synergy observed with AKT and CDK9/2 co-inhibition (e.g., increased caspase 3/7 activity and cell death) involves key apoptotic regulators such as BAK, BIM, and BAD, as silencing these genes abrogated apoptosis induction (Figure 7C). This mirrors the mechanism underlying the synergy observed with AKT and MCL1 co-inhibition (Figure 3G-H, Supplementary Figure 14 and 15).

Moreover, the PC models tested exhibited the same response profile to AKT and CDK9/2 co-inhibition as to AKT and MCL1 co-inhibition. Indeed, our new data reveal that the synergy between AKT and CDK9/2 co-inhibition occurs in PTEN-loss ProMPt-Os models but not in PTEN WT models, mirroring the response observed with AKT and MCL1 co-inhibition. These findings have been included in the final manuscript as Figure 7F and described in the results section as follows *"Furthermore, the response profile to AKT inhibitors and fadradiclib combinations mirrored that of AKT and MCL1 co-inhibition, with the same cell lines, PDX-Os and ProMPt-Os exhibiting sensitivity to both strategies (Figure 7D-F, and Supplementary Figure 25)"* (lines 709-712).

Figure 7. AKT and CDK9 co-inhibition recapitulates the effects of AKT and MCL1 co-inhibition *in vitro* and *in vivo*. (F) Prostate Cancer Preclinical Mouse Modelling Platform organoids (ProMPt-Os) were treated with vehicle (DMSO), ipatasertib (1 μ M), fadraciclib (1 μ M), and combined treatment to assess caspase 3/7 activity at 6 hours, and cell viability at 24 and 96 hours. One-way ANOVA with post-hoc Tukey test was performed. Caspase 3/7 activity and cell viability were assessed using Caspase-Glo 3/7 3D and CellTiter-Glo 3D. Asterisks (* $p < 0.05$; ** $p < 0.01$; *** $p < 0.001$) indicate statistically significant differences between groups.

Finally, new data generated for this rebuttal showed that fadraciclib does not induce acute apoptosis (no caspase 3/7 activation at 6 hours or cell death at 24 hours) in LNCaP95 cells with acquired resistance to MCL1 inhibitors. This suggests that MCL1 contributes, at least partially, to the acute effects of CDK9 inhibition, as previously reported (PMID: 31699827; PMID: 26627013; PMID: 21776020). However, longer exposures to CDK9 inhibition (6 days) resulted in similar reductions in cell viability in both parental and MCL1 inhibitor-resistant cells, likely due to the broader effects of CDK9 inhibition on multiple genes, as noted by the reviewer. Furthermore, as with AKT and MCL1 co-inhibition, AKT and CDK9 co-inhibition no longer synergize in LNCaP95 cells with acquired resistance to MCL1 inhibitors. These data have been included as Supplementary Figures 27-28, and in the results section as follows “*Finally, while AKT inhibitors combined with fadraciclib continued to synergistically induce apoptosis and cell death in C4-2 and LNCaP95 Capi-R cells, this effect was no longer observed in LNCaP95 cells with acquired resistance to MCL1 inhibitors (Supplementary Figure 27-28)*” (lines 722-725).

Supplementary Figure 28. Acquired resistance to MCL1 inhibitors abolishes the synergistic effects of AKT and CDK9/2 co-inhibition. (A-B) Caspase 3/7 activity at 6 hours (left), cell viability at 24 hours (center), and 6 days (right) in AZD5991-R (**A**) and S63845-R (**B**) vs parental cells treated with vehicle (DMSO), capivasertib (1 μ M), ipatasertib (1 μ M), fadraciclib (1 μ M), or combination treatments. Two-way ANOVA with post-hoc Tukey tests was used for analysis. All experiments were performed in triplicate. Error bars represent the standard error of the mean (SEM). Statistically significant differences are indicated (* $p < 0.05$; ** $p < 0.01$; *** $p < 0.001$).

We acknowledge that CDK9 inhibition may impact other genes and pathways, even with short-term exposure. We have consequently modified the discussion section for this point, and discuss the new data as follows “*CDK9 inhibitors rapidly downregulate MCL1 protein levels and have shown anti-tumour activity with manageable toxicity in early-phase clinical trials for haematological malignancies (53-55, 75-77). Here, we demonstrate that co-inhibition of CDK9 and AKT mirrors the effects of MCL1 and AKT co-inhibition, inducing cell death in PTEN-loss/PI3K-activated PC cells with high phospho-BAD levels. While CDK9 is a broad transcriptional regulator affecting multiple genes beyond MCL1, some studies suggest that MCL1 depletion is a key driver of the apoptosis-related anti-tumour effects of CDK9 inhibitors (53, 78, 79). Our data support that CDK9-dependent acute MCL1 downregulation contributes, at least in part, to the synergy observed with CDK9 and AKT co-inhibition. Specifically, the synergistic effects observed with both CDK9/AKT and MCL1/AKT co-inhibition strategies require the same BH3-only proteins (BIM, BAD, and BAK), and cells with acquired resistance to MCL1 inhibitors exhibit resistance to both combinations. However, we acknowledge that additional factors acutely regulated by CDK9 inhibition may also contribute to this synergy. Despite this, our findings support CDK9 inhibition combined with AKT blockade as an alternative, promising therapeutic strategy for PTEN-loss/PI3K-activated PC*” (lines 817-831).

These new results and discussion highlight a mechanistic overlap between CDK9 and MCL1 co-inhibition with AKT inhibitors, while acknowledging the broader transcriptional effects of CDK9 inhibitors.

Reviewer 3

Summary

Prostate cancer is a very common cancer in elderly men. The 5-year relative survival rate for men diagnosed with local or regional disease is almost 99%, but it drops to less than 30% for patients with distant disease. The identification of new therapeutic strategies for the treatment of advanced stages of prostate cancer is a major clinical need.

Here, Juan M. Jiménez-Vacas, Daniel Westaby and colleagues define the anti-apoptotic factor MCL1 as a valuable molecular target for the treatment of advanced prostate cancer.

MCL1 belongs to the BCL-2 family of antiapoptotic factors and is one of the most frequently amplified genes in cancer. Downregulation, degradation, or inhibition of MCL-1 have already been demonstrated a promising strategy to overcome tumor resistance to standard and targeted therapies. Potent MCL-1i are now tested in clinical trials, however identification of molecular mechanisms and signaling pathways favoring or limiting their anti-tumor activity will be crucial to capitalize on MCL-1 targeting.

Leveraging on a broad spectrum of in vitro and in vivo prostate cancer models, the authors demonstrate the efficacy of the pharmacological inhibition of MCL1 in prostate tumors characterized by MCL1 gene amplification and PI3K/AKT overactivity. The synergistic effect of combining MCL1 inhibitors with AKT inhibitors promotes apoptosis in both therapy-naïve and therapy-resistant PC models, in vitro as well as in vivo. Mechanistically, synergy converges on the interactions between BCL2 and BH3 proteins, promoting the formation of BAD-BCLXL and MCL1-BIM complexes and, consequently, unlocking the apoptotic process.

The manuscript is a solid demonstration of the relevance of combining molecular targeting and cancer stratification based on genetic and molecular criteria. It is well written (although some words such as orthogonal and synergy are used too frequently), the data presented are convincing and support the conclusions.

Below are a few suggestions that can improve the work.

These are mainly minor revisions, with the exclusion of points 11, 14, 15 and 16 for which I would ask the authors to pay particular attention.

Authors' Response: We thank the reviewer for acknowledging that our manuscript is well-written, identifies MCL1 as a valuable therapeutic target for advanced prostate cancer (PC), and highlights the importance of combining molecular targeting with cancer stratification based on genetic and molecular profiles. We hope the actions detailed below have sufficiently addressed their comments and questions.

Reviewer's comment 1: *The manuscript is too dense with data that sometimes weigh down the reading and the reasoning. I would suggest streamlining the work by removing some of the data or moving it to supplementary.*

Authors' response: We appreciate the reviewer's feedback regarding the density of the manuscript and understand the importance of streamlining the presentation of data to enhance readability and clarity. We have made several modifications to address this concern:

We have removed subsection “3.4 MCL1 RNA associates with key signalling pathways in mCRPC” from the Results section, along with the associated graphs (previously Figure 1L-M), to streamline the narrative.

Figures 2 and 3 have been reorganized for clarity and focus: Figure 2 now focuses exclusively on the effects of MCL1 inhibition as a single agent strategy. Figure 3 is dedicated to PI3Ki/AKTi + MCL1i combined treatment, incorporating the FDA screen results as new Figure 3A-B.

To simplify the comparison of synergy scores for the MCL1 inhibitor AZD5991 with AKT or PI3K inhibitors, we now focus on LNCaP95 cells as a representative model in Figure 3C-D. Results obtained using the C4-2 model, which yielded similar findings, have been moved to the supplementary material to improve the flow of the manuscript.

While we have streamlined the manuscript as much as possible, we believe the remaining data in the results section and main figures are essential for conveying the full scope and rationale of our findings. However, we remain open to further adjustments and would be happy to move specific data from the main figures to the supplementary material if the reviewer has additional recommendations.

Reviewer’s comment 2: *I would stratify the 4 patients with MCL1 gain/amplification in two distinct categories, partially sensitive (n=2) and insensitive (n=2).*

Authors’ response: We fully agree with the reviewer’s suggestion. We have stratified the five models (including the four originally presented and the newly added CP142c model) into two categories “partially sensitive (n=2) and insensitive (n=3)” in Figure 2I-K.

Reviewer’s comment 3: *Supplementary Figure 1 and 2 are not mentioned in the manuscript, Supplementary Figure 4 is mentioned before Supplementary Figure 3.*

Authors’ response: Supplementary Figures 1 and 2 are referenced in the Materials and Methods section (lines 227-228), while Supplementary Figure 3 is first mentioned in lines 275-276, and Supplementary Figure 4 is introduced later in line 477. We have carefully reviewed the text to ensure all supplementary figures are mentioned in the correct order in the revised manuscript.

Reviewer’s comment 4: *Absence of MCL1 gain/amplification association with Taxane response is interesting and should be discussed. Is the mechanism of mitotic catastrophe the possible reason?*

Authors’ response: We hypothesize, as suggested by the reviewer, that the lack of association between MCL1 gain/amplification and Taxane response could be explained by the potential ability of taxanes to induce mitotic catastrophe in an MCL1-independent manner in CRPC. This is a compelling hypothesis that warrants further investigation and will be a focus of our future studies.

This has been incorporated into the discussion section as follows “*However, no association was observed between MCL1 gain/amplification and taxane response, which could potentially*

be explained by the ability of taxanes to induce MCL1-independent mitotic catastrophe in these tumours (56, 57)" (lines 742-745).

We thank the reviewer for bringing up this important point, which has enhanced the quality of our discussion.

Reviewer's comment 5: *Mito localization of MCL1 should be tested by immunofluorescence with a specific marker of mitochondria. Optical density data are not convincing. This analysis should be normalized for cell size. Metastatic cells in the lymph node appear smaller than those in the lung.*

Authors' response: To address the comment, we examined the mitochondrial localization of MCL1 using immunofluorescence (IF) with mitotracker as a mitochondrial marker in 22Rv1 cells, which express high levels of MCL1 as shown in this study (Figure 2C).

The results show significant colocalization of MCL1 with mitochondria, evident from the orange signal in the mitotracker, MCL1, and DAPI merged channel. However, we also observed MCL1 staining that does not colocalize with mitotracker (green signal). This non-mitochondrial MCL1 staining appears as aggregates and may indicate its association with other membranous or vesicular organelles, such as the endoplasmic reticulum (ER), as previously reported (PMID: 26538029). This observation aligns with the short half-life and high turnover rate of MCL1, and is consistent with the role of the ER in synthesizing and processing proteins before their transport to the Golgi apparatus for further modification and sorting (PMID: 35452617). We thank the reviewer again for encouraging us to investigate this aspect of MCL1 biology. While the localization of MCL1 is indeed an intriguing topic, we believe it lies beyond the scope of this study. Therefore, we kindly request that no further analyses of MCL1 localization be required for the publication of our study. Representative images of the IF experiment are provided below for the reviewer's reference.

Figure for Reviewer's Purpose: Immunofluorescence (IF) experiment performed in 22Rv1 cells. Nuclei are stained with DAPI (red), E-cadherin (1:100; #14472, Cell Signaling) is shown in purple, mitochondria are labelled with MitoTracker™ Deep Red FM (red; used according to

the manufacturer's instructions, M22426, Thermo Fisher Scientific), and MCL1 (1:250; 16225-1-AP, Proteintech) is depicted in green. Alexa Fluor® 594 (1:1000) was used as secondary antibody. The scale bar indicates 10 µm

We agree that optical density (OD) should be normalized for cell size. Indeed, for this study we used HALO AI software, trained and supervised by expert genitourinary pathologist (Dr. B. Gurel), to calculate OD data as the cytoplasmic signal intensity normalized per cell (accounting for cell size) and then averaged across the number of tumour cells identified per sample. Dr. Gurel independently reviewed and verified these results. Also, OD significantly correlated with the visual H-Score (Supplementary Figure 5B), which was calculated independently and in a blinded fashion by Dr. Gurel, supporting OD data reliability. However, we apologize for not clearly describing this methodology in the initial submission and have now included these details in the methods section as follows *"The analysis algorithm was then adjusted to provide continuous data on the intensity of MCL1 cytoplasmic signal intensity normalized per cell (accounting for cell size), averaged across the number of tumour cells identified per sample, providing a value between 0 (no staining) and 1 (black) called optical density (OD)"* (lines 236-239).

We also rechecked the micrographs and scale bars, confirming their accuracy. The observed variation in cell size likely reflects differences in tumour sites, a phenomenon we have noted in previous studies. We have added two additional representative micrographs to Figure 1L to better illustrate the range of optical densities.

Figure 1. MCL1 (1q21) copy number gain/amplification is common in CRPC, occurring early in tumour evolution and associating with worse overall survival. (K) IHC for MCL1 was undertaken on 30 mCRPC biopsies from the RMH cohort with pre-existing RNA-sequencing data. Spearman correlation between MCL1 RNA and protein (OD) expression was performed. **(L)** MCL1 IHC representative micrographs. The scale bar indicates a length of 100 µm.

Reviewer's comment 6: *R value for the correlation of MCL1 with RELA and STAT3 transcripts is minimal. The association between MCL1 transcript amounts and TNFa/IL6-STAT signaling via SASP is speculative and not supported by scientific evidence. Section 3.4 MCL1 RNA associates with key signaling pathways in mCRPC is poorly informative, I would suggest removing this part from the results.*

Authors' response: In line with the reviewer's suggestion, we have removed Section 3.4. MCL1 RNA associates with key signalling pathways in mCRPC, from the Results section, as well as the associated graphs from Figure 1. We believe this enhances the overall clarity and focus of the manuscript, streamlining the study to better highlight the key findings. We appreciate

the reviewer's recommendation, which has contributed to improving the readability and scientific rigor of the paper.

Reviewer's comment 7: *Nucleolar staining of MCL1 in PDX-O is surprising. Is it specific?*

Authors' Response: We thank the reviewer for pointing this out. As with all antibodies used in our laboratory IHC in patient samples, we thoroughly validated the MCL1 antibody (16225-1-AP, Proteintech) through multiple approaches. Specifically, we confirmed a single band in western blot analyses that disappeared in response to MCL1 siRNA. We then performed IHC on paired cell pellets from the same experiment, which showed a significant reduction in staining after MCL1 siRNA treatment. Importantly, we did not observe any nucleolar staining in these cell pellets, as demonstrated in Supplementary Figure 1B.

To further address the reviewer's comment, we stained six different PDX models, each represented by two samples (independent passages). In three of these models, we observed nucleolar staining of MCL1, while the remaining three showed no such staining.

Although we cannot definitively confirm the specificity of the nucleolar staining, we hypothesize that it may represent an artefactual signal, as we have occasionally observed similar patterns with other antibodies. Importantly, nucleolar staining was not quantified or included in our analyses. Our study focused exclusively on cytoplasmic MCL1 staining. We have now clearly indicated in the axes of Figures 1K, 5D, and 7H, as well as in the legend for Figure 2J.

Figure for Reviewer's Purpose. MCL1 immunohistochemistry performed on six patient-derived xenograft (PDX) models, with each model represented by two samples from independent passages. The top three models (PDX 1-3) exhibit MCL1 nucleolar staining, while the bottom three models (PDX 4-6) show no MCL1 nucleolar staining. The scale bar represents 100 μm .

Reviewer's comment 8: *Caspase assay in CP267c shows Caspase 3/7 activity similar to those measured in CP336c and CP253c, however cell viability after 96 hours of MCL1 inhibition is substantially different. Authors should explain this inconsistency. Moreover, why inhibition of an anti-apoptotic factor such as MCL1 should be sufficient to trigger CRPC cell death?*

Authors' response: We thank the reviewer for their insightful comment. A possible explanation for this observation is caspase activation without apoptosis, a phenomenon that has been previously reported (PMID: 36669100; PMID: 19460165). We hypothesise that the threshold of caspase 3/7 activation required to induce apoptosis differs across the models herein tested. For example, while a 2.5-fold increase in caspase 3/7 activity may be sufficient to trigger apoptosis in CP267c, the same level of activation might be sublethal in CP253c and CP336c. This hypothesis aligns with previous findings showing that intermediate levels of caspase activity can result in divergent cell fates (PMID: 36669100).

Regarding the reviewer's second question on why MCL1 inhibition is sufficient to trigger CRPC cell death, we hypothesize that certain CRPC tumours exhibit an increased dependence on MCL1 for survival. Specifically, we believe that PC cells with MCL1 overexpression rely heavily on this protein to counteract pro-apoptotic signals and maintain survival. Consequently, inhibiting MCL1 disrupts this balance, removing a critical survival signal and resulting in apoptosis. This phenomenon has been previously postulated in other tumour-types (PMID: 20164920; PMID: 32913197; PMID: 31856269).

In summary, we propose that the dependency on MCL1 is context-dependent and reflects the delicate balance between pro-apoptotic and anti-apoptotic factors. In models with MCL1 overexpression, MCL1 inhibition effectively disrupts this balance, leading to cell death. This information has been included in the discussion section as follows "*We show for the first time that CRPC models with MCL1 copy number gain respond to MCL1-targeting BH3 mimetics, likely due to their increase reliance on MCL1 for survival, with inhibition inducing apoptosis, as previously observed in other tumour-types (7, 30, 47)*" (lines 749-751).

Reviewer's comment 9: *Why did the authors use cell lines for FDA-approved drug screening instead of using PDX-Os? Differently by PDX-Os, prostate cancer cell lines respond little (LNCaP95) or not at all (LNCaP, C4-2, 22RV1, PC3, Du145) to MCL1 inhibitors administered as a single treatment (Figure 4 and Supplementary Figure 7 Vehicle plus MCL1 inhibitor). Definition of MCL1 status (gene/mRNA/protein) in PC cell lines would help in the interpretation of the results obtained.*

Author's response: The decision to use PC cell lines for the FDA-approved drug screen, rather than PDX-Os, was based on logistical constraints, as PDX-Os require 2-3 months for tumour growth and often provide insufficient material for such drug screens. We then validated the effects of AKT/MCL1 and AKT/CDK9 co-inhibition in five independent PDX-O models, including four responders (CP50c, CP253c, CP267c, CP336c) and one non-responder (CP142c), proving the robustness of our data, as shown in Figures 4E-G and 7E.

Following the reviewer's suggestion, we characterized the cell lines used in this study to assess MCL1 copy number, MCL1 protein expression levels, and their response to MCL1 inhibition. Our data reveal that only 22Rv1 and DU145 harbour MCL1 copy number gains. Moreover, 22Rv1 cells exhibited higher MCL1 protein levels compared to the other cell lines, suggesting that additional mechanisms, such as mRNA translation or protein stability, may buffer or modulate MCL1 protein levels in DU145 cells. Consistent with our findings in PDX-Os with high MCL1 protein levels, 22Rv1 cells were sensitive to MCL1 inhibition. In contrast, despite

harbouring MCL1 copy number gains, DU145 cells demonstrated resistance to MCL1 inhibition. This resistance may be attributed to the absence of the mitochondrial pore-forming protein BAX, a feature previously reported to be associated with apoptosis resistance in these and other cell types (PMID: 11326099; PMID: 15757910). Finally, although LNCaP95 cells neither harboured MCL1 copy number gains nor exhibited high MCL1 protein levels, they were nearly as sensitive to MCL1 inhibition as 22Rv1 cells. This observation suggests that alternative mechanisms drive sensitivity to MCL1 inhibition, opening new avenues for investigation (Figures 2D-F).

Interestingly, it should be noted that we used fresh cells to characterize the cell lines. Unlike 22Rv1 cells, which now exhibit sensitivity to MCL1 inhibition, the behaviour of the other cell lines was consistent with our previous findings. The differences observed in 22Rv1 may reflect the acquisition of apoptosis resistance, a phenomenon previously shown to be associated with prolonged passaging (PMID: 18262222). While we acknowledge that this warrants further investigation, it is beyond the scope of the current study.

We appreciate the reviewer's comments, which prompted us to generate new data that significantly enhance the manuscript. These new data have been included as Figure 2A-F and in the results section as follows "We first characterized seven prostate-derived cell lines and found that 22Rv1 and DU145 harbour MCL1 copy number gain; however, only 22Rv1 exhibited markedly elevated MCL1 protein levels, suggesting additional mechanisms regulate MCL1 protein levels in DU145 (Figure 2A-C). Among the tested cell lines, 22Rv1 showed the highest sensitivity to MCL1 inhibition, as evidenced by increased caspase 3/7 activity and reduced cell viability at 24 hours and 6 days (Figure 2D-F). LNCaP95 displayed a comparable response, indicating that sensitivity to MCL1 inhibition can be influenced by factors beyond basal MCL1 copy number or protein levels (Figure 2D-F). The lack of response in DU145 may stem from its reported apoptosis resistance due to BAX deficiency (48) (Figure 2D-F)" (lines 514-523).

Figure 2. PC models with MCL1 copy number gain are sensitive to MCL1 inhibition. (A) Violin plot showing the MCL1:Centromere (1p12) ratio in PNT2, LNCaP, C4-2, LNCaP95, 22Rv1, PC-3 and DU145 cell lines (n=50 cells). **(B)** Representative images of MCL1 Fluorescence in situ hybridization (FISH) in

prostate-derived cell lines. The median of MCL1:Centromere (1p12) ratio is depicted. The scale bar indicates a length of 5 μm . **(C)** The protein levels of MCL1, BCLXL, and BCL2 in PNT2, LNCaP, C4-2, LNCaP95, 22Rv1, PC3, and DU145 cell lines, with GAPDH as a loading control, were detected by western blot. **(D)** The impact of AZD5991 (1 μM ; left-hand graph) and S63845 (1 μM , right-hand graph) on caspase 3/7 activity was determined using the Caspase-Glo 3/7 2D (at 6 hours) in PNT2, LNCaP, C4-2, LNCaP95, 22Rv1, PC3, and DU145 cell lines. **(E-F)** Cell viability in PNT2, LNCaP, C4-2, LNCaP95, 22Rv1, PC3, and DU145 cell lines after six doses (highest at 1 μM , decreasing two-fold) of AZD5991 (left) or S63845 (right) was assessed with CellTiter-Glo 3D at 24 hours **(E)** and 6 days **(F)**. The experiment was performed in biological triplicates with five three technical replicates, and the data are presented as fold change relative to the vehicle (DMSO).

Reviewer's comment 10: The analyses shown in Figure 2 do not demonstrate a synergistic effect of concomitant inhibition of PI3K and MCL1. Figure 3A formally demonstrates synergy.

Authors' response: We thank the reviewer for raising this point. The analyses and graphs now depicted in Figure 3A-B (previously Figure 2) were designed to integrate the substantial amount of data derived from the large-scale FDA-approved drugs screen. To improve the clarity and interpretability of the synergistic effects, we have included the raw data from the FDA-approved drugs screen in Supplementary Figure 8. This Supplementary figure shows the effect of AZD5991 as a single agent, as well as the effects of each FDA-approved drug as single agents and in combination with AZD5991 on caspase 3/7 activity and cell viability (after 24 hours).

Supplementary Figure 8. Effect of 166 FDA-approved drugs in absence and presence of AZD5991 on caspase 3/7 activity and cell viability. Caspase 3/7 activity was measured at 6 hours using Caspase-Glo 3/7 2D, and cell viability at 24 hours using CellTiter-Glo 2D. FDA-approved drugs were tested at 5 μM , and AZD5991 at 1 μM . Black bars represent FDA-approved single-agent effects, and blue bars

represent effects in combination with AZD5991. The screen was conducted as a single biological experiment (n=1) without technical replicates. **The figure has been cropped for space considerations, ensuring that PI3K inhibitors are included. For the full figure, please refer to Supplementary Figure 8.*

Reviewer's comment 11: *Western blot analysis shows less cleaved Caspase 3 in the combo Ipa + AZ compared to Cap + AZ in all triplicates of both cell lines (Suppl. Figure 8C). This is nicely recapitulated by siMCL1 plus Ipa versus siMCL1 plus Cap in Suppl Figure 10. Nevertheless, PARP cleavage and cells viability are comparable in cells treated with Ipa+AZ or Cap+AZ. Did the authors check for the activation of Caspase 7? Is active Caspase 7 more similar in PC cells treated with the different combo?*

Authors' response: Following the reviewer's suggestion, we measured cleaved caspase 7 levels in response to capivasertib, ipatasertib, AZD5991, and their combinations using the same samples analysed in Figure 3E-F. Our results show that both combinations (capivasertib+AZD5991 and ipatasertib+AZD5991) increased cleaved caspase 7 to a similar extent, indicating comparable caspase 7 activation. Figures 3E, 3F, and Supplementary Figure 12A have been updated accordingly. Details about the cleaved caspase 7 antibody used have been added to Supplementary Table 6.

Figure 3. AKT inhibition drives apoptosis when combined with MCL1 inhibition by altering the BAD-BCLXL and BIM-MCL1 interactions. (E-F) Protein expression (cleaved PARP, cleaved caspase 7, cleaved caspase 3, and p-AKT^{Ser473}; 6 hours; right panel) between vehicle (DMSO), capivasertib (1 μ M), ipatasertib (1 μ M), and combination of capivasertib (1 μ M) and ipatasertib (1 μ M) with AZD5991 (1 μ M) in LNCaP95 (E) and C4-2 (F). One-way ANOVA with post-hoc Tukey test was performed.

Reviewer's comment 12: *AKT competitive inhibitors are known to induce AKT phosphorylation at both T308 and S473 sites. This is clearly shown in WB of C4-2 (Suppl Figure 10F) and LNCaP95 (10E). In LNCaP95, however, phosphorylation increases exceptionally when AKT inhibitors are combined with MCL1 knockdown. This is an interesting aspect that should be considered, especially since it does not occur when AKT inhibitors are combined with AZ (Figure 3B).*

Authors' response: We thank the reviewer for highlighting this very interesting observation. Our results suggest that MCL1 inhibition (via BH3 mimetics) and MCL1 depletion (via siRNAs) have distinct effects, when combined with AKT inhibitors, on AKT phosphorylation at Ser473. This is plausible, as MCL1 is known to have additional functions beyond its primary anti-apoptotic role, some of which are mediated through different domains. For instance, while BH3 mimetics specifically target the BH3-binding domain of MCL1, other regions, may remain active and interact with other proteins, potentially triggering alternative functions (PMID: 10978339).

Furthermore, it is noteworthy that MCL1 depletion has been shown to shift cellular fuel usage from lipids to glucose independently of its anti-apoptotic function (PMID: 36198266). Given the well-established interplay between AKT signalling and glucose metabolism, it is conceivable that metabolic dysregulation induced by MCL1 depletion, combined with AKT inhibition, could activate feedback mechanisms leading to increased AKT phosphorylation.

While this phenomenon is outside the primary scope of our study, we agree that it warrants further investigation. We thank the reviewer for bringing this intriguing aspect to our attention that we will indeed explore in future projects.

Reviewer's comment 13: *Authors should highlight with a mark and corresponding explanation in the figure legend the technical issues that occurred in the WB of Suppl. Fig. 12 (e.g., LNCaP95-A: Ipa+AZ/BAD siRNA, BAK levels are strongly reduced; LNCaP95-B: Ipa+AZ/BAD siRNA, BIM levels are strongly reduced; LNCaP95-C: Veh/Scr the amount of BAK is too low compared to the amount of BAK in BIM and BAD siRNA lines).*

Authors' response: In accordance with the reviewer's recommendation, we have marked the mentioned Western blot membranes in Supplementary Figure 15 (previously Supplementary Figure 12). We have also updated the figure legend to include the following explanation: "*Asterisks on the membranes indicate technical problems, likely due to suboptimal transfer at membrane edges*".

We appreciate the reviewer's attention to detail, which has helped us improve the transparency of our data presentation.

Reviewer's comment 14: *As expected, AKTi increases the BAD/BCLXL interaction. In contrast, the increase in MCL1/BIM interaction is not equally evident in the IP experiments reported in panel 3E. The PLA data described in Suppl. Fig. 13 support the minimal, if any, change in MCL1/BIM interaction in the presence of Ipa. This is a critical point, as these data may represent the molecular explanation for the synergy between AKT and MCL1 inhibition. The authors should produce more convincing data to support their conclusions. Moreover, because BAK is a main target of MCL1 antiapoptotic activity, the authors should analyze the interaction between MCL1 and BAK in PC cells untreated and in presence of Ipa, AZ, or their combination.*

Authors' response: We have repeated some Co-IP experiments to better illustrate the changes reported in this study. Specifically, the increase in BIM-MCL1 binding is more evident when we pulled down BIM and blotted for MCL1. These data, together with the prevention of apoptosis induced by AKT and MCL1 co-inhibition upon BIM knockdown, suggest that

increased BIM-MCL1 interaction in response to AKT inhibition primes cells for MCL1 inhibition. Additionally, as recommended by the reviewer, we examined the interaction between BAK and MCL1. We found that MCL1 inhibition, either as a single agent or in combination with AKT inhibition, disrupts MCL1-BAK binding, consistent with its mechanism of action (PMID: 30559424). This new information has been included in the results section as follows: “Co-immunoprecipitation experiments revealed that ipatasertib increased BAD-BCLXL and BIM-MCL1 binding, with the latter being more evident when BIM was pulled down, in both LNCaP95 and C4-2 lines (Figure 3J). Additionally, AZD5991 (as a single treatment or combined with ipatasertib) disrupted BIM-MCL1 and BAK-MCL1 binding (Figure 3J), in keeping with its mechanism of action (31)” (lines 589-593). Figure 3J has been updated accordingly.

Figure 3. AKT inhibition drives apoptosis when combined with MCL1 inhibition by altering BAD-BCLXL and BIM-MCL1 interactions. (J) Immunoprecipitation of BCLXL (top panels), MCL1 (central panels) and BIM (bottom panels) in LNCaP95 (left panels) and C4-2 (right panels) treated with vehicle (DMSO), ipatasertib (1 μ M), AZD5991 (1 μ M) and combined treatment. Input (3.5%) and IgG pulldown are also depicted. BAD, BCLXL, BIM, BAK and MCL1 were blotted.

Reviewer’s comment 15: PTEN IHC in Figure 4C should be accompanied with pSer473-AKT staining (Cell Signaling antibodies against pAKT work wonderfully on FFPE sections). A PDX with no alteration in PTEN and PI3K pathway should be used to demonstrate AKT hyper-activation in CP50c, CP253c, CP267c, CP336c.

Authors’ response: We thank the reviewer for this insightful comment. We agree that incorporating markers of AKT signalling activation is essential to assess the activity status of this pathway in the patient-derived models used in our study.

To maintain consistency throughout the manuscript and given that pGSK3 β and pPRAS40 antibodies are already validated in our laboratory, we performed IHC analyses for these downstream markers of AKT activity in the specified PDX-Os samples. Our results demonstrate that CP50c, CP253c, CP267c, and CP336c exhibit elevated levels of these AKT activity markers

compared to the newly included PTEN wild-type model, CP142c, as suggested by the reviewer. These IHC micrographs have been included in Supplementary Figure 19.

Supplementary Figure 19. Characterisation of CP50c, CP253c, CP267c, CP336c and CP142c CRPC PDXs. Micrographs showing phospho-GSK3B (Ser9) and phospho-PRAS40 (Thr246) levels by IHC. The scale bar indicates a length of 100 μ m.

We also analysed RNA-seq data to evaluate the activation of the PI3K/AKT pathway in these samples. Specifically, our results show that CP50c, CP253c, CP267c, and CP336c have a higher "Reactome PI3K/AKT Signaling in Cancer" score compared to CP142c (Figure 4C). This information has been included as Figure 4C and Supplementary Figure 19, and added to the results section as follows "We next analysed CRPC PDX-Os with PI3K/AKT pathway aberrations, including CP50c (AKT1 amplification), which shows the highest PI3K-AKT pathway score among the SU2C/PCF samples (33), and CP253c, CP267c, and CP336c (PTEN loss by IHC), as well as the PTEN WT model CP142c (Supplementary Figure 18 and 19). CP142c has significantly lower PI3K/AKT pathway activity than CP50c, CP253c, CP267c and CP336c, which was inferred by the "Reactome PI3K/AKT signaling in cancer" score and AKT downstream markers including phospho-GSK3B (Ser9) and phospho-PRAS40 (Thr246) (Figure 4C and Supplementary Figure 19)" (lines 613-620).

Figure 4. AKT and MCL1 co-inhibition triggers apoptosis through dysregulation of BAD, BIM and BAK interactions in prostate cancer harbouring PI3K/AKT pathway hyperactivating aberrations. (D) PI3K/AKT pathway activity was assessed by calculating the Reactome PI3K/AKT signalling pathway score from RNAseq data in the PDX models. Statistical analysis was performed using one-way ANOVA with post-hoc Tukey test. Asterisks (* $p < 0.05$; ** $p < 0.01$; *** $p < 0.001$) indicate statistically significant differences between groups.

Additionally, we investigated the potential effects of AKT and MCL1 co-inhibition in the CP142c model, which exhibits low PI3K/AKT pathway activity. Our results reveal no synergistic effect of this co-inhibition in CP142c. This information has been included as Figure 4C-G and in the results section as follows “*Synergistic caspase 3/7 activation and reduced viability were observed in response to ipatasertib combined with MCL1 inhibitors (AZD5991 and S63845) in CP50c, CP253c, CP267c, and CP336c, all of which exhibited higher basal phospho-BAD levels than CP142, where no synergy was observed (Figure 4D-G)*” (lines 622-625).

Figure 4. AKT and MCL1 co-inhibition triggers apoptosis through dysregulation of BAD, BIM and BAK interactions in prostate cancer harbouring PI3K/AKT pathway hyperactivating aberrations. (E-F) Caspase 3/7 activity at 6 hours (E) and cell viability at 24 and 96 hours (F) in response to vehicle (DMSO), ipatasertib (1 μ M), AZD5991 (1 μ M), S63845 (1 μ M), and their combinations in CP50c, CP253c, CP267c, CP336c, and CP142c PDX-derived organoids (PDX-Os). Statistical analysis was performed using one-way ANOVA with post-hoc Tukey test. The vehicle, AZD5991, and S63845 arms (represented by dotted-pattern bars) were previously shown in Figure 2K (same experiment). (G) Representative microscopy images of the PDX-Os on day 4 after treatment. The scale bar indicates a length of 200 μ m. All the experiments were performed in three biological triplicates and the standard error of the mean is shown. Asterisks (* $p < 0.05$; ** $p < 0.01$; *** $p < 0.001$) indicate statistically significant differences between groups.

We appreciate the reviewer’s suggestion, which has significantly strengthened our findings and provided a more comprehensive analysis of the PI3K/AKT pathway across the models

Reviewer’s comment 16: The plot in Figure 5D describes Cleaved Caspase 3 quantification in PDX treated with vehicle, Ipa, S63, or their combo. Ipa+S63 treatment shows heterogeneous results with three of the five samples characterized by similar amounts of Cleaved Caspase 3-

positive cells compared to Veh, Ipa or S63. Similar variability is evident for the immunoscore of pPRAS40. If GSK3b is more sensitive to minimal AKT inhibition than PRAS40, could the results suggest reduced Ipa activity in non-responsive PDX treated with the combination? The authors should comment on this variability.

Authors' response: As suggested by the reviewer, we examined whether the tumours exhibiting high caspase 3/7 activity following the combined treatment with ipatasertib and AZD5991 were also the ones with lowest levels of pPRAS40. However, our analysis revealed no such correlation, as illustrated in the figure provided for the reviewer's reference (see below).

Figure for Reviewer's Purpose. Percentage of caspase 3-positive cells (left y-axis) and cytoplasmic pPRAS40 H-Score (right y-axis; **A**) and pGSK3B (right y-axis; **B**) measured by IHC in five individual CP253c tumours treated with ipatasertib and S63845 combination therapy. IHC analyses were performed on terminal tumours collected 6 hours after the final dose.

We attribute the observed variability to the timing of the IHC analyses, as they were performed on terminal tumours, which may not be optimal for assessing molecular changes. This information is included in the methods sections as follows "The experiment concluded either 24-days after initial treatment or when tumours reached a size of approximately 1200 mm³. Mice were euthanized 6-hours after the final dose, and tumour samples were collected for IHC analysis" (lines 406-408) and in the results section (line 654-655, and line 716) to ensure transparency.

Reviewer's comment 17: The sentences "Caspase 3/7 was induced" or "a synergistic increase in caspase 3/7 levels" are incorrect. The authors estimated the activity (or activation) of caspase 3/7, or the presence of the active form of the protein.

Authors' response: We fully agree with the reviewer and apologise for this mistake. This has been corrected in the final version of the manuscript (lines 434, 622, 655, and 674).

Reviewer #1 (Remarks to the Author): The authors have comprehensively addressed all comments I raised.

Reviewer #2 (Remarks to the Author): The author's response to my comments is satisfactory. They have performed additional experiments including multiple models. They have also amended the test when necessary to clarify all the issues that I have raised. I found that the manuscript is improved.

Reviewer #3 (Remarks to the Author): The authors responded thoroughly to all my comments and suggestions.

Authors' response: We would like to sincerely thank the reviewers once again for their valuable comments and suggestions, which have significantly improved our work. We are pleased that our revisions, additional experiments, and responses addressed all remaining concerns and fully clarified the points raised.